# The natural function of the malaria parasite's chloroquine resistance transporter

Sarah H. Shafik[1], Simon A. Cobbold [2], Kawthar Barkat[1], Sashika N. Richards[1], Nicole S. Lancaster[1], Manuel Llinás [3], Simon J. Hogg [1], Robert L. Summers[1], Malcolm J. McConville [2] & Rowena E. Martin [1✉]

The *Plasmodium falciparum* chloroquine resistance transporter (PfCRT) is a key contributor to multidrug resistance and is also essential for the survival of the malaria parasite, yet its natural function remains unresolved. We identify host-derived peptides of 4-11 residues, varying in both charge and composition, as the substrates of PfCRT in vitro and in situ, and show that PfCRT does not mediate the non-specific transport of other metabolites and/or ions. We find that drug-resistance-conferring mutations reduce both the peptide transport capacity and substrate range of PfCRT, explaining the impaired fitness of drug-resistant parasites. Our results indicate that PfCRT transports peptides from the lumen of the parasite's digestive vacuole to the cytosol, thereby providing a source of amino acids for parasite metabolism and preventing osmotic stress of this organelle. The resolution of PfCRT's native substrates will aid the development of drugs that target PfCRT and/or restore the efficacy of existing antimalarials.

[1] Research School of Biology, The Australian National University, Canberra, ACT 2601, Australia. [2] Bio21 Molecular Science and Biotechnology Institute, University of Melbourne, Melbourne, VIC 3052, Australia. [3] Department of Biochemistry and Molecular Biology, Department of Chemistry, and Huck Center for Malaria Research, The Pennsylvania State University, University Park, PA 16802, USA. ✉email: rowena.martin@anu.edu.au

To grow and multiply within the erythrocyte, the malaria parasite takes up the host cell cytosol into an acidic compartment known as the digestive vacuole (DV, pH 5.0–5.5[1,2]). The internalised host proteins, 50–80% of which are haemoglobin[3,4], are degraded by DV-localised proteases into peptides and amino acids[5]. The digestion of the erythrocyte cytosol provides space[6] and amino acids[7] for parasite growth and multiplication, and helps maintain the osmotic balance of the parasitised cell[8,9]. Consistent with its crucial role in the biology of the parasite, the DV is also where many antimalarials act, accumulate and/or are activated[10].

The *P. falciparum* 'chloroquine resistance transporter' (PfCRT) resides in the DV membrane and is pivotal to the normal physiology of this organelle as well as to its extensive involvement in drug resistance[11–13]. PfCRT was originally identified as the main determinant of chloroquine (CQ) resistance, but mutations in this protein are now known to modulate the parasite's sensitivity to many current and upcoming antimalarials[14]. The main mechanism by which these mutations confer drug resistance is through enabling the transporter to efflux a drug, such as CQ, from the DV and thus away from its primary site of action[10,15,16].

PfCRT is also essential for the survival of the parasite and has itself been identified as a drug target[17–19]. For instance, potent inhibition of mutant PfCRT results in parasite death[20,21] and downregulation of PfCRT expression causes impaired growth and DV swelling[22]. Furthermore, the mutations that confer drug resistance typically impose a fitness cost upon the parasite, the severity of which is dependent upon the number and nature of the mutations[23–27]. For example, the 'Dd2' isoform (PfCRT$^{Dd2}$) imparts a significant fitness cost (and a high level of CQ resistance), whereas the 'Ecu1110' isoform (PfCRT$^{Ecu1110}$) imposes a smaller fitness disadvantage but also confers a lower level of CQ resistance. Moreover, two PfCRT mutations that arose separately under in vitro drug pressure (C101F and L272F) incur both a fitness cost and a monstrously swollen DV[13].

Although these findings indicate that PfCRT is crucial for parasite growth and replication, and for maintaining the osmotic homoeostasis of the DV, its native substrates and function remain unresolved. Attempts to characterise PfCRT in heterologous expression systems have produced wildly disparate results, with it being claimed to function as either a Cl⁻ channel[28,29], a H⁺ pump[30], a non-specific cation channel[31], an activator of H⁺ pumps and of non-specific cation channels[31], a glutathione transporter[32], a non-specific transporter of inorganic and organic cations as well as of many other metabolites[33], or an $Fe^{2+}/Fe^{3+}$ transporter[34]. Moreover, none of these studies undertook experiments with parasites to substantiate that PfCRT exhibited the proposed transport activity in situ and that mediating this activity was the protein's physiological role.

Here, we use the *Xenopus* oocyte expression system in conjunction with measurements of solute transport, drug activity and metabolite levels in transgenic parasite lines to provide a detailed elucidation of the native substrate-specificity and physiological role of PfCRT. We show that PfCRT functions to export host-derived peptides 4–11 residues in length from the parasite's DV, and that the protein does not behave as a non-specific transporter of other metabolites and/or ions. The transport of peptides and peptide mimics via PfCRT is saturable, blocked by known PfCRT inhibitors, and is dependent on protons as well as on a co-substrate that remains to be identified. Relative to the wild-type protein, the drug-resistance-conferring isoforms of PfCRT recognise far fewer peptides and peptide mimics, and have markedly lower maximum rates of peptide transport. The reduced capacities of mutant PfCRT isoforms for transporting peptides out of the DV results in the accumulation of the substrate peptides in drug-resistant parasite lines. Together, our

findings provide a molecular basis for why PfCRT is essential for parasite survival and why drug-resistance-conferring mutations in the transporter impart a fitness cost. We present a mechanistic model in which the PfCRT-mediated transport of peptides from the DV to the cytosol serves to (1) provide a source of amino acids to support the parasite's high growth rate and (2) reduce peptide levels within the DV and thereby prevent the osmotic stress, swelling and dysfunction of this organelle.

## Results

**Cis-inhibition screens identify putative substrates of PfCRT.** A large collection of metabolites and ions was screened for molecules that interact with the substrate-binding cavity of PfCRT in the *Xenopus* oocyte system[15]. Our approach utilised two CQ-resistance-conferring isoforms of PfCRT (PfCRT$^{Ecu1110}$ and PfCRT$^{Dd2}$) which, unlike wild-type PfCRT (PfCRT$^{3D7}$), transport tritiated CQ ([³H]CQ) into the oocyte[16]. This property was exploited to detect unlabelled solutes that inhibit [³H]CQ uptake, consistent with them interacting with the mutant transporter's substrate-binding cavity (Fig. 1a).

None of the test solutes affected the accumulation of [³H]CQ in the negative controls, and most solutes also failed to *cis*-inhibit [³H]CQ transport via PfCRT$^{Ecu1110}$ or PfCRT$^{Dd2}$ (Fig. 1b–d; Supplementary Data 1, 2 and Supplementary Note 1). By contrast, almost all of the 95 peptides screened strongly *cis*-inhibited [³H]CQ transport by both PfCRT$^{Ecu1110}$ and PfCRT$^{Dd2}$. The test peptides were 2–25 residues in length and most occur in haemoglobin or other erythrocyte proteins. Only the dipeptides LH and GQ were without effect. Two haemoglobin-derived peptides, VDPVNF (VF-6) and HVDDM (HM-5), were selected for further characterisation and were confirmed to *cis*-inhibit [³H]CQ transport via PfCRT$^{Dd2}$ in a concentration-dependent manner (Fig. 1e).

**Peptides containing 4–11 residues trans-stimulate PfCRT.** The *cis*-inhibition assay identifies compounds that interact with PfCRT, but does not distinguish between solutes that inhibit (and are not translocated) and those that are substrates of PfCRT. We therefore employed a *trans*-stimulation assay to investigate whether the host-derived peptides are substrates of PfCRT (Fig. 2a). *Trans*-stimulation is a phenomenon exhibited by many carrier-type transporters (such as PfCRT[21,35]) whereby the reorientation of the binding site between the two faces of the membrane occurs more rapidly when it is occupied by a substrate than when it is empty (Supplementary Note 2). This property means that the PfCRT-mediated uptake of [³H]CQ from the extracellular solution will increase when an unlabelled substrate is present on the cytosolic face of the membrane, but will be unaffected by the presence of solutes that do not interact with the transporter[21].

None of the test solutes affected [³H]CQ accumulation in the negative controls, and LH was without effect in all oocyte types (Fig. 2b–g; Supplementary Fig. 1 and Supplementary Data 1, 2). Moreover, the solutes that had failed to *cis*-inhibit PfCRT also did not *trans*-stimulate [³H]CQ uptake, confirming that they are not substrates of the transporter (Fig. 2b, c, g; Supplementary Note 2, 3). By contrast, of the 93 peptides found to *cis*-inhibit PfCRT, 32 *trans*-stimulated [³H]CQ transport via PfCRT$^{Ecu1110}$ and 23 of these peptides also *trans*-stimulated PfCRT$^{Dd2}$. Somewhat unexpectedly, 39 peptides *trans*-stimulated [³H]CQ transport via PfCRT$^{3D7}$. This finding revealed that although CQ transport via PfCRT$^{3D7}$ cannot be detected under normal conditions in the oocyte system or in situ[36,37], the wild-type transporter can mediate a low level of CQ transport when *trans*-stimulated by high concentrations of a peptide substrate. The *trans*-stimulation of PfCRT$^{3D7}$ by VF-6

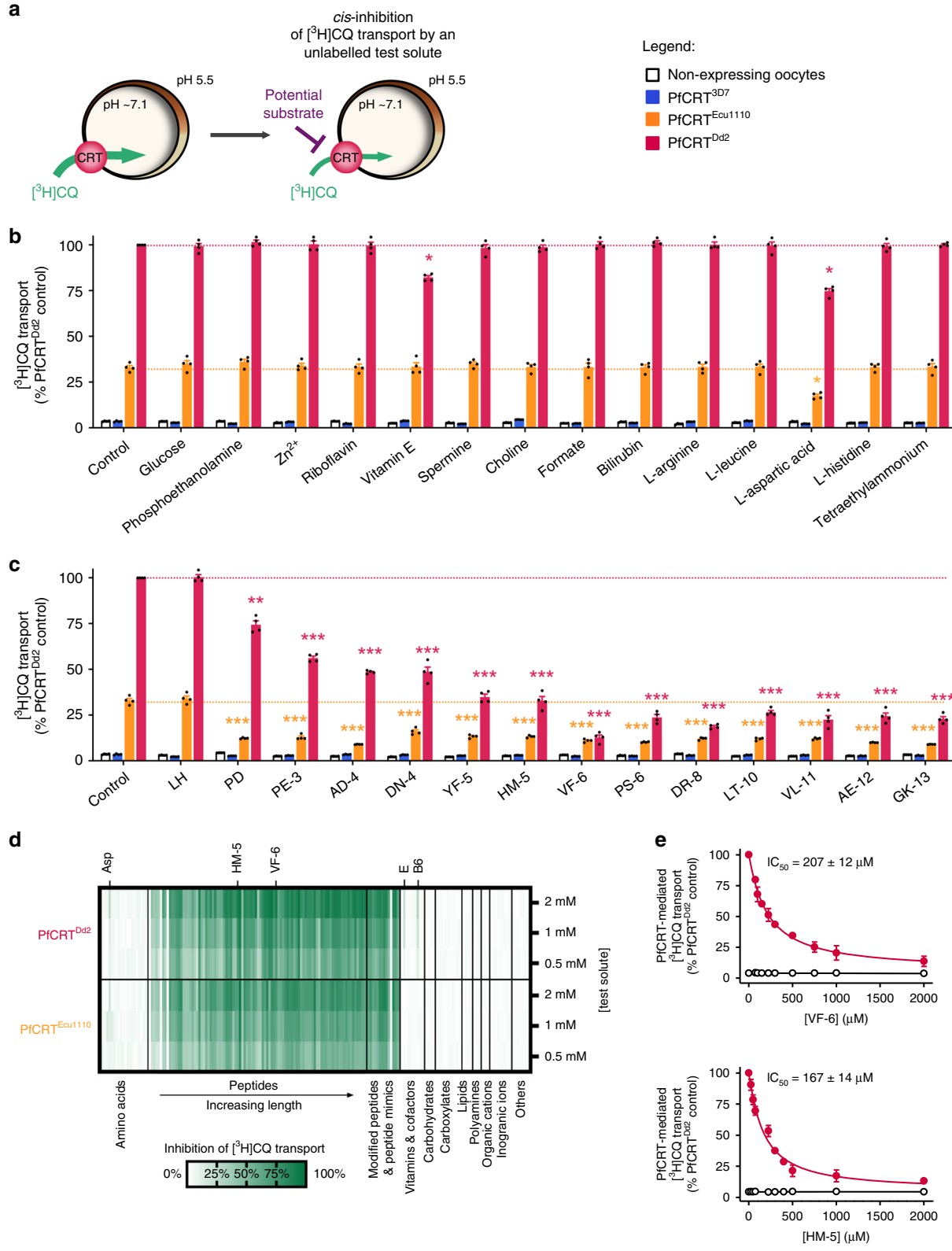

was pH-dependent, but was unaffected by the removal of Na$^+$ from the injection buffer (Fig. 2d, e).

The potent *trans*-stimulatory effects of VF-6 and HM-5 were concentration-dependent (Fig. 2f). In both cases, the Michaelis constant ($K_m$) values increased in the order PfCRT$^{3D7}$ < PfCRT$^{Ecu1110}$ < PfCRT$^{Dd2}$ and the maximal velocities ($V_{max}$) decreased in the order PfCRT$^{3D7}$ > PfCRT$^{Ecu1110}$ > PfCRT$^{Dd2}$.

That is, PfCRT$^{3D7}$ was *trans*-stimulated at lower peptide concentrations and to a greater extent than the mutant transporters.

The *trans*-stimulation of [$^3$H]CQ transport by VF-6 was not due to its breakdown into small fragments, as the injection of amino acids, dipeptides or tripeptides derived from VF-6 failed to *trans*-stimulate PfCRT. By contrast, all of the VF-6-derived

**Fig. 1 Screening of a solute library for *cis*-inhibitory activity identified potential substrates of PfCRT. a** Schematic showing the *cis*-inhibition of [3H]CQ uptake via PfCRT[Dd2] or PfCRT[Ecu1110] by an unlabelled solute in the *Xenopus* oocyte system. The library included solutes known to exist in the parasitised erythrocyte, such as vitamins, carbohydrates, metal ions, lipids, amino acids, peptides and other organic and inorganic molecules. **b, c** The effect on [3H]CQ transport via PfCRT by a subset of solutes (2 mM) (**b**) or host-derived peptides (2 mM) (**c**). Non-expressing oocytes and those expressing PfCRT[3D7] were included as negative controls; these oocytes accumulate a low level of [3H]CQ via simple diffusion of the neutral species[15,16], reflecting the background level of [3H]CQ transport. Note that aspartate and vitamins E and B6 caused modest reductions in PfCRT[Dd2]-mediated transport, but only aspartate was found to inhibit PfCRT[Ecu1110]. A diverse range of peptides cause potent inhibition of both PfCRT[Dd2] or PfCRT[Ecu1110]. **d** Heatmap of the *cis*-inhibition of [3H]CQ transport via PfCRT[Dd2] and PfCRT[Ecu1110] by 173 different solutes. **e** VF-6 (top) and HM-5 (bottom) inhibit [3H]CQ transport via PfCRT[Dd2] in a concentration-dependent manner. The half-maximum inhibitory concentrations (IC$_{50}$) for VF-6 and HM-5 are 207 ± 12 and 167 ± 14 μM, respectively. The data are the mean of $n = 4$ independent experiments (each yielding similar results and overlaid as individual data points in **b** and **c**), and the error is the SEM. Where not visible, the error bars fall within the symbols. The asterisks denote a significant difference from the relevant PfCRT[Dd2] (red asterisks) or PfCRT[Ecu1110] (orange asterisks) control; *$P < 0.05$, **$P < 0.01$, ***$P < 0.001$ (one-way ANOVA). The source datasets are provided as a Source Data file.

peptides containing ≥4 residues *trans*-stimulated [3H]CQ transport (Supplementary Fig. 1 and Supplementary Data 1, 2).

The antiretroviral drug saquinavir has previously been shown to interact with PfCRT[Dd2][38]. We therefore tested a number of peptide mimics and found that several—including saquinavir—*cis*-inhibited and *trans*-stimulated [3H]CQ transport via PfCRT[3D7], PfCRT[Ecu1110] and PfCRT[Dd2] (Fig. 2b, c; Supplementary Fig. 1 and Supplementary Data 1, 2).

Collectively, these findings indicate that PfCRT transports a broad range of peptides that contain 4–11 amino acid residues, are neutral, negatively or positively charged at pH 5.5 (with a bias towards a net negative charge), and do not have a particular sequence (Supplementary Fig. 1). Furthermore, PfCRT[3D7] accepts a broader range of peptides than either PfCRT[Ecu1110] or PfCRT[Dd2]; [3H]CQ transport via PfCRT[3D7] was *trans*-stimulated by 46 peptides and peptide mimics, which is 9 more than PfCRT[Ecu1110], and 20 more than PfCRT[Dd2].

**Characterisation of peptide transport via PfCRT.** We sought to obtain a direct demonstration of peptide transport via PfCRT by undertaking radioisotope transport assays with one of the potent *trans*-stimulators of all three PfCRT isoforms. A tritiated analogue of VF-6 was obtained and numerous unsuccessful attempts were made to measure its uptake from the extracellular solution into oocytes expressing PfCRT[3D7], PfCRT[Ecu1110] or PfCRT[Dd2]. However, when [3H]VF-6 was injected into the oocyte, all three PfCRT isoforms mediated its efflux from the oocyte, whereas [3H]VF-6 transport was not detected in the negative controls (Fig. 3a–e). This assay was therefore employed to characterise peptide transport via PfCRT. The efflux of [3H]VF-6 from PfCRT-expressing oocytes was linear over time for at least 2 h and was dependent on both the pH of the injection buffer and the oocyte membrane potential (see also Supplementary Fig. 2 and Supplementary Note 4). By contrast, removal of Na$^+$ or Cl$^-$ from the injection buffer, or varying the Ca$^{2+}$ concentration, was without effect (Supplementary Fig. 2). VF-6 transport via all three PfCRT isoforms was saturable, with PfCRT[3D7] having a slightly lower affinity, but much higher capacity, for VF-6 transport relative to the two mutant transporters.

Several drugs known to interact with the substrate-binding cavity of PfCRT were found to inhibit [3H]VF-6 transport via the three PfCRT isoforms (Fig. 3f–i; Supplementary Fig. 3). These were verapamil[15,39] and chlorpheniramine[40] (both partial reversers of CQ resistance in vitro), the quinine dimer 'Q$_2$C' (currently the most potent inhibitor of PfCRT[Dd2])[20], CQ[15] and saquinavir[38]. Q$_2$C was the most potent *cis*-inhibitor, with half-maximum inhibitory concentrations (IC$_{50}$) in the nanomolar range. With the exception of saquinavir, the drug IC$_{50}$s decreased in the order PfCRT[3D7] > PfCRT[Ecu1110] > PfCRT[Dd2], and similar IC$_{50}$s and trends were obtained when verapamil,

chlorpheniramine, CQ and Q$_2$C were tested for the ability to *trans*-inhibit [3H]VF-6 transport via PfCRT. By contrast, the saquinavir *cis*-inhibition IC$_{50}$s increased in the order PfCRT[3D7] < PfCRT[Ecu1110] < PfCRT[Dd2]. Moreover, saquinavir *trans*-stimulated [3H]VF-6 transport via all three PfCRT isoforms. Given this result, and the fact that [3H]CQ is readily taken up into the oocyte from the extracellular solution via PfCRT[Dd2] and PfCRT[Ecu1110], we explored whether high extracellular concentrations of CQ would *trans*-stimulate, rather than *trans*-inhibit, the efflux of [3H]VF-6. We observed modest *trans*-stimulation of [3H]VF-6 transport via all three PfCRT isoforms when [CQ] ≥ 500 μM.

These findings led us to test whether a subset of the host-peptides that *trans*-stimulated CQ transport via PfCRT were likewise able to *trans*-stimulate [3H]VF-6 transport. However, all of the test peptides (including unlabelled VF-6) *trans*-inhibited [3H]VF-6 transport via PfCRT, whereas the compounds included as negative controls (LH and free amino acids) were without effect (Fig. 3j). This result, together with the inability of PfCRT to take up [3H]VF-6 from the external solution, indicated that a co-substrate is required for peptide transport via PfCRT. All three solutes—peptide, proton and the unidentified co-substrate—must be present on the same side of the membrane for the PfCRT-mediated transport of host-derived peptides to occur. The requirement for a co-substrate was unexpected given that the drugs transported by PfCRT—e.g. CQ, quinine, quinidine, quinacrine, methylene blue and saquinavir—only require protons to undergo translocation[15,21,39,41] (see also Supplementary Note 4). However, the decoupling of some substrates, but not others, from the translocation of a co-substrate is a known phenomenon in carrier-type transporters[42–44]. The co-substrate is evidently present in the oocyte cytosol, but absent from the extracellular buffer. We screened many solutes, including Fe$^{2+}$, Fe$^{3+}$ and glutathione, but were unable to identify the co-substrate (Supplementary Data 1). Moreover, re-screening of a range of radiolabelled metabolites as potential substrates via injection into the oocyte failed to detect transport via PfCRT (Fig. 3k), nor were any of these radiolabelled solutes taken up via PfCRT when added to the external solution (Supplementary Fig. 3). Yet within the same assays, [3H]CQ and [3H]hypoxanthine were transported (both out of and into the oocyte) via mutant PfCRT and PfNT1, respectively (Fig. 3k; Supplementary Fig. 3i).

**Diverse isoforms of PfCRT transport the VF-6 peptide.** A broad range of PfCRT isoforms were expressed in the oocyte system (Supplementary Fig. 4). All of the field PfCRT isoforms transported VF-6, albeit to varying degrees (Fig. 4a; the positions of the mutations in PfCRT are shown in Supplementary Fig. 5). The kinetics of VF-6 transport via several of these proteins were indistinguishable from those of PfCRT[3D7] (e.g., the 2300, 106/1,

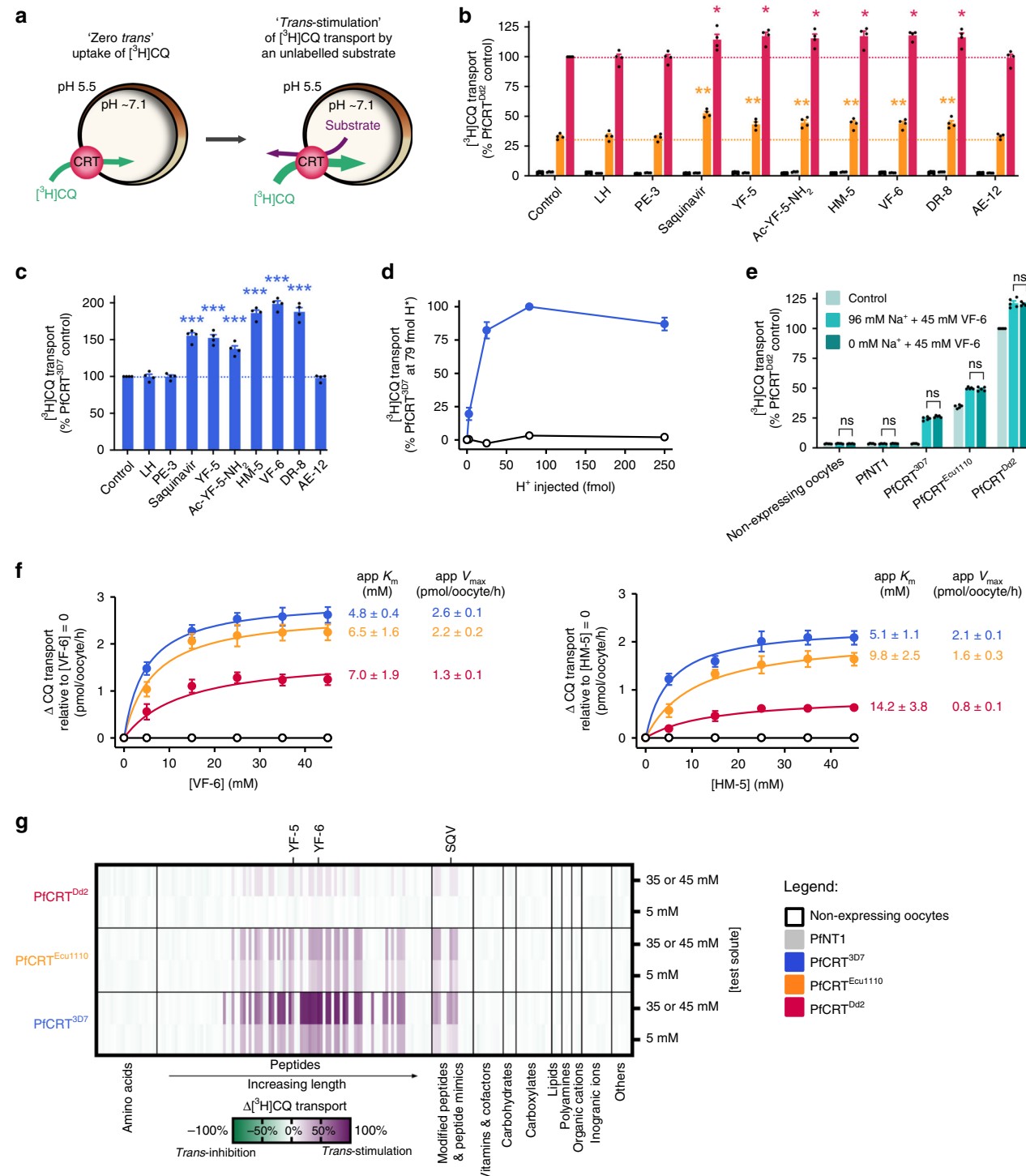

BC22 and Cam738 isoforms), whereas almost all of the CQ-resistance-conferring isoforms displayed significantly lower capacities for VF-6 transport (e.g., the China e, GB4, 783 and K1 isoforms). Moreover, field isoforms that arose from the introduction of a positively charged residue into a CQ-resistance-conferring variant of PfCRT showed a marked increase in VF-6 transport activity relative to their respective parent isoforms, but had lost the ability to transport CQ (e.g., cf. PfCRT[K1] with PfCRT[106/1] and PfCRT[2300]; Fig. 4a, Supplementary Fig. 6). These findings confirm the importance of electrostatic charges at key positions within PfCRT in determining the transporter's substrate-specificity[16,21]. They also provide a mechanistic

explanation for why the re-introduction of a positively charged residue into PfCRT's binding cavity re-sensitises the parasite to CQ while simultaneously restoring its innate fitness[23,45].

The relationship between the CQ and VF-6 transport activities of field PfCRT isoforms was examined to gain further insights into the correlation between mutations in PfCRT and reduced parasite fitness. The VF-6 transport activity of PfCRT decayed exponentially as its capacity to transport CQ increased (Fig. 4b; Supplementary Fig. 6). That is, a given reduction in peptide transport activity imparts a larger increase in CQ transport activity. A significant outlier to this relationship was PfCRT[Cam734]; its capacity for VF-6 transport exceeded that of

**Fig. 2 PfCRT is *trans*-stimulated by peptides containing 4–11 amino acid residues. a** Schematic showing the *trans*-stimulation of [$^3$H]CQ transport via PfCRT by an unlabelled substrate in *Xenopus* oocytes. **b, c** *Trans*-stimulation of [$^3$H]CQ transport via PfCRT$^{Dd2}$ or PfCRT$^{Ecu1110}$ (**b**) or via PfCRT$^{3D7}$ (**c**) by a subset of host-derived peptides and peptide mimics (45 mM). The test solute was injected into the oocyte immediately prior to the commencement of the assay, with the buffer-only and LH treatments serving as injection controls[21]. The negative controls were non-expressing oocytes and those expressing an unrelated *P. falciparum* transporter, the nucleoside transporter 1 (PfNT1[75,76]). The latter demonstrates that the expression of a similarly sized transporter in *Xenopus* oocytes does not affect the ability of the oocyte membrane to reseal following injection of a test solute[21]. **d** *Trans*-stimulation of [$^3$H]CQ transport via PfCRT$^{3D7}$ by VF-6 (45 mM) is pH-dependent, with the largest increase observed when the pH of the injection buffer was 5.5 (equating to 79 fmoles H$^+$). **e** *Trans*-stimulation of [$^3$H]CQ transport via PfCRT is unaffected by the removal of Na$^+$ from the injection buffer. **f** Concentration-dependence of the *trans*-stimulation of [$^3$H]CQ transport via PfCRT by VF-6 (left) and HM-5 (right). The non-expressing oocyte data overlays the data obtained with oocytes expressing PfNT1. **g** Heatmap of the *trans*-stimulation of [$^3$H]CQ transport via PfCRT$^{3D7}$, PfCRT$^{Ecu1110}$ or PfCRT$^{Dd2}$ by 173 different solutes. Only peptides and peptide mimics *trans*-stimulate PfCRT; all other solutes, including aspartate and vitamins E and B6, failed to increase [$^3$H]CQ transport via PfCRT are thus unlikely to be substrates of the transporter. The data are the mean of $n = 4$ independent experiments (each yielding similar results and overlaid as individual data points in **b**, **c** and **e**), and the error is the SEM. Where not visible, the error bars fall within the symbols. The asterisks denote a significant difference from the relevant PfCRT$^{Dd2}$ (red asterisks), PfCRT$^{Ecu1110}$ (orange asterisks) or PfCRT$^{3D7}$ (blue asterisks) buffer-injected control; *$P < 0.05$, **$P < 0.01$, ***$P < 0.001$, ns: not significant (one-way ANOVA). SQV: saquinavir. The source datasets are provided as a Source Data file.

PfCRT$^{3D7}$, but it also possessed a moderate level of CQ transport activity. This finding is consistent with the observation that parasites expressing PfCRT$^{Cam734}$ do not display a fitness disadvantage relative to wild-type parasites, and yet are moderately resistant to CQ[46].

Three laboratory-derived isoforms that cause a gross enlargement of the DV[13]—L272F-PfCRT$^{3D7}$, L272F-PfCRT$^{Dd2}$ and C101F-PfCRT$^{Dd2}$—displayed unusually low capacities for VF-6 transport (Fig. 4a). This observation provides a molecular basis for understanding why PfCRT is critical for maintaining the normal osmotic pressure and morphology of the DV.

We found that the fusion of polypeptides to the N- or C-termini of PfCRT interfere with its natural function. Although only one of the modified PfCRT$^{Dd2}$ proteins no longer mediates [$^3$H]CQ transport (Fig. 4c, left panel), all of the tagged versions of PfCRT$^{3D7}$ and PfCRT$^{Dd2}$ have lost the ability to transport [$^3$H] VF-6 (Fig. 4c, right panel). This observation helps to explain why previous studies, which expressed versions of PfCRT with N- and/or C-terminal fusions, did not delineate the native substrates of PfCRT (Supplementary Note 5).

**Peptide mimics are substrates of PfCRT in situ.** We sought to verify that PfCRT also functions as a peptide transporter within its native environment of the DV membrane. PfCRT activity was measured using the in situ 'H$^+$-efflux assay' (Fig. 5a) with a set of *P. falciparum* transfectants that are isogenic except for their *pfcrt* allele. These lines, known as C2$^{GC03}$, C6$^{7G8}$ and C4$^{Dd2}$, express either PfCRT$^{3D7}$, PfCRT$^{7G8}$ (which harbours the mutations present in PfCRT$^{Ecu1110}$ plus C72S) or PfCRT$^{Dd2}$, respectively[47]. The weak-base solute under study enters the parasite's acidic DV via simple diffusion of the neutral species, and/or via transporters such as the multidrug resistance protein 1 (PfMDR1[48,49]), and becomes protonated (Fig. 5a). Peptide mimics and modified host-peptides (Supplementary Fig. 6) were used in these assays due to their increased membrane permeability, and resistance to proteolytic degradation, relative to natural peptides.

Histidine, ritonavir and lopinavir were without effect in all of the lines, indicating that either they are not substrates of PfCRT, or that insufficient levels of these solutes accumulate within, and are effluxed from, the DV to induce a detectable H$^+$ leak. By contrast, several peptides and peptide mimics induced a H$^+$ leak in one or more of the parasite lines, consistent with them being substrates of PfCRT. Saquinavir and two modified host-peptides (Ac-YF-5-NH$_2$ and Ac-ST-7-NH$_2$; Supplementary Fig. 7) increased the rate of DV alkalinisation in all three lines, with the effect decreasing in the order C2$^{GC03}$ > C6$^{7G8}$ > C4$^{Dd2}$. Moreover, leupeptin and E-64 both caused a H$^+$ leak in C2$^{GC03}$ parasites, and leupeptin also induced a H$^+$ leak in the

C6$^{7G8}$ line, but neither peptide exerted an effect in C4$^{Dd2}$. That is, as was observed in the oocyte system, the range of peptides transported decreases in the order PfCRT$^{3D7}$ > PfCRT$^{7G8}$ > PfCRT$^{Dd2}$.

Verapamil and Q$_2$C inhibited the peptide-induced H$^+$ leaks in C2$^{GC03}$, C6$^{7G8}$ and C4$^{Dd2}$ parasites (Fig. 5b), providing further evidence that the peptide mimics and modified peptides are exported from the DV via PfCRT. Moreover, as observed in the oocyte system, Q$_2$C was the most potent inhibitor, and both Q$_2$C and verapamil were more effective against the mutant transporters than against PfCRT$^{3D7}$.

Kinetic analyses revealed that the peptides increase the rate of DV alkalinisation in a concentration-dependent manner (Fig. 5c; Supplementary Fig. 8), with the capacity for transport decreasing in the order PfCRT$^{3D7}$ > PfCRT$^{7G8}$ > PfCRT$^{Dd2}$. The same trend was observed in the oocyte system when we measured the concentration-dependence of the *trans*-stimulation of [$^3$H]CQ transport via PfCRT by these peptides (Fig. 5d; Supplementary Fig. 8), again confirming the strong consensus in the transport properties of PfCRT between the two experimental systems.

The finding that the mutant transporters have reduced capacities for exporting saquinavir suggested that this peptide mimic may be better retained within the DV of CQ-resistant parasites and could thereby affect their sensitivity to this protease inhibitor (which is suspected to exert its weak antimalarial activity inside the DV). We therefore determined the IC$_{50}$ of saquinavir against the three isogenic lines as well as to CQ-sensitive (3D7) and CQ-resistant (7G8 and Dd2) strains (Fig. 6, see Supplementary Fig. 8 for parasite susceptibilities to other peptide mimics). As expected, the parasites expressing mutant PfCRT isoforms were more susceptible to saquinavir than those expressing PfCRT$^{3D7}$. Moreover, verapamil was shown to increase parasite sensitivity to saquinavir.

Together, these datasets provide a robust demonstration of peptide transport via PfCRT in situ. They also indicated that the reduced capacity of the mutant transporters for exporting host-derived peptides would cause these catabolites to accumulate within the DV.

**Parasites with mutant isoforms of PfCRT accumulate peptides.** Two previous metabolomic analyses have reported that CQ-resistant parasites accumulate significantly more aspartate and peptides than their CQ-sensitive counterparts[27,50]. However, it was unclear how to reconcile these metabolite profiles with (1) the inability of PfCRT to translocate amino acids and small peptides and (2) our identification of host-peptide substrates of PfCRT that were not detected, or were reported as non-significant, in these studies. We therefore undertook quantitative peptidomic

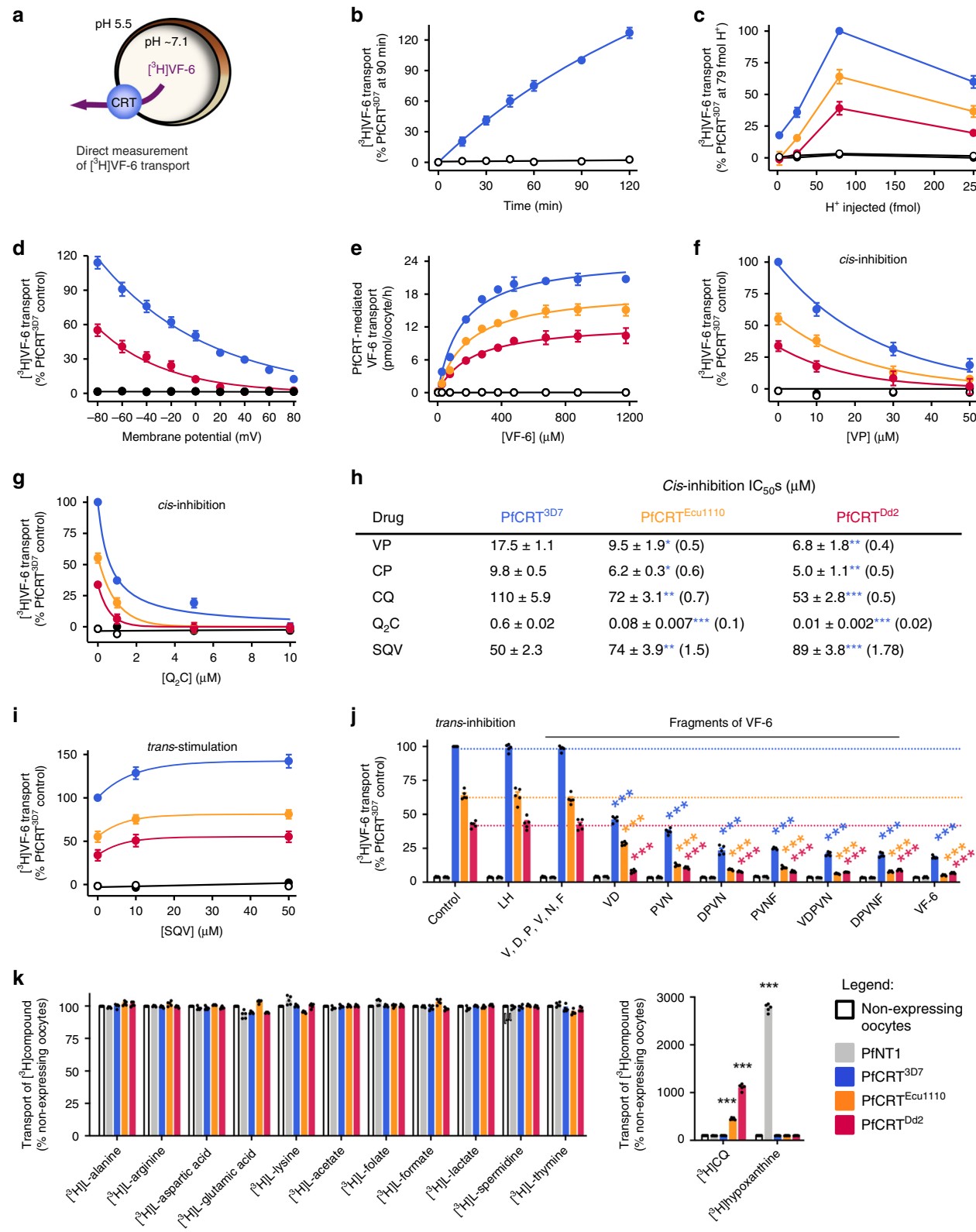

analyses to examine the populations of host-derived peptides present in the C2^GC03, C6^7G8 and C4^Dd2 parasites (Supplementary Data 3–5). Peptides of varying length, composition and charge were detected, with most originating from haemoglobin and the remainder from other erythrocytic proteins. Several of these peptides accumulated to high levels within the CQ-resistant parasites relative to the C2^GC03 line (Supplementary Data 3–5),

including VF-6 and other peptides we had designated from the trans-stimulation assays as likely substrates of PfCRT.

The apparent agreement between the trans-stimulation and peptidomic datasets for the initial set of peptides studied in the oocyte system led us to test the potential of the accumulation profiles to predict whether a given peptide was a substrate of PfCRT. These predictions also considered the length of the

**Fig. 3 Characterisation of the transport of a host-derived hexapeptide via PfCRT. a** Schematic showing the transport of [³H]VF-6 via PfCRT in *Xenopus* oocytes. **b–e** [³H]VF-6 transport via PfCRT is approximately linear with time for at least 2 h (**b**), is dependent on both the pH of the injection buffer (the highest level of transport occurred at pH 5.5, equating to 79 fmoles H⁺) (**c**) and the membrane potential (with the rate of VF-6 efflux steadily decreasing as the membrane potential became more positive) (**d**), and is saturable (PfCRT³D⁷ possesses a slightly lower affinity and higher $V_{max}$ relative to the mutant transporters) (**e**). [³H]CQ influx is also dependent on the membrane potential (Supplementary Fig. 2g and Supplementary Note 3). **f, g** *Cis*-inhibition of [³H]VF-6 transport via PfCRT by verapamil (VP) (**f**) and $Q_2C$ (**g**). **h** The half-maximum inhibitory concentrations (IC₅₀s) for the *cis*-inhibition of [³H]VF-6 transport via PfCRT by drugs known to interact with the transporter. **i** *Trans*-stimulation of [³H]VF-6 transport via PfCRT by saquinavir (SQV). **j** *Trans*-inhibition of [³H]VF-6 transport via PfCRT by haemoglobin-peptides (45 mM). **k** Left: various other radiolabelled solutes are not transported out of the oocyte by PfCRT. Right: in the same assay, [³H]CQ is effluxed via PfCRT^Ecu1110 and PfCRT^Dd2 and [³H]hypoxanthine is effluxed by PfNT1. The data are the mean of $n = 5$ independent experiments (each yielding similar results and overlaid as individual data points in **j** and **k**), and the error is the SEM. Where not visible, the error bars fall within the symbols. The non-expressing oocyte data overlays the data obtained with oocytes expressing PfNT1 in **b**, **c**, **e–g** and **i**. The asterisks denote a significant difference from the relevant PfCRT^Dd2 (red asterisks), PfCRT^Ecu1110 (orange asterisks), PfCRT³D⁷ (blue asterisks) or non-expressing (black asterisks) control; *$P < 0.05$, **$P < 0.01$, ***$P < 0.001$, ns not significant (one-way ANOVA). The source datasets are provided as a Source Data file.

peptides (i.e. 4–11 residues) transported by PfCRT. Of the 41 peptides selected for a second round of testing, 22 were predicted to be substrates, 14 were predicted not to be substrates and 5 could not be confidently assigned a prediction from the peptidomic data. All 22 of the predicted substrates *trans*-stimulated [³H]CQ transport via PfCRT and all 14 of the predicted non-substrates were without effect. Of the 5 unassigned peptides, one was found to be a PfCRT substrate and the other four failed to *trans*-stimulate the transporter. We also showed that several of the non-haemoglobin erythrocytic peptides detected in the peptidomic analyses are substrates of PfCRT (Supplementary Data 1 and 2).

Overall, there is a strong positive relationship between the accumulation within CQ-resistant parasites of peptides 4–11 residues in length and their capacity to *trans*-stimulate PfCRT (Fig. 7). Two types of exceptions to this correlation are outlined in Supplementary Note 6. These caveats aside, our findings indicate that the peptidomic datasets can be employed to infer which host-derived peptides are likely to be substrates of PfCRT.

The CQ-resistant lines tended to contain lower levels of the large host-derived oligopeptides (>15 residues) relative to the shorter fragments within a given degradation cascade. This is consistent with the digestive process (and thus parasite growth) being slowed down in CQ-resistant parasites in response to an over-accumulation of PfCRT's peptide substrates within the DV.

## Discussion

Our work has provided a definitive resolution of the natural substrates and physiological role of PfCRT, a key contributor to multidrug resistance in the malaria parasite. Contrary to previous reports, PfCRT is not a non-specific transporter of metabolites and/or of organic or inorganic ions. Instead, the datasets we have obtained using a set of complementary in vitro and in situ assays reveal that PfCRT exports host-derived peptides containing 4–11 residues out of the parasite's DV. Peptide transport via PfCRT shows a slight bias towards negatively charged peptides, is saturable at micromolar concentrations, and is blocked by known PfCRT inhibitors, including verapamil, chlorpheniramine and $Q_2C$. Moreover, the transport of peptides is dependent on protons as well as on a second solute that remains to be identified, but which is naturally present in the *Xenopus* oocyte. We found that most resistance-conferring isoforms of PfCRT, which efflux CQ from the DV but also typically impart a fitness cost to the parasite, exhibit reduced rates of peptide transport. Furthermore, relative to wild-type PfCRT, the mutant transporters accept a narrower range of peptide substrates.

The significant reductions in the capacities of the mutant proteins to efflux peptides from the DV account for the elevated levels of peptides we measured within CQ-resistant parasites.

Moreover, the relative capacities of different PfCRT isoforms to mediate peptide transport can be readily reconciled with the magnitude of the phenotypes displayed by the corresponding parasites. For example, peptide transport activity decreases in the order PfCRT³D⁷ > PfCRT⁷G⁸ > PfCRT^Dd2 and this mirrors the degree to which peptides of 4–11 residues accumulate within the respective parasite lines (C2^GC03 < C6^7G8 < C4^Dd2) as well as the extent of the fitness costs imparted by the mutant transporters[51]. Likewise, a poor capacity for peptide transport correlates with increased osmotic stress (i.e. swollen DVs). That is, the L272F and C101F variants of PfCRT have very low peptide transport activities and cause grossly enlarged DVs, and parasites carrying PfCRT^Dd2, which has one of the poorest capacities for peptide transport amongst the field isoforms of the protein, display moderately swollen DVs.

Our quantitative peptidomic analyses of the C2^GC03, C6^7G8 and C4^Dd2 lines, together with previous metabolomic studies of these parasites[27,50], point to further metabolic manifestations of the reduced capabilities of mutant isoforms for peptide transport. First, the peptide substrates accumulating within the DV of CQ-resistant parasites appear to be promiscuously degraded into smaller peptides and amino acids. The generation of these catabolites would add significantly to the osmotic stress of the DV, and this could explain why polymorphisms in a putative amino acid transporter (PfAAT1), a resident of the DV membrane, have been linked to resistance to CQ and other compounds[14,52]. The changes to PfAAT1 likely increase its capacity for exporting amino acids and/or small peptides from the DV. Secondly, the build-up of peptides and amino acids within the DV of CQ-resistant parasites appear to exert feedback inhibition upon the degradation of host polypeptides, resulting in an accumulation of large oligopeptides relative to the CQ-sensitive line. All of the above perturbations to DV morphology and metabolism, along with our finding that the PfCRT isoforms responsible have significantly reduced capacities for exporting host-derived peptides from this organelle, provide the long-awaited molecular explanations for how CQ-resistance-conferring mutations in PfCRT decrease parasite fitness and why the transporter is essential for parasite survival. Together, our findings indicate that peptide transport via PfCRT serves two roles in the physiology of the intraerythrocytic parasite (Fig. 8). First, the delivery of host-derived peptides to the cytosol provides a source of amino acids to support the parasite's high rates of growth and multiplication. Secondly, the concomitant reduction in peptide levels within the DV prevents the osmotic stress, swelling and dysfunction of this organelle.

The ongoing emergence and spread of new PfCRT isoforms (e.g. in piperaquine-resistant isolates in Cambodia[53,54]) indicates that the transporter is still evolving in response to drug pressure.

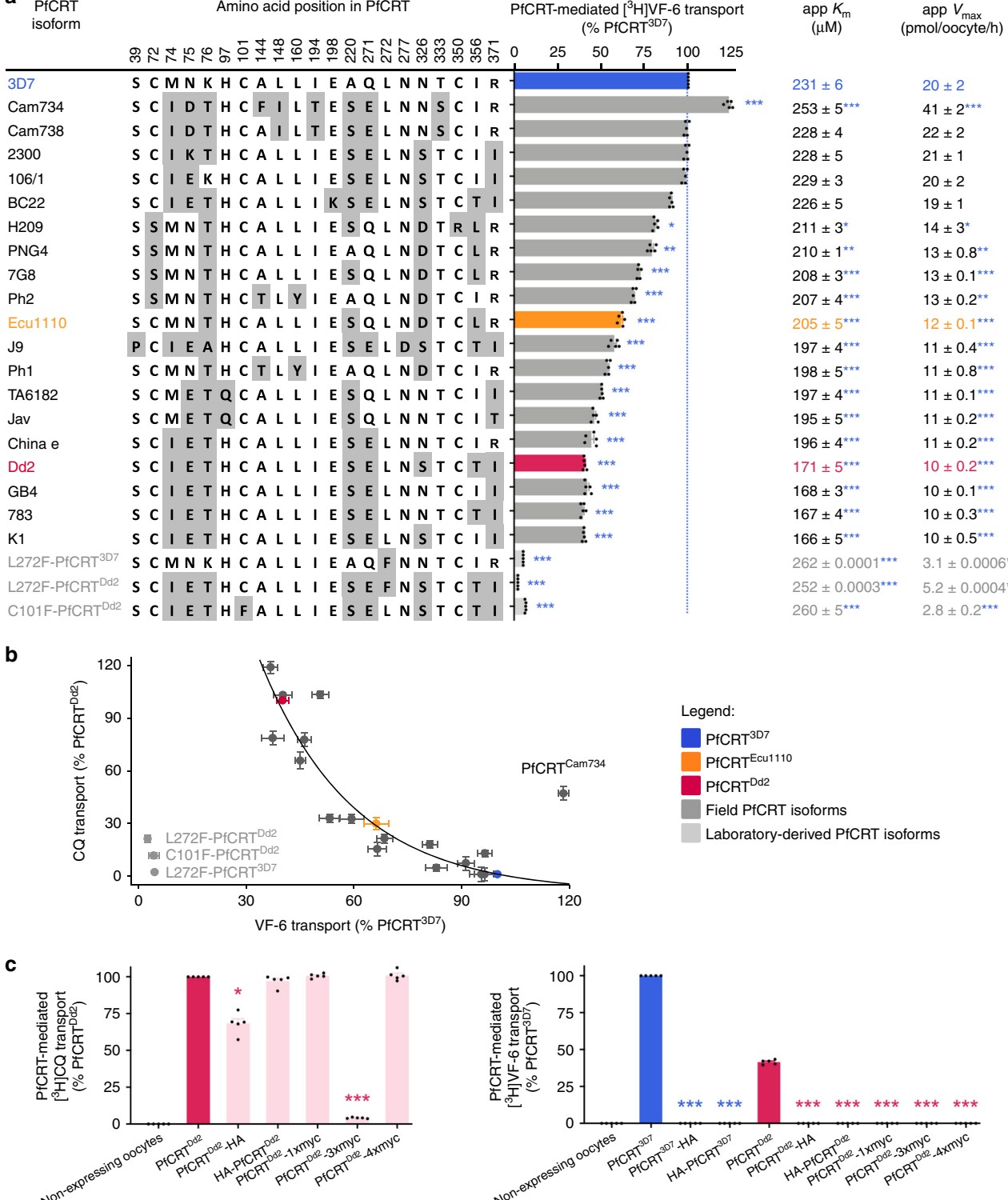

**Fig. 4 Diverse field and laboratory-derived isoforms of PfCRT transport the VF-6 peptide. a** The VF-6 transport activities, including kinetic parameters, of various field and laboratory-derived PfCRT isoforms in *Xenopus* oocytes ($K_m$, Michaelis–Menten constant; $V_{max}$, maximum velocity). **b** The capacity of PfCRT to transport VF-6 decays exponentially as the protein's CQ transport activity increases ($R^2 = 0.93$). Exceptions to this trend include PfCRT$^{Cam734}$, L272F-PfCRT$^{Dd2}$, L272F-PfCRT$^{3D7}$ and C101F-PfCRT$^{Dd2}$. **c** [$^3$H]CQ transport (left) and [$^3$H]VF-6 transport (right) via epitope-tagged versions of PfCRT$^{3D7}$ and PfCRT$^{Dd2}$. The version of PfCRT$^{Dd2}$ carrying a C-terminal 3xmyc tag does not mediate [$^3$H]CQ transport. The other four variants of PfCRT$^{Dd2}$ retain all or most of their CQ transport activity. By contrast, none of the tagged versions of PfCRT$^{3D7}$ or PfCRT$^{Dd2}$ transport [$^3$H]VF-6. The fusion of polypeptides to PfCRT can therefore abolish its ability to transport peptides, even when the protein remains able to transport CQ. The data are the mean of $n = 5$ independent experiments (each yielding similar results and overlaid as individual data points in **a** and **c**), and the error is the SEM. Where not visible, the error bars fall within the symbols. The asterisks denote a significant difference from the relevant PfCRT$^{Dd2}$ (red asterisks) or PfCRT$^{3D7}$ (blue asterisks) control; *$P < 0.05$, **$P < 0.01$, ***$P < 0.001$ (one-way ANOVA). The source datasets are provided as a Source Data file.

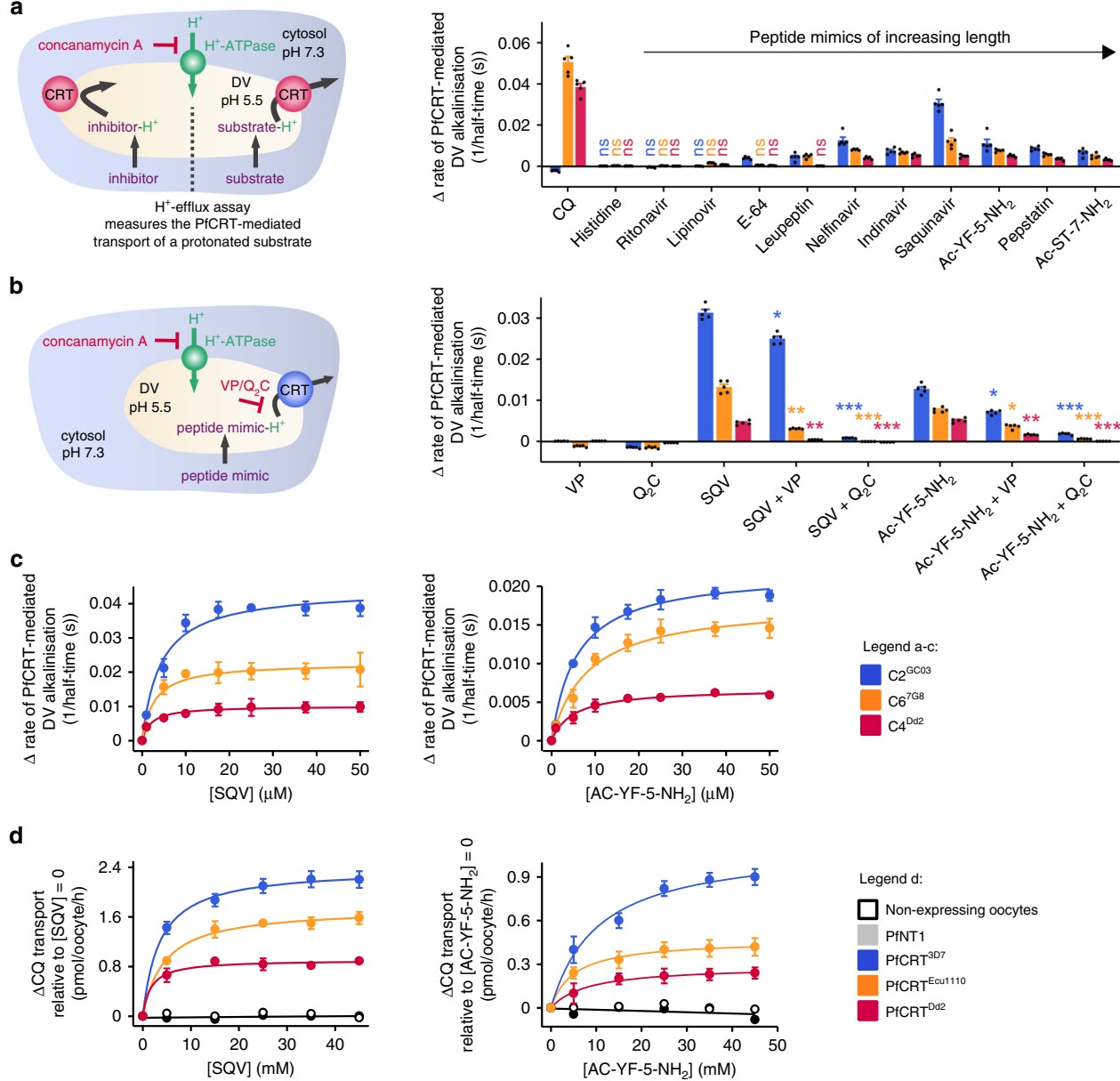

**Fig. 5 Peptide mimics are substrates of PfCRT in situ. a** Left: schematic showing the H$^+$-efflux assay. If a protonated solute is a substrate of PfCRT it will cause a H$^+$ leak when effluxed from the DV. Inhibition of the DV's H$^+$-ATPase by concanamycin A enables this H$^+$ leak to be detected (with a pH-sensitive probe) as an increase in the rate of DV alkalinisation. Right: several peptide mimics (10 μM) increase the rate of DV alkalinisation in C2$^{GC03}$, C6$^{7G8}$ and C4$^{Dd2}$ parasites. CQ (the positive control) causes the rate of DV alkalinisation to increase in the C6$^{7G8}$ and C4$^{Dd2}$ lines and to decrease slightly in C2$^{GC03}$ parasites. This is consistent with previous applications of the H$^+$-efflux assay[36,37] and with the abilities of PfCRT$^{Ecu1110}$, PfCRT$^{7G8}$ and PfCRT$^{Dd2}$ (but not PfCRT$^{3D7}$) to transport CQ when tested under normal conditions in the oocyte system[15,16]. Unless labelled ns, $P < 0.05$ relative to the absence of a test solute. **b** Left: schematic showing the inhibition of PfCRT by verapamil (VP) or the quinine dimer Q$_2$C in the H$^+$-efflux assay. Right: the increase in the rate of DV alkalinisation caused by saquinavir (SQV; 10 μM) and Ac-YF-5-NH$_2$ (10 μM) is inhibited by VP (50 μM) and Q$_2$C (1 μM). The statistical analyses were performed relative to the SQV or Ac-YF-5-NH$_2$ controls. **c** SQV (left) and Ac-YF-5-NH$_2$ (right) increase the rate of DV alkalinisation in a concentration-dependent manner. **d** Concentration-dependence of the *trans*-stimulation of [$^3$H]CQ transport via PfCRT by SQV (left) and Ac-YF-5-NH$_2$ (right) in *Xenopus* oocytes. The non-expressing oocyte data overlays the data obtained with oocytes expressing PfNT1. The data are the mean of $n = 5$ independent experiments (each yielding similar results and overlaid as individual data points in **a** and **b**), and the error is the SEM. Where not visible, the error bars fall within the symbols. The asterisks denote a significant difference from the relevant C4$^{Dd2}$ (red asterisks), C6$^{7G8}$ (orange asterisks) or C2$^{GC03}$ (blue asterisks) control; *$P < 0.05$, **$P < 0.01$, ***$P < 0.001$, ns: non-significant (one-way ANOVA). The source datasets are provided as a Source Data file.

A point of concern in this regard is PfCRT$^{Cam734}$, as its peptide transport capability is superior to that of the wild-type protein, yet it has retained the ability to transport CQ. We now have the means to evaluate both the peptide and drug transport functions of these new isoforms and thereby gain significant insights into the causes and constraints that are dictating the

evolution of PfCRT in different malarious regions. Moreover, the knowledge we have gained of PfCRT's native substrates, of its ability to transport—and affect the antiplasmodial activity of—peptide mimics, as well as of the inhibition of its natural function by existing drugs, provides a foundation for the development of strategies that combat malaria parasites by targeting

**a**

| Parasite line/strain | CQ IC$_{50}$ (nM) | | SQV IC$_{50}$ (μM) | |
|---|---|---|---|---|
| | Control | + 1 μM VP | Control | + 1 μM VP |
| C2$^{GC03}$ | 27 ± 1.6 | 26 ± 1.3 | 6.8 ± 0.4 | 5.1 ± 0.2** |
| C6$^{7G8}$ | 87 ± 2.1 | 25 ± 0.2*** | 4.2 ± 0.4 | 2.3 ± 0.3** |
| C4$^{Dd2}$ | 149 ± 4.6 | 26 ± 0.1*** | 3.5 ± 0.5 | 1.9 ± 0.1** |
| 3D7 | 25 ± 1.1 | 24 ± 0.9 | 7.4 ± 0.3 | 5.9 ± 0.6** |
| 7G8 | 94 ± 4.4 | 28 ± 1.6*** | 4.7 ± 0.1 | 2.2 ± 0.1** |
| Dd2 | 151 ± 4.8 | 25 ± 1.0*** | 3.3 ± 0.2 | 2.1 ± 0.3** |

**b**

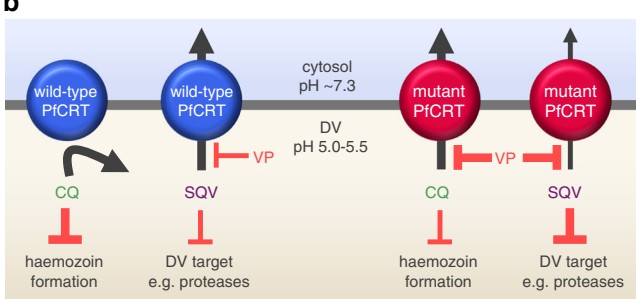

**Fig. 6 CQ-resistance-conferring isoforms of PfCRT increase parasite susceptibility to saquinavir. a** The antiplasmodial activities of CQ and saquinavir (SQV) against CQ-sensitive (C2$^{GC03}$ and 3D7) and CQ-resistant (C6$^{7G8}$, C4$^{Dd2}$, 7G8 and Dd2) parasites in the presence or absence of verapamil (VP). The data are the mean of $n = 5$ independent experiments (each yielding similar results) and the error is the SEM. The asterisks denote a significant difference from the relevant control (red asterisks, C4$^{Dd2}$ or Dd2 control; orange asterisks, C6$^{7G8}$ or 7G8 control; blue asterisks, C2$^{GC03}$ or 3D7 control); **$P < 0.01$, ***$P < 0.001$, ns: non-significant (one-way ANOVA). The source datasets are provided as a Source Data file. **b** Mechanistic model for the increased susceptibility of CQ-resistant parasites to SQV. PfCRT$^{3D7}$ (wild-type PfCRT) lacks detectable CQ transport activity, whereas the CQ-resistance-conferring isoforms of PfCRT (mutant PfCRT) efflux CQ from the DV. VP inhibits CQ transport via the mutant transporters, thereby re-sensitising the CQ-resistant parasites to this antimalarial drug. By contrast, PfCRT$^{3D7}$ has a greater capacity for transporting SQV out of the DV than do the mutant isoforms. Moreover, it is the wild-type protein that confers reduced sensitivity to SQV. Given that VP (1) inhibits SQV transport via both the wild-type and mutant PfCRT transporters and (2) increases the antiplasmodial activity of SQV in all of the parasite types, our findings indicate that SQV acts on a target within the DV.

both the normal and drug-resistance-conferring functions of PfCRT.

## Methods

***Xenopus laevis* frogs**. Ethical approval of the work performed with female *X. laevis* frogs was obtained from the Australian National University (ANU) Animal Experimentation Ethics Committee (Animal Ethics Protocol Numbers R.BSB.01.10 and A2013/13) in accordance with the Australian Code of Practice for the Care and Use of Animals for Scientific Purposes. The frogs were purchased from Nasco (Cat# LM00535M) and were housed in the *Xenopus* Frog Facility of the ANU Research School of Biology in compliance with the relevant institutional and Australian Government regulations.

**Culture of parasitised erythrocytes**. The use of human blood for this work was approved by the ANU Human Research Ethics Committee (Human Ethics Approval Numbers 2011/266 and 2017/351). The asexual intraerythrocytic stages of three *P. falciparum* strains and three *P. falciparum pfcrt* transfectant lines were used in this study. The strains were wild-type '3D7' parasites (isolated from the Netherlands but probably of African origin[55]) and the CQ-resistant parasites 'Dd2' (isolated from Southeast Asia) and '7G8' (isolated from Brazil). The three *pfcrt* transfectant lines, generated by Sidhu et al.[47], were C2$^{GC03}$ (a recombinant control that retains the wild-type *pfcrt* allele, PfCRT$^{3D7}$) and the CQ-resistant lines C6$^{7G8}$ and C4$^{Dd2}$ (in which the wild-type *pfcrt* allele of the 'GC03' line has been replaced with the allele from the 7G8 or Dd2 strain, respectively[47]). A PCR error in the

generation of the C6$^{7G8}$ line introduced an additional mutation (I351M) which is not normally found in the *pfcrt* allele of 7G8 parasites[56]. The parasite cultures were maintained at 4% haematocrit and synchronised with 5% (w/v) sorbitol[57]. The *pfcrt* transfectant lines were maintained in the presence of the selection agents blasticidin (5 μM; Sigma-Aldrich) and WR99210 (5 nM; Jacobus Pharmaceuticals). These selection agents were not present during the experiments.

Two cell line authentication methods were used to verify the identities of the six strains and lines used in this study. The first evaluated the CQ-resistance phenotype of each parasite culture by performing cell proliferation assays, thereby yielding CQ IC$_{50}$s for each parasite type. The results were verified by comparing them to previously published data[21,36,37] (see the *P. falciparum* proliferation assay section below for details). Mycoplasma detection assays were also performed periodically on each of the six parasite cultures using PCR in conjunction with a primer mix that amplifies ribosomal DNA from different mycoplasma strains[58]. No mycoplasma infections were detected in any of the parasite cultures throughout the course of this study.

**PfCRT coding sequences and cRNA synthesis**. The coding sequences of each of the PfCRT isoforms were generated via site-directed mutagenesis[16] using the primers listed in Supplementary Data 6. The DNA template was a codon-harmonised sequence of PfCRT that had been inserted into the pGEM-He-Juel oocyte expression vector[59]. This sequence encodes a version of PfCRT that is free of endosomal–lysosomal trafficking motifs and thus results in the expression of the transporter at the plasma membrane of *Xenopus* oocytes[15,16]. The plasmids were linearised with SalI (ThermoFisher Scientific) from which 5′-capped complementary RNA (cRNA) was synthesised using the mMessage mMachine T7 transcription kit (Ambion), and then purified with the MEGAclear kit (Ambion).

**Isolation and injection of *Xenopus* oocytes**. Adult female frogs were anaesthetised in a solution of ethyl 3-aminobenzoate methanesulfonate salt (Sigma-Aldrich) and 1 mM NaHCO$_3$ (Sigma-Aldrich)[41]. Sections of the ovary were removed and the collagenous membrane encasing the oocytes was degraded by collagenase A and collagenase D (Sigma-Aldrich)[41]. Stage V–VI oocytes were microinjected with cRNA (20 ng per oocyte) encoding a PfCRT isoform or PfNT1, and were stored at 16–18 °C in OR$^{2+}$ buffer supplemented with 50 μg/mL penicillin and streptomycin.

**Transport assays using *Xenopus* oocytes**. The orientation of PfCRT in the parasite's DV membrane, and also in the oocyte plasma membrane, is such that its N- and C-termini extend into the cytosol. The oocytes were suspended in a reaction buffer adjusted to pH 5.5 to mimic the pH of the DV lumen[1,2]. In this scenario, the transport of a substrate via PfCRT from the acidic extracellular solution into the oocyte cytosol[15] is analogous to the parasite's PfCRT-mediated efflux of a substrate from the acidic DV and into the cytosol[15]. Unless specified otherwise, this is the direction in which substrate transport was measured in the oocyte assays.

The radiolabelled compounds were purchased from either Pharmaron ([$^3$H]VF-6, 23 Ci/mmol), American Radiolabelled Chemicals ([$^3$H]CQ, 20 Ci/mmol), PerkinElmer (e.g., [$^3$H]L-aspartate and [$^3$H]hypoxanthine monochloride), Amersham Biosciences ([$^3$H]acetate and [$^3$H]folate) or ICN Pharmaceuticals ([$^3$H]thymine). All of the host-derived peptides were custom-synthesised (GenScript; see Supplementary Data 1 and 2 for peptide sequences).

Unless specified otherwise, the transport assays were conducted over 1.5 h at 27.5 °C and the reaction buffer was ND96 (96 mM NaCl, 2 mM KCl, 1 mM MgCl$_2$, 1.8 mM CaCl$_2$, 10 mM MES and 10 mM Tris-base) at pH 5.5.

Measurements of the *cis*-inhibition of [$^3$H]CQ transport via PfCRT$^{Dd2}$ or PfCRT$^{Ecu1110}$ by a test solute (data presented in Fig. 1 and Supplementary Data 1,2) were performed on oocytes three days post-cRNA-injection[15,21]. Ten oocytes were washed twice with the relevant ND96 buffer. The assay commenced with the addition of ND96 buffer supplemented with 0.25 μM [$^3$H]CQ, 15 μM unlabelled CQ, and the unlabelled solute under study (at a concentration specified in the figure legends and Supplementary Data 1, 2). The assay was terminated by removing the reaction buffer and washing the oocytes with ice-cold ND96 buffer. The oocytes were lysed in 10% (w/v) sodium dodecyl sulfate (SDS) and the radioactivity accumulated within each oocyte was measured with a MicroBeta$^2$ microplate liquid scintillation analyser (PerkinElmer).

The IC$_{50}$ values presented in Fig. 1e were determined in SigmaPlot Windows Version 11.0 by a least-squares fit of the equation $y = y_{min} + [(y_{max} - y_{min})/(1 + ([\text{test compound}]/\text{IC}_{50})^c)]$) to the data, where $y$ is PfCRT-mediated CQ transport, $y_{min}$ and $y_{max}$ are the minimum and maximum values of $y$, and $c$ is a fitted constant. The PfCRT-mediated CQ transport presented in Supplementary Fig. 5 was calculated by subtracting the level of [$^3$H]CQ accumulation that was detected in the negative control oocytes (non-expressing oocytes and oocytes expressing PfCRT$^{3D7}$) from that measured in oocytes expressing PfCRT$^{Dd2}$ or PfCRT$^{Ecu1110}$. At least four independent experiments were performed (on different days and using oocytes from different frogs), and within each experiment measurements were made from ten oocytes per treatment.

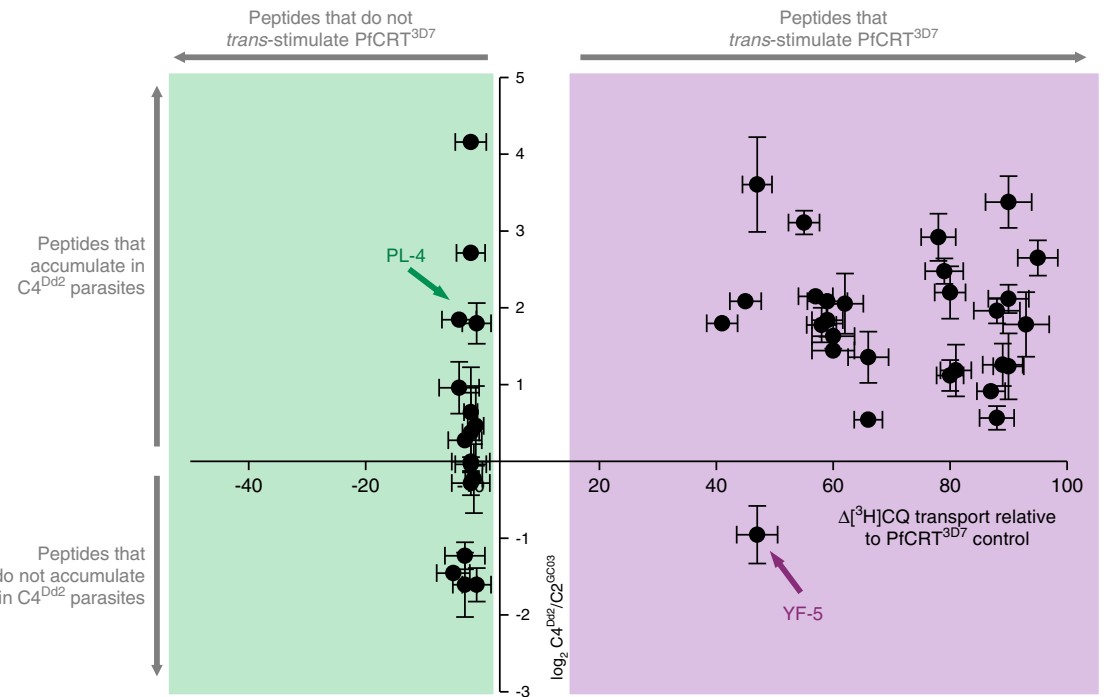

**Fig. 7 The peptide substrates of PfCRT accumulate in parasites expressing mutant isoforms of the transporter.** Host-derived peptides in erythrocytes infected with C2[GC03] or C4[Dd2] parasites were quantified using tandem liquid-chromatography mass-spectrometry and the peptide levels within the C4[Dd2] line were expressed relative to those measured in the C2[GC03] parasites (Supplementary Data 3 and 5). For peptides containing 4–11 residues, a positive relationship exists between the ability to *trans*-stimulate CQ transport via PfCRT[3D7] and accumulation within the CQ-resistant C4[Dd2] parasites. The analysis used the PfCRT[3D7] *trans*-stimulation dataset, rather than that generated for PfCRT[Dd2], because the peptides most likely to accumulate in the C4[Dd2] line will include those that are very poor substrates of (or no longer transported by) PfCRT[Dd2]. An analysis of the data with a Bayesian Information Criteria model identified two distinct populations. The *trans*-stimulation data are the mean of four independent experiments (each yielding similar results) and the peptide accumulation data are the mean of 2–6 independent experiments (each yielding similar results). Error bars are shown for data points that are $n \geq 3$; the $Y$ error is the SD and the $X$ error is the SEM. The source datasets are provided as a Source Data file.

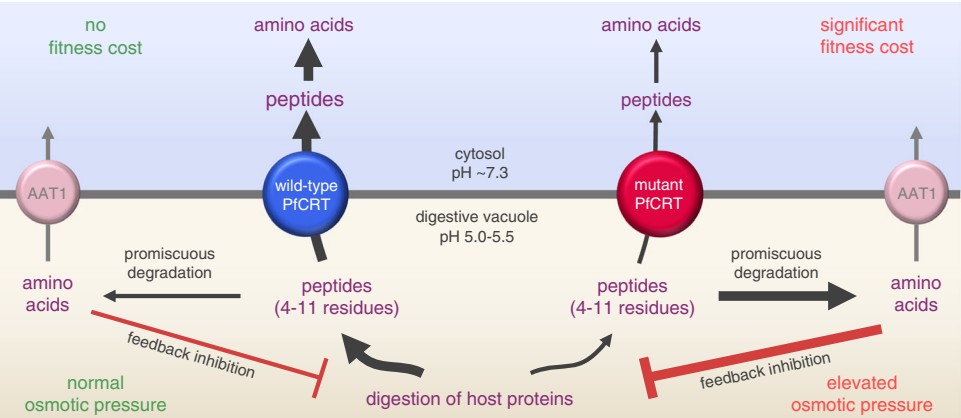

**Fig. 8 Model for the natural function and physiological role of PfCRT.** PfCRT transports host-derived peptides 4–11 residues in length, and of varying composition and charge, out of the DV and into the parasite's cytosol. Wild-type PfCRT transports a broader range of peptides, and has a higher capacity for peptide transport, than most CQ-resistance-conferring isoforms of PfCRT. The diminished abilities of the mutant transporters to efflux peptides results in the accumulation of these peptides within CQ-resistant parasites. Moreover, the promiscuous degradation of the accumulated peptides leads to the build-up of small peptide fragments and amino acids, which exerts further osmotic pressure upon the DV and causes feedback inhibition of the digestion of host proteins. PfCRT therefore serves two roles: (1) it prevents osmotic stress of the DV by exporting peptides and (2) it delivers these peptides to the cytosol where they are degraded into amino acids to fuel parasite growth. AAT1: amino acid transporter 1.

The ability of an unlabelled test solute to *trans*-stimulate [3H]CQ transport via PfCRT (Figs. 2, 5d; Supplementary Fig. 1 and Supplementary Data 1, 2) was measured using oocytes two days post-cRNA-injection[21]. The oocytes were injected with 25 nL of an unlabelled solute (to achieve estimated intracellular concentrations of 5–45 mM) or a control treatment (ND96 buffer pH 5.5 or the dipeptide LH; the estimated intracellular concentration of LH was 45 mM). The intracellular concentrations were calculated using previous estimates of the volume of stage V–VI oocytes (~400 nL[60]). A subset of experiments measured the pH-dependence of the *trans*-stimulatory effect of VF-6 on [3H]CQ uptake via PfCRT. The oocytes were injected with VF-6 suspended in 25 nL ND96 buffer adjusted to pH 5.0, 5.5, 6.0 or 7.0, which corresponded to the injection of 2.5–250 fmol of H+. The Na+-dependence of the *trans*-stimulatory effect of VF-6 was assessed by injecting the oocytes with VF-6 suspended in ND96 buffer (pH 5.5) or a Na+-free version of this buffer

(the Na$^+$ was replaced with choline). The transport of [$^3$H]CQ was measured as described above.

For the data presented in Figs. 2f and 5d, the rate of CQ transport above that measured in the relevant buffer-injected control was calculated. A least-squares fit of the Michaelis–Menten equation ($v = V_{max}$[substrate]/($K_m$ + [substrate])) was then fitted to the data in SigmaPlot Windows Version 11.0 to derive the kinetic parameters for the *trans*-stimulation of [$^3$H]CQ transport by an unlabelled substrate. At least four independent experiments were performed (on different days and using oocytes from different frogs), and within each experiment measurements were made from ten oocytes per treatment.

A total of 95 peptides derived from either haemoglobin or other erythrocytic proteins were tested in the *cis*-inhibition and *trans*-stimulation assays. These tests were conducted in two phases: peptides that were screened before the peptidomic datasets were obtained and those which were selected for study on the basis of these datasets. The first round of testing consisted of 54 peptides that were predicted by an in silico analysis to be generated from human haemoglobin by the parasite-encoded proteases present within the DV. The known cleavage sites of these proteases were plotted within the α- and β-haemoglobin sequences to yield peptide fragments that were likely to exist in the DV[61–66]. The second round of testing was performed with 41 peptides that were selected from the peptidomic datasets as candidate substrates or non-substrates of PfCRT. These experiments were undertaken to determine if the peptidomic results could be used to predict whether a given host-derived peptide is likely to be a substrate of PfCRT.

The characterisation of [$^3$H]VF-6 transport via 23 different field and laboratory-derived isoforms of PfCRT was undertaken with oocytes two days post-cRNA-injection (Figs. 3a–j, 4; Supplementary Fig. 2). The oocytes were injected with 25 nL of ND96 buffer (pH 5.5) supplemented with [$^3$H]VF-6 and unlabelled VF-6 to achieve estimated intracellular concentrations of 1.4 and 200 μM, respectively. Ten oocytes that had resealed at the injection site were transferred to a 5 mL polystyrene round bottom tube (In Vitro Technologies) and washed with 3.5 mL of ND96 buffer before being transferred to separate wells of a white 96-well plate (PerkinElmer). This treatment served to establish the amount of radioactivity within the oocytes immediately prior to the commencement of the transport assay. Another 10 resealed oocytes were transferred to a 5 mL tube and washed with 3.5 mL of ND96 buffer. The assay commenced with the addition of 100 μL of ND96 buffer and the oocytes were incubated at 27.5 °C for 1.5 h (unless specified otherwise; Fig. 3b). The assay was terminated by removing the reaction buffer with a pipette and washing the oocytes twice with 3.5 mL of ice-cold ND96 buffer. Each oocyte was transferred to a separate well of a white 96-well plate. The oocytes were lysed by the addition of 20 μL of 10% (w/v) SDS and overnight incubation at room temperature. The lysed oocyte was then combined with 150 μL of MicroScint-40 microscintillant (PerkinElmer), the plate covered with a TopSeal-A (PerkinElmer), and the radioactivity measured with a MicroBeta$^2$ microplate liquid scintillation analyser (PerkinElmer). VF-6 transport was calculated by subtracting the amount of radioactivity (measured in counts per minute) present within the oocytes at the end of the incubation from that measured in the oocytes sampled immediately before the commencement of the assay. The pH-dependence of transport (Fig. 3c) was determined by injecting the oocytes with [$^3$H]VF-6 suspended in 25 nL ND96 buffer adjusted to pH 5.0, 5.5, 6.0 or 7.0, which corresponded to the injection of 2.5–250 fmol of H$^+$. The dependence of [$^3$H]VF-6 transport on Na$^+$ or Cl$^−$ was examined by injecting the oocytes with [$^3$H]VF-6 suspended in the control ND96 buffer (pH 5.5) or a version of this buffer that lacked either Na$^+$ or Cl$^−$ (Supplementary Fig. 2). Na$^+$ was replaced with 96 mM of KCl and Cl$^−$ was replaced with gluconate salts of Na$^+$ (NaC$_6$H$_{11}$O$_7$; 96 mM), K$^+$ (KC$_6$H$_{11}$O$_7$; 2 mM), Mg$^{2+}$ (MgC$_{12}$H$_{22}$O$_{14}$; 1 mM) and Ca$^{2+}$ (CaC$_{12}$H$_{22}$O$_{14}$; 1.8 mM). The dependence of peptide transport on Ca$^{2+}$ was determined by injecting the oocytes with [$^3$H]VF-6 suspended in ND96 buffers that contained different concentrations of Ca$^{2+}$ (0–5 mM; Supplementary Fig. 2). The dependence of [$^3$H]CQ transport on Na$^+$, Cl$^−$ or Ca$^{2+}$ were conducted by suspending the oocytes in the aforementioned ND96 buffers and measuring the uptake of [$^3$H]CQ transport into the oocyte. The dependence of PfCRT-mediated transport on the membrane potential was examined by using (1) Na$^+$ ionophore III in conjunction with the Na$^+$ gradients listed in Supplementary Data 7 and (2) Cl$^−$ ionophore III in conjunction with the Cl$^−$ gradients listed in Supplementary Data 7 to clamp the membrane potential of the oocyte from −80 to +80 mV. Both the efflux of [$^3$H]VF-6 from the oocyte (Fig. 3d) and the influx of [$^3$H]CQ into the oocyte (Supplementary Fig. 2g) were measured under these conditions. Analyses of the kinetics of VF-6 transport via PfCRT (Figs. 3e and 4a) were undertaken by injecting the oocytes with ND96 buffer pH 5.5 supplemented with [$^3$H]VF-6 (to achieve an estimated intracellular concentration of 1.4 μM) and unlabelled VF-6 (to achieve estimated intracellular concentrations between 0–1150 μM). The kinetic parameters for VF-6 transport via different isoforms of PfCRT (Figs. 3e and 4a) were determined in SigmaPlot Windows Version 11.0 by a least-squares fit of the Michaelis–Menten equation to the data.

A subset of assays measured the ability of solutes known to interact with PfCRT to either *cis*-inhibit, *trans*-inhibit or *trans*-stimulate [$^3$H]VF-6 transport. These experiments involved adding these solutes to either the *cis*- or *trans*-side of the membrane relative to [$^3$H]VF-6. The IC$_{50}$ values presented in Fig. 3h and Supplementary Fig. 2 were determined in SigmaPlot Windows Version 11.0 by a

least-squares fit of the equation $y = y_{min} + [(y_{max} − y_{min})/(1 + ([compound]/IC_{50})^c)]$ to the data, where $y$ is PfCRT-mediated VF-6 transport, $y_{min}$ and $y_{max}$ are the minimum and maximum values of $y$ and $c$ is a fitted constant. At least five independent experiments were performed (on different days and using oocytes from different frogs), and within each experiment measurements were made from ten oocytes per treatment.

Measurements of the uptake or efflux of a number of other radiolabelled compounds (Fig. 3k; Supplementary Fig. 3) were performed using the methods described above for [$^3$H]CQ and [$^3$H]VF-6, respectively.

**Western blot analyses**. Semiquantitative measurements of the level of PfCRT protein in the oocyte membrane were performed using an established Western blot protocol[16] and with oocytes three days post-cRNA-injection. Briefly, the proteins present in the oocyte membrane preparations were separated on a 4–12% NuPAGE Bis-Tris SDS-polyacrylamide gel (ThermoFisher Scientific) and transferred to a 0.45 μM nitrocellulose blotting membrane (Amersham, GE Healthcare Life Sciences). The membranes were probed with rabbit anti-PfCRT antibody (concentration of 1:4000; GenScript) followed by horseradish peroxidase-conjugated goat anti-rabbit antibody (1:8000; ThermoFisher Scientific). Validation of the specificity of the anti-PfCRT antibody has been published in detail elsewhere[16]. The band for each PfCRT isoform was detected with Supersignal West Pico Chemiluminescent reagent (Pierce), quantified using the Image J software[67], and expressed as a percentage of the intensity measured for the PfCRT$^{3D7}$ band. Total protein staining was used to evaluate sample loading and efficiency of transfer[16]. At least five independent experiments were performed (using oocytes from different frogs), and in each experiment measurements were averaged from two separate replicates.

**Immunofluorescence assays**. An immunofluorescence assay was used to localise PfCRT in oocytes three days post-cRNA-injection[21]. The oocytes were fixed with 4% (v/v) paraformaldehyde and permeabilised with 100% methanol. After a series of incubations in blocking solutions, the oocytes were first incubated with the rabbit anti-PfCRT antibody (1:100; GenScript), and then with the Alexa Fluor 488 donkey anti-rabbit antibody (1:500; ThermoFisher Scientific). The oocytes were embedded in an acrylic resin using the Technovit 7100 plastic embedding system (Kulzer). A microtome was used to obtain ~4 μm slices of the oocytes, which were mounted on microscope slides. Images of the slices were obtained with a Leica Sp5 inverted confocal laser microscope (Leica Microsystems) using the ×63 objective. Excitation was achieved with a 488 nm argon laser and the emissions were captured using a 500–550 nm filter. Images were acquired using the Leica Application Suite Advanced Fluorescence software (Leica Microsystems). At least two independent experiments were performed (on oocytes from different frogs) for each oocyte type, within which slices were examined from at least five oocytes. All of the slices taken from oocytes expressing a PfCRT isoform displayed a fluorescent band above the pigment layer (i.e. consistent with the localisation of PfCRT to the plasma membrane) that was not present in non-PfCRT-expressing oocytes.

***P. falciparum* H$^+$-efflux assays**. Saponin-isolated trophozoite-stage parasites containing the membrane-impermant pH-sensitive fluorescent indicator fluorescein-dextran (10,000 MW; Life Technologies) in their DVs were prepared using a lysis-and-resealing technique that has been detailed elsewhere[36,37]. Trophozoite-stage parasites were isolated with saponin and suspended in a saline solution (125 mM NaCl, 5 mM KCl, 1 mM MgCl$_2$, 20 mM glucose, 25 mM HEPES; pH 7.1) at a density of 1 × 10$^7$ cells/mL. The fluorometry experiments were performed as outlined previously[37]. The pH of the DV was monitored at 37 °C using a PerkinElmer Life Sciences LS50B fluorometer with a dual excitation Fast Filter accessory (excitation wavelengths 490 and 450 nm; emission wavelength 520 nm). The experiments entailed monitoring the alkalinisation of the DV upon the addition of the V-type H$^+$-ATPase inhibitor concanamycin A (100 nM; Sigma-Aldrich), in the presence or absence of the test solutes. Half-times for the rate of DV alkalinisation were determined in SigmaPlot Windows Version 11.0 by a least-squares fit of the equation $F = F_0 + F_{max}/[1 + (t/t_{1/2})^c]$, where $F$ is the fluorescence ratio, $F_0$ is the initial fluorescence ratio (averaged over 20 s immediately prior to opening the chamber of the fluorometer and adding the concanamycin A), $t$ is time, $t_{1/2}$ is the half-time for DV alkalinisation, $F_{max}$ is the maximal change in fluorescence ratio and $c$ is a fitted constant[36]. The rate of PfCRT-mediated DV alkalinisation was calculated by subtracting the rate of DV alkalinisation of the solvent control from that of each treatment within each parasite line (Fig. 5a–c; Supplementary Fig. 8). In all cases, five independent experiments were performed on different days.

***P. falciparum* proliferation assays**. Parasite proliferation was measured in 96-well plates using a fluorescent DNA-intercalating dye[68,69]. Synchronous ring-stage parasite cultures (~1% haematocrit and 1% parasitaemia) were incubated with different concentrations of the test compound for 72 h at 37 °C and under reduced O$_2$ conditions. The assay was terminated by freezing and thawing the samples, after which 100 μL of SYBR Safe DNA Gel Stain (Molecular Probes; 0.2 μL/mL) in a lysis buffer (20 mM Tris-HCl, 5 mM EDTA, 0.008% (w/v) saponin and 0.08% (v/v) Triton X-100; pH 7.5) was added to each well. The fluorescence from each well was

measured using a Tecan Infinite M1000 PRO microplate reader (excitation wavelength 490 nm; emission wavelength 520 nm). The fluorescence values for the wells containing the highest concentration of the compound were averaged and this value was then subtracted from each of the well values obtained for that 96-well plate. The level of parasite proliferation in the presence of each compound concentration was expressed as a percentage of the proliferation measured in the absence of the compound. The $IC_{50}$ were determined in SigmaPlot Windows Version 11.0 by a least-squares fit of the equation $y = a/[1 + ([compound]/IC_{50})^c]$ to the data, where $y$ is the percent parasite proliferation, $a$ is the maximum change in the percent parasite proliferation, and $c$ is a fitted constant. In all cases, five independent experiments were performed (on different days), and within each experiment measurements were averaged from three replicates.

**P. falciparum peptidomic analyses.** The isogenic lines C2[GC03], C6[7G8] and C4[Dd2] were maintained at the same developmental stage via sorbitol synchronisation. At the trophozoite-stage, *P. falciparum*-infected erythrocytes were enriched (>95% parasitaemia) via magnetic separation (Colebrook Bioscience) and incubated at 37 °C for 1 h prior to metabolite extraction (to ensure recovery). Two peptide analyses were conducted, one for smaller peptides (< 8mer) and one for larger peptides (>8 mer). Cell density was estimated by haemocytometer and $1 \times 10^8$ and $6 \times 10^8$ cells were used per sample for the small and large peptide analyses, respectively. The cell suspension was transferred into a microcentrifuge tube (2 mL) and metabolites/endogenous peptides were extracted using 80% (v/v) acetonitrile or 90% (v/v) methanol for the small or large peptide analyses, respectively[70]. Following centrifugation, the supernatants were collected and stored at −80 °C until further processing commenced.

Detection and quantification of polar metabolites including endogenous small peptides (typically < 8 mers) was performed on an Agilent 6550 Q-TOF mass spectrometer using a binary gradient with a 1200 series HPLC system (Agilent)[71].

The LC–MS/MS large peptide (>8mer) analysis was performed according to the protocol outlined previously[72]. Methanol extracts (90% (v/v)) were dried under nitrogen. The samples were resuspended in 1% (v/v) formic acid in $H_2O$ and solid-phase extraction was performed using the Oasis TM columns. Recovered samples were lyophilised and resuspended in 50 mM triethylammonium bicarbonate (TEAB) to a concentration of 0.33 µg/µL, with 25 µg of peptide material per sample used for downstream dimethyl labelling. Dimethyl labelling was performed[72]. A fresh 4% (v/v) solution of light ($CH_2O$) and medium ($CD_2O$) formaldehyde was prepared and the light formaldehyde was added to the C2[GC03] samples and the heavy formaldehyde was added to the corresponding C4[Dd2] and C6[7G8] samples. 0.6 M NaBH₃CN was added, incubated for 1 h (ambient) and then quenched with 1% (v/v) ammonia solution. Samples were re-acidified (100% formic acid) and C2[GC03] samples were mixed equally (but separately) to both the C4[Dd2] and C6[7G8] samples. The three biological replicates were also reverse labelled (dimethyl heavy for C2[GC03] and dimethyl light for C4[Dd2] and C6[7G8]) and analysed to ensure accurate quantitation between each cell line.

LC–MS/MS was carried out on a LTQ Orbitrap Elite (Thermo Scientific) with a nanoESI interface in conjunction with an Ultimate 3000 RSLC nanoHPLC (Dionex Ultimate 3000). The LC system was equipped with an Acclaim Pepmap nano-trap column (Dionex-C18, 100 Å, 75 µm × 2 cm) and an Acclaim Pepmap RSLC analytical column (Dionex-C18, 100 Å, 75 µm × 50 cm). The peptides were injected onto the enrichment column at an isocratic flow of 5 µL/min of 3% (v/v) acetonitrile containing 0.1% (v/v) formic acid before the enrichment column was switched in-line with the analytical column. The eluents were 0.1% (v/v) formic acid (solvent A) and 100% (v/v) acetonitrile in 0.1% (v/v) formic acid (solvent B). The flow gradient was (1) 0–6 min at 3% (v/v) solvent B, (2) 6–95 min at 3–20% (v/v) solvent B, (3) 95–105 min at 20–40% (v/v) solvent B, (4) 105–110 min at 40–80% (v/v) solvent B, (5) 110–115 min at 80–85% (v/v) solvent B, (6) 115–117 min at 85–3% (v/v) and (7) equilibrated at 3% (v/v) solvent B for 10 min before the next sample injection. The LTQ Orbitrap Elite spectrometer was operated in the data-dependent mode with nanoESI spray voltage of 1.8 kV, capillary temperature of 250 °C and S-lens RF value of 55%. All spectra were acquired in positive mode with full scan MS spectra from m/z 300–1650 in the FT mode at 240,000 resolution. The automated gain control was set to a target value of 1.0e⁶ and a lock mass of 445.120025 was used. The top 20 most intense precursors were subjected to rapid collision induced dissociation (rCID) with normalised collision energy of 30 and activation q of 0.25. Dynamic exclusion of 30 s was applied for repeated precursors.

The small peptide data collected by LC–MS were converted into mzXML format using MSconvert and analysed using MAVEN[73]. The data were aligned using a polynomial degree of 5 and ion chromatograms were extracted using a permissive threshold (signal/noise > 4) with a mass tolerance of 10 ppm. Peptide identification was achieved with a pre-existing exact mass peptide library and where possible, confirmation with MS-MS spectra matching and/or reference to authentic peptide standards. The data are presented as the mean across 2–6 biological replicates presented as $log_2$ ratios.

The large peptide data were analysed using the MaxQuant (1.5.2.8) software package using the default parameters unless specified. Peptide identification was achieved using a custom protein library containing the top 50 proteins detected within the sample (the most abundantly degraded proteins in the *P. falciparum*-infected erythrocyte), using the unspecific digestion search mode and minimum peptide size of five amino acid residues. Variable modifications included oxidation

and N-terminal acetylation, and no fixed modifications were permitted. Peptide spectral matching false-discovery rate was set to 1% using the reverse decoy database. Peptide quantification was achieved using the dimethyl lysine and dimethyl terminal light and medium labels (0 and 4, respectively) and re-quantification performed when only singletons were detected. The raw heavy/light ratios (reflecting the relative peptide abundance between two cell lines) were compared to the inverse ratios acquired from the label switch data and the average taken between both samples. The mean of the $log_2$ ratios across three biological replicates was calculated and peptides only detected in one replicate were removed from the dataset. The large peptide dataset was then merged with the small peptide dataset and where the same peptide was detected in both datasets the LC–MS/MS quantification was used. The data are presented as the mean across three biological replicates presented as $log_2$ ratios.

Statistical analysis of the peptidomic data presented in Fig. 7 and Supplementary Data 3–5 was performed using the R package mclust[74]. mclust fits a mixture model to the data using an expectation-maximisation algorithm and Bayes Information Criteria to select the optimal number of clusters and to estimate the mean and covariance structure for each cluster.

**Quantification and statistical analyses.** The statistical tests and data analyses were performed using InStat Version 3, Image J Version 1.8.0, SigmaPlot Windows Version 11.0, MAVEN Version 3.6, MaxQuant Version 1.5.2.8 or R statistical software Version 3.6.3. Statistical comparisons were made using one-way ANOVAs in conjunction with either Tukey's multiple comparisons test or a two-tailed one-sample *t*-test corrected for multiple hypothesis testing using the Benjamini–Hochberg procedure. Unless stated otherwise, all errors cited in the text and shown in the figures represent the standard error of the mean (SEM). Where not shown, error bars fall within the symbols. Significance was defined as $P < 0.05$. The key statistical details for each dataset are included in the figure legends.

**Reporting summary.** Further information on research design is available in the Nature Research Reporting Summary linked to this article.

## Data availability
The data supporting the findings of this study are available within the paper and Supplementary Information. Source data are provided with this paper.

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

## Acknowledgements

We thank Prof David Fidock for providing the transfectant *P. falciparum* lines, Prof Jean Chmielewski for providing Q2C, Prof Kiaran Kirk for providing several of the radiolabelled solutes, Dr Teresa Neeman for assistance with statistical modelling, Dr Adele Lehane for helpful discussions and comments on the manuscript, Megan Nash for conducting preliminary *cis*-inhibition oocyte assays, and the Canberra Branch of the Australian Red Cross Blood Service for the provision of blood. This work was supported by the Australian Research Council (fellowship 1053082 to R.E.M.), the L'Oréal Australia For Women in Science program (to R.E.M.), the National Health and Medical Research Council (Project Grants 1007035 and 1127338 to R.E.M., fellowship 1154540 to M.J.M., fellowship 1053082 to R.E.M. and fellowship 1120690 to R.L.S.), the National Institutes of Health (1DP2OD001315-01 to M.L.) and the Australian Department of Education (Australian Postgraduate Awards to S.H.S., S.N.R. and R.L.S.).

## Author contributions

Conceptualisation: S.H.S. and R.E.M.; methodology: S.H.S., S.A.C., M.J.M. and R.E.M.; formal analysis: S.H.S., S.A.C. and R.E.M.; investigation: S.H.S., S.A.C., K.B., S.N.R., N.S.L., R.L.S. and R.E.M.; resources: S.H.S., K.B., S.N.R., N.S.L., M.L. and S.J.H.; writing—original draft: S.H.S., S.A.C. and R.E.M.; writing—review and editing: S.H.S., S.A.C., S.N.R., R.L.S., M.J.M. and R.E.M.; visualisation: S.H.S., S.A.C., R.L.S. and R.E.M.; Supervision: M.J.M. and R.E.M.; project administration: R.E.M.; funding acquisition: R.E.M.

## Competing interests
The authors declare no competing interests.
