## [Peer Review File · Nature Communications]

Reviewers' Comments:

Reviewer #1:

Remarks to the Author:

The manuscript by Shafik et al. explores biochemical and peptidomic approaches to investigate the putative native substrates of the *Plasmodium falciparum* chloroquine resistance transporter (PfCRT) in the *Xenopus laevis* oocyte expression system and parasite lines expressing different PfCRT isoforms. The authors describe a panel of host-derived peptides as the likely substrates of PfCRT and have shown that chloroquine (CQ) resistance-conferring mutations present in the Ecu1110 (South American) and Dd2 (Southeast Asian) variants reduce both the peptide transport capacity and range of peptide substrates that interact with PfCRT. The manuscript is well written and the studies are presented well.

These data provide compelling evidence that PfCRT is able to transport a range of peptides, preferably 4-11 amino acids in length and derived mostly from degraded host haemoglobin. The authors focus on two peptides (termed VF-6 and HM-5) to show that they can serve as transported substrates and also inhibit the transport of tritiated CQ. Observations in *Xenopus* oocytes are complemented by studies with transgenic parasite lines that confirm peptide accumulation in the resistant lines and that show this to be H⁺-dependent, as measured using concanamycin A.

This is an interesting study that is well documented. There is nonetheless some question about the novelty of the findings as haemoglobin-derived peptides have long been implicated in the natural function of PfCRT. Dr. Martin reported preliminary evidence of peptide transport in her 2009 article in *Science* (PMID 19779197). In 2014 the co-author Dr. Llinas published a report in *PLOS Genetics* showing accumulation of peptides in the CQ-resistant Dd2 variant (Lewis et al. 2014 *PLOS Genetics* PMID 24391526). This work was confirmed presumably by the Llinas lab in the article by Lee et al. 2018 *Sci Reports* (PMID 30206341) showing that the L272F mutation resulted in high levels of peptide accumulation. That mutation was first discovered by the Krishna lab (Pulcini et al. 2015, *Sci Reports* PMID 26420308). Nevertheless, this present manuscript presents detailed and extensive evidence showing a wide range of peptides transported by PfCRT and extends this to show differences between the WT and variant isoforms and how they impact drug transport. The work is comprehensive and a notable contribution to our understanding of PfCRT function.

More specific comments are listed below.

1. The authors express transport measurements of [3H]CQ or peptides as proportions normalized (as percentages) to either the PfCRT Dd2 or 3D7 isoforms. This allows a quick visual assessment of the effect of peptides on cis- inhibited or trans-stimulated transport of CQ and substrates. However, it is important to also show absolute values. The authors need to add Supplemental Tables that document the amount of accumulated [3H]CQ in their various oocyte assays. That pertains to multiple figures and panels. What percentage of [3H]CQ accumulation is background and what does that leave in terms of signal? This is vital, as in some of their trans assays (notably Figure 2b), which involve injection of candidate substrates, they ascribe small differences (typically a 10-15% increase over Dd2) as being significant evidence of trans-stimulation of CQ transport (in their case, into the oocyte). Statistical significance here is achieved by the very low variance across experiments. The physiological relevance of trans-stimulation of CQ transport is also not particularly clear. Are they arguing that in parasites the presence of peptides in the cytosol can trans-stimulate CQ efflux? Surely most of those are further degraded quite rapidly by aminopeptidases. Their evidence of cis-inhibition of transport, whereby peptides in the food vacuole can compete with drug transport, is stronger and also more physiologically relevant.

2. The Figure 1 legend reports that [3H]CQ transport was at background levels in non-expressing oocytes as well as those expressing the PfCRT 3D7 isoform. In that case, how in Figure 2C can they

observe a 3 to 4.5-fold trans-stimulation of CQ transport via WT PfCRT with certain peptides, yet no cis-inhibition, when the data with Dd2 show that peptides have a far more substantial effect on CQ transport when presented in cis? They argue that CQ transport by 3D7 is so low as to only be detectable upon peptide trans-stimulation. Again, the presentation of absolute values, with and without background subtraction, is key to enabling readers to evaluate the robustness of their data.

3. Figure 3 is confusing. It provides evidence of [3H]VF-6 being transported out of the oocyte, i.e. in the opposite direction to CQ. Transport is most evident with the 3D7 isoform, and less in Ecu1110 and Dd2 (the latter shows ~40%). In Figures 1 and 2 one sees that VF-6 is most effective at inhibiting CQ transport (via Ecu1110 or Dd2) when it is in cis, i.e. in the external media, with only a modest trans-stimulation of CQ transport. Yet here in Figure 3 transport was only measured going out from the oocyte to the acidic exterior. Why would the authors not have seen transport towards the acidic pH if that were the more natural flow of substrate? Is it an issue of not being able to reach sufficiently high levels of [3H]VF-6 in the medium? Can the authors show the cis data with [3H]VF-6?

4. On lines 91-92 the authors state: "The trans-stimulation of PfCRT_3D7 by VF-6 was pH-dependent, but was unaffected by the removal of Na⁺ from the injection buffer". Transport via PfCRT has widely been shown to be a function of both intravacuolar pH and membrane potential. While the authors have shown a pH effect in the interaction between PfCRT and substrate, the data presented herein make an important omission in not addressing the role of membrane potential in transport. This should be rectified. If it is not possible in the oocyte system then this should be explained as it would represent a caveat to their study.

5. Also, the authors do not address the relationship (if any) between PfCRT and Ca²⁺ ions especially in the context of specific mutations on the transporter. Ca²⁺ homeostasis has been reported to be important to the food vacuole (Garcia et al. 1998 Eur J Cell Biol PMID 9696353; Alves et al. 2011 J Biol Chem PMID 21149448) and certain mutations in PfCRT have been reported to influence Ca²⁺ mobilization in the DV (Lee et al. 2018 Sci Reports, cited above). As such it would have been of interest to test whether different concentrations of this cation influence substrate specificity or transport by PfCRT.

6. On lines 87-91 the authors state: "Transport of peptides via PfCRT was tested at concentrations of up to 45 mM". What is known about the physiological concentrations of the peptides and other candidate solutes tested in the parasite food vacuole? Is it plausible that some metabolites, ions, amino acids or peptides that showed no trans-stimulation signals are actually substrate at high DV concentrations?

7. On lines 175-9 the authors state: "We found that the fusion of polypeptides to the N- or C-termini of PfCRT interfere with its function (Fig. 4c). Whilst some of the modified proteins retain all or most of their capacities for [3H]CQ transport, none are able to transport [3H]VF-6. This observation helps to explain why previous studies, which expressed versions of PfCRT with N- and/or C-terminal fusions, did not delineate the native substrates of PfCRT." Fig. 4C shows vastly different results between the 3xmyc version and the 1x or 4x versions. Can the authors elaborate on this? The text does not seem to match the Figure.

8. Do the authors have a hypothesis on how short-length peptides are likely to be channelled out of the food vacuole into the cytosol for further breakdown into amino acid nutrient source in CQ-sensitive parasites that are also WT at their PfaAT1 locus? To the best of my knowledge there was no association observed between PfaAT1 and CQ resistance in the Wellems (Dd2 and Hb3) cross. Does Dd2 carry a PfaAT1 mutation? What field-based association data if any support AAT1 as being associated with CQ resistance? Many groups from the NIH, Harvard and the Sanger Institute have looked for secondary determinants of CQ resistance without much evidence beyond CRT.

9. Current Supplemental Tables did not have numbers or titles, making cross referencing difficult – an easy fix.

Reviewer #2:

Remarks to the Author:

A. Background

A.1. Chloroquine resistance (CQR) in *Plasmodium falciparum* parasites is caused primarily by mutations in the chloroquine resistance transporter (PfCRT); these mutations confer a gain of function by which PfCRTCQR acts as a carrier to efflux CQ drug from the parasite's digestive vacuole (DV, the acidic compartment where CQ exerts its effect). The wild-type form of the transporter in CQ-sensitive *P. falciparum*, PfCRTCQS, does not transport CQ. Apart from its function in CQR, PfCRT is thought to have an essential physiological role in *P. falciparum*, as attempts to knock out the PfCRT gene have been unsuccessful. Investigations by a number of laboratories including studies with various heterologous expression systems have led to a number of different hypotheses (cf. lines 53 – 56 of the present submission), but identification of PfCRT's physiological role has remained frustratingly elusive.

A.2. Blood stage malaria parasites engulf and catabolize hemoglobin (Hb) from their host erythrocytes to acquire amino acids and clear out the physical space they need to grow. Proteases in the DV digest the Hb in an orderly cascade, producing peptides and amino acids that are transported across the DV membrane into the parasite cytoplasm. Because PfCRT localizes to the DV membrane and could function in peptide/amino acid transport, Lewis et al. (submission ref #23) hypothesized that PfCRT polymorphisms of CQR might affect Hb catabolism and alter peptide profiles in the DV. Analysis by high-resolution nanospray LC-MS/MS identified 362 endogenous peptides ranging from 3-mers to 32-mers that corresponded exactly to sequences in the Hb α or β chains. Further, quantitative analyses identified 87 peptides that showed evidence for differential accumulation between CQS and CQR isogenic parasite lines that differed only by PfCRTCQS or PfCRTCQR allele. A fitness cost was also identified with the mutations in PfCRTCQR. Lewis et al. considered: (1) if PfCRT might function as a peptide transporter and that CQR mutations interfere with this activity; or (2) if PfCRT might have another function and that CQR mutations affect Hb catabolism indirectly by altering the permeability of the digestive vacuole membrane (e.g. causing the DV to leak protons, glutathione, heme, or other osmolytes, thereby altering conditions in the vacuolar compartment). Weighing against the possibility of PfCRT of a peptide transporter, Lewis et al. argued that the broad diversity of sizes up to 32-mers and physical properties of peptides accumulated by CQR parasites are inconsistent with the relatively narrow range of substrates carried by most peptide transporters; conversely, in favor of an indirect effect on Hb catabolism, Lewis et al. observed that protease activities are highly sensitive to solution conditions and that perturbations of these conditions can alter protein-protein interactions including those of the Hb degradation complex. Teng et al. (submission ref #43) also considered the possibility that PfCRT might function as a direct amino acid transporter but in conclusion were inclined to agree with Lewis et al. for an indirect effect to explain the altered DV amino acid profiles of PfCRTCQR vs. PfCRTCQS parasites.

A.3. In previous studies (submission ref #35), Martin et al. expressed wild-type and resistant forms of PfCRT at the surfaces of *Xenopus laevis* oocytes, effectively creating an "inside-out" model of the DV for studies of PfCRT function. Results showed that PfCRTCQR transports CQ, whereas PfCRTCQS does not; further, CQ transport by PfCRTCQR was inhibited by a variety of quinoline compounds and by resistance-reversers including verapamil (VP, 30 μ M IC₅₀). In tests of three dozen peptides (2 – 10 mers) at 1 mM concentration, 10 of 36 peptides cause a pronounced inhibition of CQ transport by PfCRTCQR. Of 8 peptides in this collection thought to be generated from parasite digestion of Hb, three inhibited CQ transport but the other 5 peptides did not (ref #35 Table S5). Most of the inhibitory peptides shared elements of a 'CQR reverser pharmacophore' with VP and the quinoline drugs, with specific structural features likely recognized by PfCRT. One of the 10 inhibitory peptides, YPWF-NH₂ (opioid receptor agonist endomorphin-1) was [³H]-labelled and assessed for uptake into oocytes expressing different PfCRT constructs. Results showed that [³H]YPWF-NH₂ was transported by

PfCRTCQR; however, the same peptide was not transported by PfCRTCQS, suggesting that interaction of this peptide with PfCRTCQR was due to its features in common with VP and the quinoline drugs, instead of structural resemblance to an endogenous substrate.

B. Evaluation of present submission, major criticisms

The *Xenopus laevis* oocytes expression system, methods, and results presented here have much in common with the previous Science paper of Martin et al. (ref #35). But major conclusions are not supported by the present data (B.1 below), are based on results that differ without explanation from those previously presented (B.2 below), or contradict previous published findings without evidence (B.3, B.4 below):

B.1. Discussion lines 253 – 254: “datasets we have obtained using a set of complementary in vitro and in situ assays reveal that PfCRT exports host-derived peptides containing 4-11 residues out of the parasite’s DV.” Abstract lines 21 – 23: “The transport of peptides ... via PfCRT is saturable and inhibited by verapamil and quinoline drugs. Our results indicate that PfCRT transports peptides from the lumen of the parasite’s digestive vacuole to the cytosol, ...”.

Criticism: PfCRT transport of only one peptide (VF-6) was characterized and this transport was in the outward direction from the PfCRT-transformed *Xenopus laevis*; multiple attempts failed to demonstrate uptake into the transformed oocytes from the extracellular solution (lines 114 – 117). In other words, the transport demonstrated for VF-6 is in the “wrong” direction for movement of peptides out of the vacuole into the parasite cytosol and corresponds to influx by PfCRT into the DV. Previously, Martin et al. (ref #35) found that another peptide, YPWF-NH₂ of endomorphin-1, was transported into the PfCRTCQR-transformed *Xenopus laevis* oocytes (the direction corresponding to efflux from the parasite DV), although influx of YPWF-NH₂ did not happen with PfCRTCQS-transformed *Xenopus laevis* oocytes (paragraph A.3 above). PfCRT-transformed *Xenopus laevis* oocytes thus are competent for investigations of peptide transport in both directions; however, the results simply don’t support (and are even opposite to) the authors’ claims.

B.2. Lines 72 – 74, “almost all of the 91 peptides screened strongly cis-inhibited [3H]CQ transport by both PfCRTEcu1110 and PfCRTDd2. The test peptides were 2-13 residues in length and most occur in haemoglobin or other erythrocyte proteins.”

Criticism: In the previous publication of Martin et al., of 8 peptides thought to be generated from parasite digestion of Hb, 3 inhibited CQ transport but the other 5 peptides did not (ref #35 Table S5; paragraph A.3 above). Comparison of those peptide sequences with those of the peptides in the present report (Supplementary spreadsheet Tables S1 and S2) shows entirely different results for peptides that were previously shown not to inhibit CQ transport: GLHAL and LSDLHAK are now listed as strong cis-inhibitors, and peptide sub-fragments of the remaining 3 of the 5 non-inhibitory peptides are also listed as cis-inhibitors (Strangely, data for these 3 as complete peptides do not seem to be in the spreadsheets). These important inconsistencies, not mentioned at all in the submitted manuscript, need to be addressed and explained.

Ref #35 further showed that most of the inhibitory peptides shared elements of a ‘CQR reverser pharmacophore’ with VP and the quinoline drugs, with specific structural features thought to be recognized by PfCRTCQR. Is this also the case for the 91 peptides in the present submission?

B.3. Discussion and Abstract, lines 17 – 18 and 251 – 252: “[We] show that the protein does not mediate the non-specific transport of ions and other metabolites” and “Contrary to previous reports, PfCRT is not a non-specific transporter of metabolites and/or of organic or inorganic ions.”

Criticism: Research studies and publications from several different laboratories have presented evidence for PfCRT activity as a proton, chloride channel, a general cation channel, a transporter of glutathione, an iron atom transporter, a metabolite transporter ... (lines 53 – 56). These possibilities are not examined in the present study and evidence is not presented against those reports. The

statements in lines 17 – 18 and 251 – 252 are therefore unsupportable.

B.4. Lines 195 – 196: “Verapamil and Q2C inhibited the peptide-induced H⁺ leaks in C2GC03, C67G8, and C4Dd2 parasites (Fig. 5b), providing further evidence that these solutes are effluxed from the DV via PfCRT.”

Criticism: To the contrary, Hrycyna and Martin et al. concluded that Q2C (compound 6) is not a substrate of PfCRT and is “not effluxed by the protein” (submission ref #17, pg. 728). This glaring discrepancy needs to be clarified and addressed.

Regarding verapamil, Summers et al. [Cell Mol Life Sci. 2012 Jun;69(12):1967-95] state on page 1975 that “direct measurements of transport are required to confirm that PfCRTCQR possesses the ability to translocate VP (or any other resistance-reverser).” If direct measurements have since then demonstrated PfCRT transport of verapamil, please provide the information and a citation.

The caveats taken from Hrycyna et al. and Summers et al. apply similarly to the authors’ present claim/assumption that evidence of cis-inhibition (Fig. 1) or trans-stimulation (Fig. 2) demonstrates peptide transport via PfCRT.

C. Evaluation, additional criticisms

C.1. The LC-MS/MS studies of Lewis et al. (submission ref #23) identified 362 endogenous peptides ranging from 3-mers to 32-mers that corresponded exactly to sequences in the Hb α or β chains, and quantitative analyses found 87 peptides that showed evidence for differential accumulation between isogenic *P. falciparum* lines that differed only by the PfCRTCQS or PfCRTCQR allele (paragraph A.2 above). In light of those findings, why were results and analyses of the present submission limited to peptides of 4 – 11 amino acids?

C.2. Considering the very different results from transport experiments with [3H]-labeled peptides YPWF-NH₂ and VF-6, have experiments with other [3H]-labeled peptides been attempted to directly demonstrate their influx and/or efflux by PfCRTCQS- and PfCRTCQR-transformed *Xenopus laevis* oocytes? If so, with what results? If not, why haven’t experiments to directly demonstrate the transport of additional [3H]-labeled peptides been a priority, to test the authors’ hypothesis for PfCRT’s natural function?

C.3. In Fig. 6a, peptide trans-stimulation results from PfCRT3D7-transformed (PfCRTCQS-transformed) *Xenopus laevis* oocytes are compared to relative peptide accumulation results from C4Dd2 (CQR) parasites. Why weren’t peptide trans-stimulation results from corresponding PfCRTDd2-transformed oocytes used instead for this comparison? This would be a much better and possibly conclusion-changing comparison, especially considering that, of a set of 89 peptides, 39 peptides trans-stimulated [3H]CQ transport via PfCRT3D7 but only 23 trans-stimulated [3H]CQ transport via PfCRTDd2 (line 86 – 88).

C.4. The trans-stimulation results presented in Fig. 2 and described in lines 79 – 92 were performed at much higher mM peptide concentrations than those used for the cis-inhibition studies. Some discussion of these much higher concentrations, and of their relatively small effects on [3H]CQ accumulation in the trans-stimulation experiments, would be helpful.

C.5. Regarding lines 68 – 69: “... detect unlabelled solutes that inhibit [3H]CQ uptake, consistent with them interacting with the mutant transporter’s substrate-binding cavity (Fig. 1a)”.

In a previous study Bellanca et al. (submission ref #36) found that PfCRT contains multiple substrate-binding sites, and that compounds can compete for transport via PfCRT by mixed-type inhibition. Could mixed-type inhibition be involved in the outcomes shown in Fig. 1 of the present submission? This possibility needs to be addressed.

C.6. Line 213: "Together, these datasets provide a robust demonstration of peptide transport via PfCRT in situ."

For reasons detailed above, this reviewer does not agree with this statement that the datasets provide a robust demonstration of peptide transport via PfCRT in situ – not at all.

Reviewer #3:

Remarks to the Author:

Shafit et al. "The natural function of the malaria parasite's chloroquine resistance 1 transporter"

I would like to congratulate the authors for the very well written manuscript. The data represent a heroic amount of work which provides important insights into the role of PfCRT as a peptide transporter, a very important protein in the malaria field.

Technically, the experiments were well thought of. The data appear to be internally consistent throughout. While I expect that the experiments would be fairly sophisticated for general readers who are not familiar with membrane transport, I think that the authors have done a good job explaining the 'gist' of each experiment.

This manuscript is one of the very few that I do not have things to criticize. I fully recommend the journal to publish this work.

Reviewer #1

The manuscript by Shafik et al. explores biochemical and peptidomic approaches to investigate the putative native substrates of the Plasmodium falciparum chloroquine resistance transporter (PfCRT) in the Xenopus laevis oocyte expression system and parasite lines expressing different PfCRT isoforms. The authors describe a panel of host-derived peptides as the likely substrates of PfCRT and have shown that chloroquine (CQ) resistance-conferring mutations present in the Ecu1110 (South American) and Dd2 (Southeast Asian) variants reduce both the peptide transport capacity and range of peptide substrates that interact with PfCRT. The manuscript is well written and the studies are presented well.

These data provide compelling evidence that PfCRT is able to transport a range of peptides, preferably 4-11 amino acids in length and derived mostly from degraded host haemoglobin. The authors focus on two peptides (termed VF-6 and HM-5) to show that they can serve as transported substrates and also inhibit the transport of tritiated CQ. Observations in Xenopus oocytes are complemented by studies with transgenic parasite lines that confirm peptide accumulation in the resistant lines and that show this to be H⁺-dependent, as measured using concanamycin A.

This is an interesting study that is well documented. There is nonetheless some question about the novelty of the findings as haemoglobin-derived peptides have long been implicated in the natural function of PfCRT. Dr. Martin reported preliminary evidence of peptide transport in her 2009 article in Science (PMID 19779197). In 2014 the co-author Dr. Llinas published a report in PLOS Genetics showing accumulation of peptides in the CQ-resistant Dd2 variant (Lewis et al. 2014 PLOS Genetics PMID 24391526). This work was confirmed presumably by the Llinas lab in the article by Lee et al. 2018 Sci Reports (PMID 30206341) showing that the L272F mutation resulted in high levels of peptide accumulation. That mutation was first discovered by the Krishna lab (Pulcini et al. 2015, Sci Reports PMID 26420308). Nevertheless, this present manuscript presents detailed and extensive evidence showing a wide range of peptides transported by PfCRT and extends this to show differences between the WT and variant isoforms and how they impact drug transport. The work is comprehensive and a notable contribution to our understanding of PfCRT function.

More specific comments are listed below.

Our response: We thank the Reviewer for his/her positive appraisal of our study and for the constructive comments. We believe that addressing these queries has strengthened several datasets, enhanced the clarity of the text and figures, and improved the accessibility of the manuscript to non-specialist readers.

In regard to the novelty of the work and advances over earlier work, there can be no doubt that the natural function of PfCRT remains unresolved. Reviewer 1 suggested that *"haemoglobin-derived peptides have long been implicated in the natural function of PfCRT"*. However, no study has been able to provide evidence of peptide transport via wild-type PfCRT and, as summarised by Reviewer 2, the studies reporting an association between peptide accumulation in the parasite and mutations in PfCRT concluded that PfCRT was unlikely to be a peptide transporter. Moreover, Reviewer 2 emphasised that the *"identification of PfCRT's physiological role has remained frustratingly elusive"*. Our manuscript provides the long-awaited elucidation of PfCRT's natural function and thus resolves a major unanswered question in malaria parasite biology.

1a. The authors express transport measurements of [3H]CQ or peptides as proportions normalized (as percentages) to either the PfCRT Dd2 or 3D7 isoforms. This allows a quick visual assessment of the effect of peptides on cis- inhibited or trans-stimulated transport of CQ and substrates. However, it is important to also show absolute values. The authors need to add Supplemental Tables that document the amount of accumulated [3H]CQ in their various oocyte assays. That pertains to multiple figures and panels. What percentage of [3H]CQ accumulation is background and what does that leave in terms of signal? This is vital, as in some of their trans assays (notably Figure 2b), which involve injection of

candidate substrates, they ascribe small differences (typically a 10-15% increase over Dd2) as being significant evidence of trans-stimulation of CQ transport (in their case, into the oocyte). Statistical significance here is achieved by the very low variance across experiments.

Our response: Almost all of the oocyte-derived datasets presented in the Figures (including Fig. 2b) already show the full transport signal for the radiolabelled solute. In most cases no “background” has been subtracted. The “counts per minute” (cpms) obtained for each treatment were simply normalised to the relevant PfCRT control within that experiment and the resulting values were then averaged across $n = 3-5$ independent experiments and shown \pm the SEM. Hence, datasets such as Fig. 1b,c,e, and Fig. 2b-f show a low level of [^3H]chloroquine accumulation in the negative controls that is attributable to the simple diffusion of uncharged [^3H]chloroquine into the oocyte (Martin et al. *Science* 2009; Summers et al. *Proc Natl Acad Sci U S A* 2014) and which is the “background” referred to by Reviewer 1. In all cases, the vast majority of the [^3H]chloroquine transport signal is due to uptake via PfCRT; **only 3-4% is the “background” diffusion of [^3H]chloroquine**. We retained this background component of [^3H]chloroquine accumulation in most of the figures, but it was necessary to subtract it from the datasets when generating the heatmaps (Fig. 1d and 2g). The heatmaps serve to highlight where an unlabelled solute caused a decrease or increase in [^3H]chloroquine transport via a given PfCRT isoform, which required normalising the PfCRT-mediated component of uptake to the control for that isoform. For example, all of the PfCRT^{3D7} treatments were expressed as a percentage of the PfCRT^{3D7} control. Aside from the heatmaps, all of the [^3H]chloroquine transport figures already show the full transport signal.

Revisions: We have added the following datasets to Supplementary Tables 1,2:

- (1) The **total transport** values for the *cis*-inhibition and *trans*-stimulation [^3H]chloroquine datasets presented in the heatmaps (Fig. 1d and 2g). The “background” value – i.e. the low level of [^3H]chloroquine diffusion into the negative controls – has not been subtracted from these datasets.
- (2) The **PfCRT-mediated transport** values for the *cis*-inhibition and *trans*-stimulation [^3H]chloroquine datasets in a less “normalised” format than that used for the heatmap figures. All of the values generated for transport via PfCRT^{3D7}, PfCRT^{Ecu1110}, and PfCRT^{Dd2} are shown as a percentage of the PfCRT^{Dd2} positive control.
- (3) Examples of the **raw cpms for all oocyte types**, including the negative controls used to estimate the “background” component of [^3H]chloroquine accumulation, and the calculations used to express [^3H]chloroquine transport as a percentage of the cpms obtained for the positive control.

1b. The physiological relevance of trans-stimulation of CQ transport is also not particularly clear. Are they arguing that in parasites the presence of peptides in the cytosol can trans-stimulate CQ efflux? Surely most of those are further degraded quite rapidly by aminopeptidases. Their evidence of *cis*-inhibition of transport, whereby peptides in the food vacuole can compete with drug transport, is stronger and also more physiologically relevant.

Our response: The *trans*-stimulation assay is a valuable method for studying carrier-type transporters because it enables the substrates of the transporter to be identified without the prohibitive expense of purchasing multiple radiolabelled solutes. It is not used to determine the net direction in which substrate transport occurs *in vivo*. In any case, as was noted in the original manuscript, the large outwardly-directed gradients across the digestive vacuole membrane for both protons and peptides would enable little, if any, transport of peptides into the parasite’s vacuole.

We used the *trans*-stimulation assay in conjunction with the oocyte expression system to distinguish between inhibitors and substrates of PfCRT and, once we had identified peptides as the transporter’s natural substrates, to determine the range of peptides transported via PfCRT. We then used the ‘H⁺-efflux’ assay to confirm that PfCRT effluxes peptides out of the digestive vacuoles of live parasites. We did not suggest that peptides would be present at sufficient levels in the parasite cytosol to *trans*-stimulate drug efflux from the digestive vacuole. We did employ the *trans*-stimulation assay to define the substrate-specificity of PfCRT.

Revisions: A Supplementary Note has been added to summarise the phenomenon of *trans*-stimulation, why it is commonly used to elucidate the substrate-specificity of a carrier-type transporter, and how it was harnessed in this study to discover the native substrates of PfCRT. An abbreviated version of this information was already included in the Fig. 2. legend, but the addition of an expanded description should aid in clarifying these details for readers not familiar with this transporter assay. The relevant text in Supplementary Note 2 is as follows:

“Identification of the native substrates of PfCRT using a *trans*-stimulation assay. We employed a *trans*-stimulation assay in conjunction with the oocyte expression system to distinguish between inhibitors and substrates of PfCRT and, once we had identified peptides as the transporter’s natural substrates, to determine the range of peptides transported via PfCRT (Fig. 2b-g, 5d, Supplementary Fig. 1, and Supplementary Tables 1,2).

Trans-stimulation is a phenomenon exhibited by many carrier-type transporters (i.e. the class of membrane transport protein to which PfCRT belongs⁹), whereby the reorientation of the binding site between the two faces of the membrane occurs more rapidly when it is occupied by a substrate than when it is empty. This property means that the PfCRT-mediated uptake of [³H]CQ from the extracellular solution will increase when an unlabelled substrate is added to the cytosolic face of the membrane (Fig. 2a), but not when the unlabelled solute is not translocated by PfCRT⁷. Hence, by measuring the ability of unlabelled solutes to *trans*-stimulate the transport of a labelled substrate, the substrate-specificity of a transporter can be elucidated without the prohibitive expense of purchasing multiple radiolabelled solutes.

Another important characteristic of carrier-type transporters is that any solute shown to be transported in one direction across the membrane can also be transported in the opposite direction. This is a macroscopic consequence of the principle of microscopic reversibility (an adjunct to the laws of thermodynamics). That is, there is no “right” or “wrong” direction and the activity of the protein can be assessed by measuring transport in either direction¹⁰⁻¹³. In this study, we have shown peptide/peptide mimic transport via PfCRT in both directions. For example, the experiments in which the peptide mimic saquinavir was used to *trans*-stimulate the PfCRT-mediated efflux of [³H]VF-6 from the oocyte is an example of transport occurring in the direction that equates to the efflux of the peptide mimic from the parasite’s digestive vacuole. It should be noted, however, that the *trans*-stimulation assay does not reveal the net direction in which substrate transport occurs *in vivo*. In any case, the large outwardly-directed gradients across the digestive vacuole membrane for both protons and peptides would enable little, if any, transport of peptides into the vacuole.

In the experiments giving rise to the datasets presented in Fig. 2b-g, 5d, Supplementary Fig. 1, and Supplementary Tables 1,2, the test solute was injected into the oocyte immediately prior to the commencement of the assay. The approximate intracellular concentrations of the test solutes were 5 and 45 mM (or 35 mM if solubility in the aqueous buffer was limiting), and the injection controls were the buffer-only and LH treatments⁷. The negative controls were non-expressing oocytes and those expressing an unrelated *P. falciparum* transporter, the nucleoside transporter 1 (PfNT1^{14, 15}). The latter demonstrates that the expression of a similarly-sized transporter in *Xenopus* oocytes does not affect the ability of the oocyte membrane to reseal following the injection of a test solute⁷.”

2. The Figure 1 legend reports that [³H]CQ transport was at background levels in non-expressing oocytes as well as those expressing the PfCRT 3D7 isoform. In that case, how in Figure 2C can they observe a 3 to 4.5-fold *trans*-stimulation of CQ transport via WT PfCRT with certain peptides, yet no *cis*-inhibition, when the data with Dd2 show that peptides have a far more substantial effect on CQ transport when presented in *cis*? They argue that CQ transport by 3D7 is so low as to only be detectable upon peptide *trans*-stimulation. Again, the presentation of absolute values, with and without background subtraction, is key to enabling readers to evaluate the robustness of their data.

Our response: As stated in our response to Comment 1a, many of the relevant figures already show the full transport signal for [³H]chloroquine. I.e., in most cases no “background” was subtracted and the full transport values have been presented (including in Fig. 2b,c). The main text of the manuscript notes that chloroquine transport via wild-type PfCRT (PfCRT^{3D7}) is not detected under normal conditions. This has

been observed in both the *Xenopus* oocyte system (using [³H]chloroquine transport assays; Martin et al. *Science* 2009; Summers et al. *Proc Natl Acad Sci U S A* 2014) and in live parasites (using the H⁺-efflux assay; Lehane et al. *J Cell Sci* 2008; Lehane and Kirk *Antimicrob Agents Chemother* 2008; Lehane and Kirk *Mol Microbiol* 2010). The low level of [³H]chloroquine accumulation detected in oocytes expressing PfCRT^{3D7}, which is the same level as that measured in the non-expressing oocytes, is due to the simple diffusion of the neutral species of the drug into the oocyte (Martin et al. *Science* 2009; Summers et al. *Proc Natl Acad Sci U S A* 2014). This component of uptake is the “background” accumulation of [³H]chloroquine that occurs independently of PfCRT, and which was referred to in Comment 1a of Reviewer 1. The process of simple diffusion is not saturable, nor is it inhibited by other solutes. Hence, when the unlabelled solutes are screened for the ability to *cis*-inhibit the uptake of [³H]chloroquine into oocytes, all are without effect in both the non-expressing and PfCRT^{3D7}-expressing oocytes (Fig. 1b,c,e, Supplementary Fig. 1a,b, and Supplementary Tables 1,2) because [³H]chloroquine enters these oocytes only via simple diffusion. By contrast, the uptake of [³H]chloroquine via the two mutant isoforms of PfCRT (PfCRT^{Ecu1110} and PfCRT^{Dd2}) can be *cis*-inhibited by the binding of unlabelled inhibitors or substrates to the transporter.

Only when PfCRT^{3D7} is *trans*-stimulated by millimolar concentrations of one of its unlabelled native substrates is there a modest but significant rate of [³H]chloroquine transport above the background rate of diffusion (Fig. 2, 5d, and Supplementary Tables 1,2). These results show that whilst PfCRT^{3D7} mediates little or no [³H]chloroquine transport under normal conditions, strong *trans*-stimulation of the protein produces a [³H]chloroquine transport signal that is readily detected and reproduced.

Revisions: As outlined above, we have now added sheets to the Supplementary Information that contain (1) the total transport values for the two figure panels where the background was subtracted from the dataset before normalisation to the respective positive control and (2) less normalised forms of the *cis*-inhibition and *trans*-stimulation [³H]chloroquine datasets. We have also added the following text to the Supplementary Notes to expand upon the information already included in the main manuscript:

“**Detecting *cis*-inhibition of [³H]CQ uptake into oocytes expressing PfCRT.** As mentioned in the main text, CQ transport via wild-type PfCRT (PfCRT^{3D7}) is not detected under normal conditions. This has been observed in both the *Xenopus* oocyte system (using [³H]CQ transport assays^{1, 2}) and in live parasites (using the H⁺-efflux assay³⁻⁵). The low level of [³H]CQ accumulation measured in oocytes expressing PfCRT^{3D7}, which is the same level as that present in the non-expressing oocytes, is due to the simple diffusion of the neutral species of the drug into the oocyte¹. This component of uptake is the “background” accumulation of [³H]CQ that occurs independently of PfCRT. The process of simple diffusion is not saturable, nor is it inhibited by other solutes. Hence, when the unlabelled solutes are screened for the ability to *cis*-inhibit the uptake of [³H]CQ into oocytes, all are without effect in both the non-expressing and PfCRT^{3D7}-expressing oocytes (Fig. 1b,c,e, Supplementary Fig. 1a,b, and Supplementary Tables 1,2) because [³H]CQ enters these oocytes only via simple diffusion. By contrast, the uptake of [³H]CQ via the two mutant isoforms of PfCRT (PfCRT^{Ecu1110} and PfCRT^{Dd2}) can be *cis*-inhibited by the binding of unlabelled inhibitors or substrates to the transporter.

It should be noted, however, that when PfCRT^{3D7} is *trans*-stimulated by millimolar concentrations of one of its unlabelled native substrates (see Supplementary Note 2, Fig. 2, 5d, and Supplementary Tables 1,2), a modest but significant rate of [³H]CQ transport is observed above the background rate of diffusion. These results show that whilst PfCRT^{3D7} mediates little or no [³H]CQ transport under normal conditions, strong *trans*-stimulation of the protein produces a [³H]CQ transport signal that is readily detected and reproduced.”

3. Figure 3 is confusing. It provides evidence of [3H]VF-6 being transported out of the oocyte, i.e. in the opposite direction to CQ. Transport is most evident with the 3D7 isoform, and less in Ecu1110 and Dd2 (the latter shows ~40%). In Figures 1 and 2 one sees that VF-6 is most effective at inhibiting CQ transport (via Ecu1110 or Dd2) when it is in *cis*, i.e. in the external media, with only a modest *trans*-stimulation of CQ transport. Yet here in Figure 3 transport was only measured going out from the oocyte to the acidic exterior. Why would the authors not have seen transport towards the acidic pH if

that were the more natural flow of substrate? Is it an issue of not being able to reach sufficiently high levels of [³H]VF-6 in the medium? Can the authors show the *cis* data with [³H]VF-6?

Our response: Several points need to be clarified in the context of this comment. First, the extent to which a given unlabelled solute *cis*-inhibits the translocation of a radiolabelled substrate cannot be taken as an accurate predictor of (1) whether the unlabelled solute is itself a substrate of the transporter or (2) the extent to which it may *trans*-stimulate the transport of the radiolabelled substrate.

For example, a solute can be a potent inhibitor, but poor substrate, of a transporter – and such cases have previously been described for PfCRT (e.g. Bellanca et al. *J Biol Chem* 2014; Richards et al. *PLoS Pathog* 2016; van Schalkwyk et al. *J Infect Dis* 2016; Hrycyna et al. *ACS Chem Biol* 2014). Since a poor substrate will also be a poor *trans*-stimulator (because the low rate of transport results in a correspondingly low rate of carrier reorientation), its potent *cis*-inhibition activity does not translate into a strong *trans*-stimulatory effect. Furthermore, if the transporter has a poly-specific binding cavity (as does PfCRT; Bellanca et al. *J Biol Chem* 2014), there can be cases where the unlabelled solute is a relatively modest *cis*-inhibitor of the transport of another (radiolabelled) substrate, but is nonetheless a strong substrate of the carrier. Such a case has again been demonstrated for PfCRT (van Schalkwyk et al. *J Infect Dis* 2016). The potential for there to be a disconnect between the magnitude of *cis*-inhibition exerted by a solute and its ability to *trans*-stimulate the carrier has been well established in the transporter literature, which is why we tested all of the unlabelled solutes for both *cis*-inhibitory and *trans*-stimulatory activities.

Another important characteristic of carrier-type transporters, such as PfCRT, is that **any solute shown to be transported in one direction across the membrane can also be transported in the opposite direction**. This is a macroscopic consequence of the principle of microscopic reversibility, which is an adjunct to the laws of thermodynamics. For a secondary-active carrier such as PfCRT, the net direction of transport is usually determined by the electrochemical gradient of the driving ion(s). Our study shows that peptide transport via PfCRT is dependent on protons and is also coupled to a second solute (the identity of which remains to be discovered). All three co-substrates must be present on the same side of the membrane for peptide transport to occur. Since the second ion is naturally present within the oocyte cytosol, the peptide [³H]VF-6 is co-injected with protons into the oocyte and its efflux measured. This is entirely consistent with the proton-dependent transport of modified peptides/peptide mimics that we measured with the 'H⁺-efflux' assay in live parasites.

There is no value in presenting the attempts to measure [³H]VF-6 uptake into the oocytes because in all cases the cpm's remained at the background level. This was definitely not due to an insufficient concentration of [³H]VF-6 in the extracellular buffer. It is instead due to the absence of the second co-substrate from the extracellular buffer. The translocation step does not occur if only the peptide and proton have bound – the second co-substrate must also bind for transport to occur.

Revisions: The following text has been added to the Supplementary Notes to expand upon the information already included in the main manuscript:

“Rationale for including the solutes with little or no *cis*-inhibitory activity against PfCRT in the *trans*-stimulation assay. The extent to which a given unlabelled solute *cis*-inhibits the translocation of a radiolabelled substrate is not always an accurate predictor of (1) whether the unlabelled solute is itself a substrate of the transporter or (2) the extent to which it may *trans*-stimulate the transport of the radiolabelled substrate.

For example, a solute can be a potent inhibitor, but a poor substrate, of a transporter – and such cases have previously been described for PfCRT⁶⁻⁸. Since a poor substrate will also be a poor *trans*-stimulator (because the low rate of transport results in a correspondingly low rate of carrier reorientation), its potent *cis*-inhibition activity does not translate into a strong *trans*-stimulatory effect. Furthermore, if the transporter has a poly-specific binding cavity (as does PfCRT⁶), there can be cases where the unlabelled solute is a relatively modest *cis*-inhibitor of the transport of another (radiolabelled) substrate, but is nonetheless a strong substrate of the carrier. Such a case has been demonstrated for PfCRT⁸. Given the

potential for there to be a disconnect between the magnitude of *cis*-inhibition exerted by a solute and its ability to *trans*-stimulate the carrier, we tested **all** of the unlabelled solutes for both *cis*-inhibitory and *trans*-stimulatory activities.”

4. On lines 91-92 the authors state: “The *trans*-stimulation of PfCRT_{3D7} by VF-6 was pH-dependent, but was unaffected by the removal of Na⁺ from the injection buffer”. Transport via PfCRT has widely been shown to be a function of both intravacuolar pH and membrane potential. While the authors have shown a pH effect in the interaction between PfCRT and substrate, the data presented herein make an important omission in not addressing the role of membrane potential in transport. This should be rectified. If it is not possible in the oocyte system then this should be explained as it would represent a caveat to their study.

Our response: Agreed.

Revisions: The dependence of PfCRT-mediated transport on the oocyte membrane potential was measured for both [³H]VF-6 efflux and [³H]chloroquine influx and the resulting datasets have been included in the revised manuscript (Fig. 3d and Supplementary Fig. 2g, respectively). We had previously shown that chloroquine uptake via PfCRT^{Dd2} decreases when the oocyte membrane potential is depolarised (Martin et al. *Science* 2009), which is consistent with the translocation step involving the movement of a net positive charge – chloroquine and at least one proton – into the oocyte. The new dataset presented in Supplementary Fig. 2g confirms this finding and characterises the effect in greater detail – i.e. numerous measurements of [³H]chloroquine influx were made between -80 mV and +80 mV. Application of the same assay to the efflux of [³H]VF-6 revealed that the rate of PfCRT-mediated peptide efflux steadily decreases as the oocyte membrane potential becomes more positive. This finding indicates that the efflux of VF-6 and its co-substrates results in the movement of a net negative charge out of the oocyte. Given that the peptide is negatively charged and is translocated in symport with a proton, one possible scenario would be that the second co-transported ion is also negatively-charged (as this would produce a net negative charge). The relevant section of the Results now reads:

“The efflux of [³H]VF-6 from PfCRT-expressing oocytes was linear over time for at least 2 hours and was dependent on both the pH of the injection buffer and the oocyte membrane potential (see also Supplementary Fig. 2 and Supplementary Note 4).”

We have also included a more detailed description of the findings as a Supplementary Note:

“The effect of the oocyte membrane potential on transport via PfCRT. We have previously shown that CQ uptake via PfCRT^{Dd2} decreases when the oocyte membrane potential is depolarised¹, which is consistent with the translocation step involving the movement of a net positive charge – CQ and at least one proton – into the oocyte. The dataset presented in Supplementary Fig. 2g confirms this finding and characterises the effect in greater detail – i.e. numerous measurements of [³H]CQ influx were made between -80 mV and +80 mV.

Application of a similar assay to the efflux of [³H]VF-6 from the oocyte revealed that the rate of PfCRT-mediated transport steadily decreases as the oocyte membrane potential becomes more positive. As mentioned in the main text, peptide transport via PfCRT is dependent both on protons and a second solute that remains to be identified, but which is naturally present within the oocyte. All three co-substrates must be present on the same side of the membrane for the transport of host-derived peptides to occur. We therefore co-injected [³H]VF-6 and protons into the oocyte and measured peptide efflux as the potential was varied between -80 mV and +80 mV. The resulting dataset (Fig. 3d) indicates that the efflux of [³H]VF-6 and its co-substrates involves the movement of a net negative-charge out of the oocyte. Given that the peptide is negatively-charged and is translocated in symport with a proton, one possible scenario would be that the second co-transported ion is also negatively-charged (as this would produce a net negative charge).

In this regard, we note that several substrates of PfCRT – e.g. CQ, other quinoline-type drugs, the

peptide mimic saquinavir, and the endomorphin peptide YPWF-NH₂ – only require protons to undergo translocation. That is, their transport is decoupled from the second ion, thus enabling measurements of their uptake from the extracellular solution into the oocyte. The decoupling of some substrates, but not others, from the translocation of a co-substrate is a known phenomenon for carrier-type transporters²⁰.”

5. Also, the authors do not address the relationship (if any) between PfCRT and Ca²⁺ ions especially in the context of specific mutations on the transporter. Ca²⁺ homeostasis has been reported to be important to the food vacuole (Garcia et al. 1998 Eur J Cell Biol PMID 9696353; Alves et al. 2011 J Biol Chem PMID 21149448) and certain mutations in PfCRT have been reported to influence Ca²⁺ mobilization in the DV (Lee et al. 2018 Sci Reports, cited above). As such it would have been of interest to test whether different concentrations of this cation influence substrate specificity or transport by PfCRT.

Our response: Agreed.

Revisions: We undertook these experiments and found that varying the Ca²⁺ concentration had no effect on the transport of [³H]VF-6 or [³H]chloroquine via PfCRT. The datasets are presented in Supplementary Fig. 2e,f of the revised manuscript. The relevant section of the Results now reads:

“The efflux of [³H]VF-6 from PfCRT-expressing oocytes was linear over time for at least 2 hours and was dependent on both the pH of the injection buffer and the oocyte membrane potential (see also Supplementary Fig. 2 and Supplementary Note 4). By contrast, removal of Na⁺ or Cl⁻ from the injection buffer, or varying the Ca²⁺ concentration, were all without effect (Supplementary Fig. 2).”

6. On lines 87-91 the authors state: “Transport of peptides via PfCRT was tested at concentrations of up to 45 mM”. What is known about the physiological concentrations of the peptides and other candidate solutes tested in the parasite food vacuole? Is it plausible that some metabolites, ions, amino acids or peptides that showed no *trans*-stimulation signals are actually substrate at high DV concentrations?

Our response: During the 48 hours it grows and replicates within an erythrocyte, the malaria parasite consumes 50-80% of the host cell's haemoglobin. The intraerythrocytic concentration of haemoglobin is ~5 mM and its quaternary structure consists of four haem molecules per haemoglobin tetramer (Kumar and Bandyopadhyay *Toxicol Lett* 2005). Some of the peptide sequences occur twice in the haemoglobin tetramer (e.g. VF-6), others occur six (DL and HL) or eight (VD) times.

Several other factors need to be considered when estimating the level to which a given peptide is likely to be present within the vacuole. First, the volume of the erythrocyte is ~75 fl (Saliba et al. *J Biol Chem* 1998) and the volume of the digestive vacuole is estimated to reach ~7 fl during the trophozoite stage of the parasite (Saliba et al. *J Biol Chem* 1998; Pulcini et al. *Sci Rep* 2015). Secondly, haemoglobin digestion begins in the ring stage and continues throughout the trophozoite stage (to at least 32 hours post-invasion), and over this period PfCRT will be working to remove these peptides from the vacuole. That is, only a fraction of the peptide generated by the degradation of the host's haemoglobin will be present in the vacuole at any one time. If 50-80% of the host's haemoglobin was degraded into peptides, and none of these peptides were exported from the vacuole or promiscuously degraded, the concentration of VF-6 would be 27 mM (5 mM × 75 fl × 0.5 ÷ 7 fl) to 43 mM (5 mM × 75 fl × 0.8 ÷ 7 fl). However, the actual concentration will be much lower than these values because haemoglobin digestion takes a number of hours to complete, and the resulting peptides are constantly being exported from the vacuole. The two concentrations we tested – 5 mM and 35/45 mM – are thus approximations of the levels to which peptides – and other metabolites/catabolites – could be expected to accumulate within the digestive vacuole.

Moreover, any solute that fails to elicit a *trans*-stimulation signal when present at 35/45 mM is not a

physiologically-relevant substrate of the carrier. The digestion of haemoglobin (and of other host cell proteins) will generate many different species of peptide substrates for PfCRT; all of these will be present at millimolar levels and all will compete for translocation via PfCRT. Under such highly competitive and saturating conditions, a poor substrate of the carrier will undergo negligible transport.

Revisions: The following text has been added to Supplementary Note 2.

“These two solute concentrations are also approximations of the levels to which peptides – and other metabolites/catabolites – could be expected to accumulate within the digestive vacuole. For example, during the 48 hours it grows and replicates within an erythrocyte, the malaria parasite consumes 50-80% of the host cell’s haemoglobin. The intraerythrocytic concentration of haemoglobin is ~5 mM and its quaternary structure consists of four haem molecules per haemoglobin tetramer¹⁷. Some of the peptide sequences occur twice in the haemoglobin tetramer (e.g. VF-6), others occur six (DL and HL) or eight (VD) times. Several other factors need to be considered when estimating the level to which a given peptide is likely to be present within the vacuole. First, the volume of the erythrocyte is ~75 fl¹⁸ and the volume of the digestive vacuole is estimated to reach ~7 fl during the trophozoite stage of the parasite^{18, 19}. Secondly, haemoglobin digestion begins in the ring stage and continues throughout the trophozoite stage (to at least 32 hours post-invasion), and over this period PfCRT will be working to remove these peptides from the vacuole. That is, only a fraction of the peptide generated by the degradation of the host’s haemoglobin will be present in the vacuole at any one time.

If 50-80% of host’s haemoglobin was degraded into peptides, and none of these peptides were exported from the vacuole or promiscuously degraded, the concentration of VF-6 would be 27 mM ($5 \text{ mM} \times 75 \text{ fl} \times 0.5 \div 7 \text{ fl}$) to 43 mM ($5 \text{ mM} \times 75 \text{ fl} \times 0.8 \div 7 \text{ fl}$). However, the actual concentration will be much lower than these values because haemoglobin digestion takes a number of hours to complete, and the resulting peptides are constantly being exported from the vacuole. That said, the digestion of haemoglobin (and of other host cell proteins) will generate many different species of peptide substrates for PfCRT; all of these will be present at millimolar levels and all will compete for translocation via PfCRT. Under such highly competitive and saturating conditions, a poor substrate of the carrier will undergo negligible transport. Hence, any solute that fails to elicit a *trans*-stimulation signal when present at 35/45 mM is not a physiologically-relevant substrate of the carrier.”

7. On lines 175-9 the authors state: “We found that the fusion of polypeptides to the N- or C-termini of PfCRT interfere with its function (Fig. 4c). Whilst some of the modified proteins retain all or most of their capacities for [³H]CQ transport, none are able to transport [³H]VF-6. This observation helps to explain why previous studies, which expressed versions of PfCRT with N- and/or C-terminal fusions, did not delineate the native substrates of PfCRT.” Fig. 4C shows vastly different results between the 3xmyc version and the 1x or 4x versions. Can the authors elaborate on this? The text does not seem to match the Figure.

Our response: This comment appears to have arisen from a misreading of the two datasets presented in Fig. 4c, which show either [³H]chloroquine transport (left panel) or [³H]VF-6 transport (right panel) via various polypeptide-tagged versions of PfCRT.

To clarify, we found that the addition of a tag to PfCRT – regardless of polypeptide sequence, length, or it being at the N- or C-terminus – abolished VF-6 transport activity (Fig. 4c, right panel). Hence, in regard to PfCRT’s ability to transport its native substrate, which was the focus of this paragraph in the Results, there was no difference in the abilities of the tagged PfCRT proteins to transport [³H]VF-6 – irrespective of whether they were tagged with 1, 3, or 4 myc polypeptides. All tagged versions of PfCRT failed to transport the peptide. By contrast, only one of the tagged PfCRT^{Dd2} proteins lost all chloroquine transport activity (PfCRT^{Dd2} carrying a C-terminal 3xmyc tag; Fig. 4c, left panel). All the other tagged proteins retained full or substantial chloroquine transport activity.

These findings suggest that the extension of either termini prevents the peptide from entering the binding

cavity and/or from binding therein. This could occur because the termini are located near the substrate-binding cavity, and the added polypeptide juts either into, or across the top of, the cavity. Alternatively, the effect could be indirect, whereby the terminus is not proximal to the cavity and the addition of the polypeptide induces a conformational change in PfCRT that prevents the peptide from accessing, or binding to, the cavity. The observation that all but one of the tagged proteins retained the ability to translocate chloroquine suggests that the drug, which is much smaller than the peptide, is able to access, and bind to, the altered cavity. It is not clear why the C-terminal 3xmyc tag obstructs chloroquine transport; it appears that there is a subtle, but functionally significant, difference in the conformation of the transporter between the version carrying the 3xmyc tag and those carrying 1xmyc, 4xmyc, or 1xHA. Further insight into this phenomenon could be gained from structural determinations of PfCRT. However, only one structure of PfCRT has been published thus far (Kim et al. *Nature* 2019), and it did not encompass the N- or C-terminus of the transporter.

Revisions: We have reworded the relevant paragraph in the Results section to avoid the reader confusing the [³H]chloroquine dataset presented in the left panel of Fig. 4c with the [³H]VF-6 dataset presented in the right panel of the same figure. The revised text now reads:

“We found that the fusion of polypeptides to the N- or C-termini of PfCRT interfere with its natural function. Whilst only one of the modified PfCRT^{Dd2} proteins no longer mediates [³H]CQ transport (Fig. 4c left panel), all of the tagged versions of PfCRT^{3D7} and PfCRT^{Dd2} have lost the ability to transport [³H]VF-6 (Fig. 4c right panel). This observation helps to explain why previous studies, which expressed versions of PfCRT with N- and/or C-terminal fusions, did not delineate the native substrates of PfCRT (Supplementary Note 5).”

Moreover, we have likewise reworded and rearranged the relevant Figure legend to improve clarity. The relevant text now reads:

“c, [³H]CQ transport (left) and [³H]VF-6 transport (right) via epitope-tagged versions of PfCRT^{3D7} and PfCRT^{Dd2}. The version of PfCRT^{Dd2} carrying a C-terminal 3xmyc tag does not mediate [³H]CQ transport. The other four variants of PfCRT^{Dd2} retain all or most of their CQ transport activity. By contrast, none of the tagged versions of PfCRT^{3D7} or PfCRT^{Dd2} transport [³H]VF-6. The fusion of polypeptides to PfCRT can therefore abolish its ability to transport peptides, even when the protein remains able to transport CQ.”

We have also added the following Supplementary Note to expand upon the information already provided in the main manuscript:

“The addition of polypeptide sequences to the termini of PfCRT abolishes VF-6 transport activity. We found that the addition of a tag to PfCRT – regardless of polypeptide sequence, length, or it being at the N- or C-terminus – abolishes VF-6 transport activity (Fig. 4c, right panel). That is, regardless of whether PfCRT is tagged with 1xmyc, 3xmyc, 4xmyc, or 1xHA, in all cases the modified proteins fail to transport the peptide. By contrast, only one of the tagged PfCRT^{Dd2} proteins is unable to transport CQ (PfCRT^{Dd2} carrying a C-terminal 3xmyc tag; Fig. 4c, left panel). The other four variants of PfCRT^{Dd2} retain all or most of their CQ transport activity.

These findings suggest that the extension of either termini prevents the peptide from entering the binding cavity and/or from binding therein. This could occur because the termini are located near the substrate-binding cavity, and the added polypeptide juts either into, or across the top of, the cavity. Alternatively, the effect could be indirect, whereby the terminus is not proximal to the cavity and the addition of the polypeptide induces a conformational change in PfCRT that prevents the peptide from accessing, or binding to, the cavity. The observation that all but one of the tagged PfCRT^{Dd2} proteins retain the ability to translocate CQ suggests that the drug, which is much smaller than the peptide, is able to access, and bind to, the altered cavity. It is not clear why the C-terminal 3xmyc tag obstructs CQ transport; it appears that there is a subtle, but functionally significant, difference in the conformation of the transporter between the version carrying the 3xmyc tag and those carrying 1xmyc, 4xmyc, or 1xHA. Further insight into this phenomenon could be gained from structural determinations of PfCRT. However,

only one structure of PfCRT has been published thus far²¹, and it did not encompass the N- or C-terminus of the transporter.”

8. Do the authors have a hypothesis on how short-length peptides are likely to be channelled out of the food vacuole into the cytosol for further breakdown into amino acid nutrient source in CQ-sensitive parasites that are also WT at their PfAAT1 locus? To the best of my knowledge there was no association observed between PfAAT1 and CQ resistance in the Wellem (Dd2 and Hb3) cross. Does Dd2 carry a PfAAT1 mutation? What field-based association data if any support AAT1 as being associated with CQ resistance? Many groups from the NIH, Harvard and the Sanger Institute have looked for secondary determinants of CQ resistance without much evidence beyond CRT.

Our response: In the model presented in Fig. 6b, the CQ-sensitive parasites do not accumulate high levels of di- or tri-peptides because the longer peptides are exported by PfCRT^{3D7} before extensive degradation can occur, and any short-length peptides that are generated are proposed to be exported by PfAAT1 (either as small peptides or, following promiscuous degradation, as amino acids).

Polymorphisms in PfAAT1 were linked with CQ resistance in a genome-wide association analysis of parasites from the China-Myanmar border (Wang et al. *Sci. Rep.* 2016) and mutations in PfAAT1 have also been associated with resistance to MMV007224, MMV668399, and MMV011895 *in vitro* (Cowell et al. *Science* 2018). Moreover, a mutation (A173E) in PcAAT1 was a determinant of CQ resistance in *P. chabaudi* (Kinga Modrzyńska et al. *BMC Genomics* 2012).

Revisions: None required.

9. Current Supplemental Tables did not have numbers or titles, making cross referencing difficult – an easy fix.

Our response: Agreed.

Revisions: Titles and table numbers have been added to the Supplementary files.

Reviewer #2

A. Background

A.1. Chloroquine resistance (CQR) in *Plasmodium falciparum* parasites is caused primarily by mutations in the chloroquine resistance transporter (PfCRT); these mutations confer a gain of function by which PfCRT^{CQR} acts as a carrier to efflux CQ drug from the parasite's digestive vacuole (DV, the acidic compartment where CQ exerts its effect). The wild-type form of the transporter in CQ-sensitive *P. falciparum*, PfCRT^{QS}, does not transport CQ. Apart from its function in CQR, PfCRT is thought to have an essential physiological role in *P. falciparum*, as attempts to knock out the PfCRT gene have been unsuccessful. Investigations by a number of laboratories including studies with various heterologous expression systems have led to a number of different hypotheses (cf. lines 53 – 56 of the present submission), but identification of PfCRT's physiological role has remained frustratingly elusive.

A.2. Blood stage malaria parasites engulf and catabolize hemoglobin (Hb) from their host erythrocytes to acquire amino acids and clear out the physical space they need to grow. Proteases in the DV digest the Hb in an orderly cascade, producing peptides and amino acids that are transported across the DV membrane into the parasite cytoplasm. Because PfCRT localizes to the DV membrane and could function in peptide/amino acid transport, Lewis et al. (submission ref #23) hypothesized that PfCRT polymorphisms of CQR might affect Hb catabolism and alter peptide profiles in the DV. Analysis by

high-resolution nanospray LC-MS/MS identified 362 endogenous peptides ranging from 3-mers to 32-mers that corresponded exactly to sequences in the Hb α or β chains. Further, quantitative analyses identified 87 peptides that showed evidence for differential accumulation between CQS and CQR isogenic parasite lines that differed only by PfcRTCQS or PfcRTCQR allele. A fitness cost was also identified with the mutations in PfcRTCQR. Lewis et al. considered: (1) if PfcCRT might function as a peptide transporter and that CQR mutations interfere with this activity; or (2) if PfcCRT might have another function and that CQR mutations affect Hb catabolism indirectly by altering the permeability of the digestive vacuole membrane (e.g. causing the DV to leak protons, glutathione, heme, or other osmolytes, thereby altering conditions in the vacuolar compartment). Weighing against the possibility of PfcCRT of a peptide transporter, Lewis et al. argued that the broad diversity of sizes up to 32-mers and physical properties of peptides accumulated by CQR parasites are inconsistent with the relatively narrow range of substrates carried by most peptide transporters; conversely, in favor of an indirect effect on Hb catabolism, Lewis et al. observed that protease activities are highly sensitive to solution conditions and that perturbations of these conditions can alter protein-protein interactions including those of the Hb degradation complex. Teng et al. (submission ref #43) also considered the possibility that PfcCRT might function as a direct amino acid transporter but in conclusion were inclined to agree with Lewis et al. for an indirect effect to explain the altered DV amino acid profiles of PfcRTCQR vs. PfcRTCQS parasites.

A.3. In previous studies (submission ref #35), Martin et al. expressed wild-type and resistant forms of PfcCRT at the surfaces of *Xenopus laevis* oocytes, effectively creating an “inside-out” model of the DV for studies of PfcCRT function. Results showed that PfcRTCQR transports CQ, whereas PfcRTCQS does not; further, CQ transport by PfcRTCQR was inhibited by a variety of quinoline compounds and by resistance-reversers including verapamil (VP, 30 μ M IC₅₀). In tests of three dozen peptides (2 – 10 mers) at 1 mM concentration, 10 of 36 peptides cause a pronounced inhibition of CQ transport by PfcRTCQR. Of 8 peptides in this collection thought to be generated from parasite digestion of Hb, three inhibited CQ transport but the other 5 peptides did not (ref #35 Table S5). Most of the inhibitory peptides shared elements of a ‘CQR reverser pharmacophore’ with VP and the quinoline drugs, with specific structural features likely recognized by PfcCRT. One of the 10 inhibitory peptides, YPWF-NH2 (opioid receptor agonist endomorphin-1) was [³H]-labelled and assessed for uptake into oocytes expressing different PfcCRT constructs. Results showed that [³H]YPWF-NH2 was transported by PfcRTCQR; however, the same peptide was not transported by PfcRTCQS, suggesting that interaction of this peptide with PfcRTCQR was due to its features in common with VP and the quinoline drugs, instead of structural resemblance to an endogenous substrate.

B. Evaluation of present submission, major criticisms

The *Xenopus laevis* oocytes expression system, methods, and results presented here have much in common with the previous *Science* paper of Martin et al. (ref #35). But major conclusions are not supported by the present data (B.1 below), are based on results that differ without explanation from those previously presented (B.2 below), or contradict previous published findings without evidence (B.3, B.4 below):

Our response: We appreciate the time Reviewer 2 invested in assessing our manuscript, including the effort he/she made to refer to a number of our previous manuscripts.

However, as detailed below in our responses to Reviewer 2’s comments, we believe that many of his/her concerns have arisen from (1) misinterpretation of the purpose and outcomes of several of the assays utilised in this study, (1) overlooking key methodological differences between this manuscript and Martin et al. *Science* 2009, and (3) misunderstanding, and/or having only a passing knowledge of, other studies we have published (e.g. Martin et al. *Science* 2009; Hrycyna et al. *ACS Chemical Biology* 2014; Bellanca et al. *JBC* 2014; Summers, Nash, & Martin *Cell Mol Life Sci* 2012). We have elaborated upon these points below and have implemented a range of changes to the manuscript and figures to address the Reviewer’s concerns and to improve clarity for readers not familiar with detailed characterisations of transporters, including the specialised assays employed to elucidate their activities.

B.1. Discussion lines 253 – 254: “datasets we have obtained using a set of complementary *in vitro* and *in situ* assays reveal that PfCRT exports host-derived peptides containing 4-11 residues out of the parasite’s DV.” Abstract lines 21 – 23: “The transport of peptides ... via PfCRT is saturable and inhibited by verapamil and quinoline drugs. Our results indicate that PfCRT transports peptides from the lumen of the parasite’s digestive vacuole to the cytosol, ...”.

Criticism: PfCRT transport of only one peptide (VF-6) was characterized and this transport was in the outward direction from the PfCRT-transformed *Xenopus laevis*; multiple attempts failed to demonstrate uptake into the transformed oocytes from the extracellular solution (lines 114 – 117). In other words, the transport demonstrated for VF-6 is in the “wrong” direction for movement of peptides out of the vacuole into the parasite cytosol and corresponds to influx by PfCRT into the DV. Previously, Martin et al. (ref #35) found that another peptide, YPWF-NH₂ of endomorphin-1, was transported into the PfCRT^{QCR}-transformed *Xenopus laevis* oocytes (the direction corresponding to efflux from the parasite DV), although influx of YPWF-NH₂ did not happen with PfCRT^{QCS}-transformed *Xenopus laevis* oocytes (paragraph A.3 above). PfCRT-transformed *Xenopus laevis* oocytes thus are competent for investigations of peptide transport in both directions; however, the results simply don’t support (and are even opposite to) the authors’ claims.

Our response: These claims, which are incorrect, appear to have arisen from misinterpretations of our datasets and/or a misunderstanding of what the transport assays can (or cannot) reveal about a transporter’s activity.

To clarify, we have used the *trans*-stimulation assay in conjunction with the oocyte expression system to show that 46 peptides/peptide mimics are substrates of PfCRT. Furthermore, we studied the PfCRT-mediated transport of 4 of these peptides/peptide mimics in some detail (again using the *trans*-stimulation assay) and a comprehensive characterisation was performed with a radiolabelled form of the hexapeptide VF-6 (the custom-synthesis of which cost \$16,500). Moreover, we measured the transport of 8 modified peptides/peptide mimics via PfCRT in live parasites using the ‘H⁺-efflux’ assay. Hence, we did not limit our study to a single peptide – we have instead detected the transport of a multitude of peptides/peptide mimics via PfCRT.

Secondly, PfCRT is a carrier-type transporter, and carriers can transport their substrates in either direction. This is a macroscopic consequence of the principle of microscopic reversibility (an adjunct to the laws of thermodynamics). That is, there is no “right” or “wrong” direction and substrate specificity, kinetics, ion-dependence etc can be assessed by measuring transport in either direction (e.g. Martin and Kirk *Blood* 2007; Marchetti et al. *Nat Commun* 2015; Hapuarachchi et al. *PLoS pathogens* 2017; Elliott et al. *Biochem J* 2001). We described this property in our response to Comment 3 of Reviewer 1 (see above). To reiterate, in our manuscript we have shown peptide/peptide mimic transport via PfCRT in both directions. For example:

- (1) The ‘H⁺-efflux’ assays measured the transport of 8 modified peptides/peptide mimics via PfCRT in the so-called “right” direction (i.e. the direction in which net transport is expected to occur *in situ*). Using this assay, we showed that PfCRT mediates the proton-dependent efflux of 8 modified peptides/peptide mimics from digestive vacuole into the parasite’s cytosol.
- (2) The experiments in which the peptide mimic saquinavir was used to *trans*-stimulate the efflux of radiolabelled peptide from the oocyte is also an example of transport in the “right” direction. I.e. saquinavir is transported into the oocyte via PfCRT, which is the equivalent of PfCRT transporting saquinavir out of the parasite’s digestive vacuole.
- (3) For a secondary-active carrier such as PfCRT, the net direction of transport is usually determined by the electrochemical gradient of the driving ion(s). Our study shows that peptide transport via PfCRT is dependent on protons and is also coupled to a second solute (the identity of which remains to be discovered). All three co-substrates must be present on the same side of the membrane for peptide

transport to occur. Since the second ion is naturally present within the oocyte cytosol, [³H]VF-6 is co-injected with protons into the oocyte and its efflux measured. This is entirely consistent with the proton-dependent transport of modified peptides/peptide mimics that we measured with the 'H⁺-efflux' assay in live parasites.

Finally, some substrates of PfCRT – e.g. chloroquine, other quinoline-type drugs, the peptide mimic saquinavir, and the endomorphin peptide YPWF-NH₂ – only require protons to undergo translocation. That is, their transport is decoupled from the second ion, thus enabling measurements of their uptake from the extracellular solution into the oocyte (the so-called “right” direction). The decoupling of some substrates, but not others, from the translocation of a co-substrate is again a known phenomenon for carrier-type transporters (e.g. Yanagida et al. *Biochim Biophys Acta* 2001).

Revisions: We have made a number of changes to the manuscript to aid the reader’s understanding of these concepts, which are less well-known outside of the transporter field. These additions have been outlined in detail above (see response to Reviewer 1) and include the following:

- 1) The inclusion of Supplementary Note 2 to summarise the phenomenon of *trans*-stimulation, why it is commonly used to elucidate the substrate-specificity of a transporter, how it was harnessed in this study to discover the native substrates of PfCRT, and why it does not reveal the net direction of substrate transport *in vivo*.
- 2) An explanation for why the activity of a carrier-type transporter can be assessed in either direction (i.e. there is no “right” or “wrong” direction for measuring transport) as well as clarification that we have shown peptide/peptide mimic transport via PfCRT in both directions (Supplementary Note 2).
- 3) An expanded comment on our finding that PfCRT’s translocation of several substrates – e.g. chloroquine, other quinoline-type drugs, the peptide mimic saquinavir, and the endomorphin peptide YPWF-NH₂ – has been decoupled from the second ion (whilst remaining dependent on protons) (Supplementary Note 4).

B.2. Lines 72 – 74, “almost all of the 91 peptides screened strongly *cis*-inhibited [³H]CQ transport by both PfCRTEcu1110 and PfCRTDd2. The test peptides were 2-13 residues in length and most occur in haemoglobin or other erythrocyte proteins.”

Criticism: In the previous publication of Martin et al., of 8 peptides thought to be generated from parasite digestion of Hb, 3 inhibited CQ transport but the other 5 peptides did not (ref #35 Table S5; paragraph A.3 above). Comparison of those peptide sequences with those of the peptides in the present report (Supplementary spreadsheet Tables S1 and S2) shows entirely different results for peptides that were previously shown not to inhibit CQ transport: GLHAL and LSDLHAHK are now listed as strong *cis*-inhibitors, and peptide sub-fragments of the remaining 3 of the 5 non-inhibitory peptides are also listed as *cis*-inhibitors (Strangely, data for these 3 as complete peptides do not seem to be in the spreadsheets). These important inconsistencies, not mentioned at all in the submitted manuscript, need to be addressed and explained.

Ref #35 further showed that most of the inhibitory peptides shared elements of a ‘CQR reverser pharmacophore’ with VP and the quinoline drugs, with specific structural features thought to be recognized by PfCRTCQR. Is this also the case for the 91 peptides in the present submission?

Our response: These comments by Reviewer 2, which are incorrect, appear to have resulted from an incomplete understanding of both the relevant literature and our new manuscript.

Reviewer 2 has overlooked key differences in the methods between this manuscript and the Martin et al. 2009 *Science* paper, resulting in some mistaken conclusions. For example, in this study we lowered the pH of the oocyte reaction buffer to mimic the pH of the digestive vacuole (pH 5.5). The lower pH resulted in a much higher signal-to-background ratio for [³H]-chloroquine uptake via mutant PfCRT (PfCRT^{Ecu1110} and PfCRT^{Dd2}), thereby greatly improving our ability to measure the decreases in transport caused by *cis*-inhibition as well as enabling us to detect the increases in transport caused by *trans*-stimulation

(insufficient sensitivity in the assay had meant the latter simply was not possible when we conducted the Martin et al. 2009 *Science* study). Lowering the pH also increased the *cis*-inhibitory activities of most of the peptides against the mutant isoforms. These changes substantially increased the sensitivity of the oocyte transport assay and thus allowed us to detect *cis*-inhibitory activity for haemoglobin-derived peptides that had been without significant effect against PfCRT^{Dd2} in Martin et al. *Science* 2009.

Furthermore, in claiming that it was strange we had not included three of the peptides tested in Martin *Science* et al. 2009, the Reviewer has overlooked the rationale we applied to select peptides in this study (see lines 400-406 of the original manuscript). The current understanding of where the parasite's enzymes cut the human haemoglobin α and β chains indicates that the three peptides in question would not be generated in the digestive vacuole. These peptides were therefore not tested in the new study.

The Reviewer has likewise missed both the text and graphs that emphasised that many of the peptide substrates of PfCRT are either neutral or negatively-charged at pH 5.5. I.e. they do not possess the characteristics of a "CQR reverser pharmacophore". We also note that only 39 of the >90 peptides tested in our new study were found to be substrates of wild-type PfCRT (PfCRT^{3D7}), and it is thus the properties of these 39 peptide substrates – and not those that only inhibit PfCRT – that should be considered when summarising the substrate-specificity of the transporter. These are very important distinctions given that the observation presented in the Martin et al. *Science* 2009 study was based on a much smaller sample size (only 10 peptides inhibited PfCRT^{Dd2}) and the transport of just a single peptide was tested ([³H]YPWF-NH₂, and it was not a substrate of PfCRT^{3D7}).

Of the 39 peptides we have shown to be substrates of PfCRT^{3D7}, only a single peptide contained at least one aromatic residue and is positively charged at pH 5.5. Of the remaining peptide substrates, 22 are negatively charged or neutral at pH 5.5, 14 contain one or more aromatic residues but are not positively charged at pH 5.5, and 2 are positively charged but do not contain an aromatic residue. These results are not unexpected given that PfCRT^{3D7} does not possess the mutations that enable the transporter to translocate chloroquine (which is positively-charged and aromatic) and there is, therefore, no basis for assuming that the wild-type protein would prefer compounds containing a "CQR reverser pharmacophore". PfCRT^{Dd2} also does not show a strong preference for peptides that contain at least one aromatic residue and are positively charged. Moreover, relative to PfCRT^{3D7}, PfCRT^{Dd2} lost the ability to transport 26% of the negatively-charged peptides, 20% of the positively-charged peptides, and 30% of the neutral peptides. These fairly uniform reductions in substrate specificity are not consistent with the idea that PfCRT^{Dd2} has a preference for peptides that share the features of the "CQR reverser pharmacophore".

Revisions: None required.

B.3. Discussion and Abstract, lines 17 – 18 and 251 – 252: “[We] show that the protein does not mediate the non-specific transport of ions and other metabolites” and “Contrary to previous reports, PfCRT is not a non-specific transporter of metabolites and/or of organic or inorganic ions.”

Criticism: Research studies and publications from several different laboratories have presented evidence for PfCRT activity as a proton, chloride channel, a general cation channel, a transporter of glutathione, an iron atom transporter, a metabolite transporter ... (lines 53 – 56). These possibilities are not examined in the present study and evidence is not presented against those reports. The statements in lines 17 – 18 and 251 – 252 are therefore unsupportable.

Our response: These claims are incorrect. We screened 43 metabolites and ions – including arginine, glutathione, and iron – for the ability to be transported via PfCRT using the *trans*-stimulation assay (Fig. 2 and 5d and Supplementary Tables 1,2). None were substrates. We confirmed these results by undertaking measurements with radiolabelled versions of 11 of these solutes (Fig. 3k and Supplementary Fig. 3i). The outcome was the same whether the radiolabelled solutes were added to the extracellular buffer or injected into the oocytes; none of these solutes were substrates of PfCRT. We have extended this work to include a

demonstration that the removal of chloride has no effect on the transport of [³H]VF-6 or [³H]chloroquine via PfCRT. These findings provide further evidence that PfCRT functions independently of chloride; i.e. chloride is not a substrate of PfCRT.

Revisions: The effect of chloride on PfCRT-mediated transport was assessed for both [³H]VF-6 efflux and [³H]chloroquine influx by replacing chloride with gluconate. The resulting datasets are shown in Supplementary Fig. 2c,d of the revised manuscript and the relevant section of the Results now reads:

“The efflux of [³H]VF-6 from PfCRT-expressing oocytes was linear over time for at least 2 hours and was dependent on both the pH of the injection buffer and the oocyte membrane potential (see also Supplementary Fig. 2 and Supplementary Note 4). By contrast, removal of Na⁺ or Cl⁻ from the injection buffer, or varying the Ca²⁺ concentration, were all without effect (Supplementary Fig. 2).”

B.4. Lines 195 – 196: “Verapamil and Q2C inhibited the peptide-induced H⁺ leaks in C2GC03, C67G8, and C4Dd2 parasites (Fig. 5b), providing further evidence that these solutes are effluxed from the DV via PfCRT.”

Criticism: To the contrary, Hrycyna and Martin et al. concluded that Q2C (compound 6) is not a substrate of PfCRT and is “not effluxed by the protein” (submission ref #17, pg. 728). This glaring discrepancy needs to be clarified and addressed.

Regarding verapamil, Summers et al. [Cell Mol Life Sci. 2012 Jun;69(12):1967-95] state on page 1975 that “direct measurements of transport are required to confirm that PfCRT^{Q2C} possesses the ability to translocate VP (or any other resistance-reverser).” If direct measurements have since then demonstrated PfCRT transport of verapamil, please provide the information and a citation.

The caveats taken from Hrycyna et al. and Summers et al. apply similarly to the authors’ present claim/assumption that evidence of cis-inhibition (Fig. 1) or trans-stimulation (Fig. 2) demonstrates peptide transport via PfCRT.

Our response: These claims are incorrect. We did not indicate that Q₂C is a substrate of PfCRT, we instead stated that it inhibited the PfCRT-mediated efflux of modified peptides and peptide mimics from the digestive vacuole. Indeed, we used Q₂C in these assays precisely because it is a potent inhibitor, but not a substrate, of PfCRT^{Dd2} (Hrycyna et al. *ACS Chemical Biology* 2014). Hence, there is no discrepancy between this manuscript and Hrycyna et al. *ACS Chemical Biology* 2014.

The review we published 8 years ago (Summers, Nash, & Martin *Cell Mol Life Sci* 2012) will of course contain statements that have been superseded by new studies. Refer to Bellanca et al. *JBC* 2014 (Fig. 10) for direct measurements of the transport of [³H]verapamil via PfCRT^{Dd2}. Verapamil is both a substrate (albeit a relatively poor one) and a good inhibitor of PfCRT^{Dd2}. At the high concentrations used in the H⁺-efflux assay (i.e. 50 μM), verapamil inhibits mutant PfCRT and is not itself transported – most likely because the binding of two or more verapamil molecules to the substrate-binding cavity blocks the transporter (as is already described in Bellanca et al. *JBC* 2014).

There are, therefore, no “caveats” to be taken from Hrycyna et al. 2014 or Summers, Nash, & Martin 2012. Furthermore, nowhere in the manuscript did we claim that cis-inhibition is evidence of transport. To the contrary, we stated that “*The cis-inhibition assay identifies compounds that interact with PfCRT, but does not distinguish between solutes that inhibit (and are not translocated) and those that are substrates of PfCRT*”. Moreover, Reviewer 2 is incorrect in his/her belief that a trans-stimulation assay cannot be used to identify the substrates of a transporter. This assay is a long-established method for delineating the substrate-specificity of carrier-type transporters (Luteijn et al. *Nature* 2019; Enomoto et al. *Nature* 2002; Reig et al. *Embo j* 2002; Geier et al. *Proc Natl Acad Sci U S A* 2013). It also has the significant advantage of not requiring the prohibitive cost of purchasing multiple radiolabelled solutes.

Revisions: We have reworded the relevant sentences in the Results section to further emphasise that it

was the transport of the peptide mimics and modified peptides (and not the transport of the inhibitors Q₂C and verapamil) that was measured with the H⁺-efflux assay in live parasites. The revised text now reads:

“Verapamil and Q₂C inhibited the peptide-induced H⁺ leaks in C2^{GC03}, C6^{7G8}, and C4^{Dd2} parasites (Fig. 5b), providing further evidence that the peptide mimics and modified peptides are exported from the DV via PfCRT. Moreover, as observed in the oocyte system, Q₂C was the most potent inhibitor, and both Q₂C and verapamil were more effective against the mutant transporters than against PfCRT^{3D7}.”

C. Evaluation, additional criticisms

C.1. The LC-MS/MS studies of Lewis et al. (submission ref #23) identified 362 endogenous peptides ranging from 3-mers to 32-mers that corresponded exactly to sequences in the Hb α or β chains, and quantitative analyses found 87 peptides that showed evidence for differential accumulation between isogenic *P. falciparum* lines that differed only by the PfCRT^{CQS} or PfCRT^{QQR} allele (paragraph A.2 above). In light of those findings, why were results and analyses of the present submission limited to peptides of 4 – 11 amino acids?

Our response: This claim is incorrect. We did not limit our study to peptides of 4-11 residues. We tested peptides 2-13 residues in length. Those containing 4-11 residues were found to be substrates of PfCRT, with the transporter showing a preference for peptides 5-6 residues in length.

Revisions: We have revised Supplementary Fig. 1c to further emphasise the range of peptides tested in this study. This figure illustrates PfCRT's preference for transporting peptides 5-6 residues in length. To further define the outer limits of PfCRT's specificity for peptide substrates, we have tested an additional four peptides that are 16, 20, or 25 residues in length (DA-16, NH-20, FK-20, and LA-25). Whilst all four of these peptides were capable of *cis*-inhibiting [³H]chloroquine transport via PfCRT^{Ecu1110} and PfCRT^{Dd2}, none *trans*-stimulated [³H]chloroquine transport via PfCRT (the new data have been included in Supplementary Fig. 1c and Supplementary Tables 1,2). The substrate-specificity of PfCRT therefore remains as peptides between 4-11 residues in length.

C.2. Considering the very different results from transport experiments with [3H]-labelled peptides YPWF-NH₂ and VF-6, have experiments with other [3H]-labelled peptides been attempted to directly demonstrate their influx and/or efflux by PfCRT^{CQS}- and PfCRT^{QQR}-transformed *Xenopus laevis* oocytes? If so, with what results? If not, why haven't experiments to directly demonstrate the transport of additional [3H]-labelled peptides been a priority, to test the authors' hypothesis for PfCRT's natural function?

Our response: Reviewer 2 has conflated the present study with one of our previous publications (Martin et al. *Science* 2009) and has overlooked key methodological differences between the two studies.

For example, the experiments with the opioid receptor agonist 'endomorphin-1' YPWF-NH₂ were not performed at pH 5.5, which is the pH used in the present study because it improves the transport signal obtained for PfCRT. Nor were the YPWF-NH₂ experiments undertaken with the assay we now know is optimal, and in most cases necessary, for detecting the transport of peptides/peptide mimics via PfCRT. Hence, it is not surprising that YPWF-NH₂ transport was detected for only one of the PfCRT variants tested in the 2009 paper and, as such, the older work does not contradict the present study, nor does it necessitate the substantial expense of purchasing and characterising additional radiolabelled peptides. Furthermore, undertaking measurements of [³H]YPWF-NH₂ transport with the new, optimised assays would be of little benefit or relevance given that YPWF-NH₂ is not present in human haemoglobin.

In any case, the datasets presented in the current manuscript already provide extensive and detailed evidence of the transport of a number of different peptides via PfCRT (a point that has already been addressed above). Moreover, this view is shared and supported by both Reviewers 1 and 3 in their

respective reports.

Revisions: None required.

C.3. In Fig. 6a, peptide trans-stimulation results from PfCRT3D7-transformed (PfCRTCQS-transformed) *Xenopus laevis* oocytes are compared to relative peptide accumulation results from C4Dd2 (CQR) parasites. Why weren't peptide trans-stimulation results from corresponding PfCRTDd2-transformed oocytes used instead for this comparison? This would be a much better and possibly conclusion-changing comparison, especially considering that, of a set of 89 peptides, 39 peptides trans-stimulated [3H]CQ transport via PfCRT3D7 but only 23 trans-stimulated [3H]CQ transport via PfCRTDd2 (line 86 – 88).

Our response: Reviewer 2 has misinterpreted the objective of the analysis presented in Fig. 6a. The datasets we generated with the *Xenopus* oocyte system revealed that wild-type PfCRT (PfCRT^{3D7}) can transport a broad range of host-derived peptides from the parasite's digestive vacuole, and that the mutant transporters (PfCRT^{Ecu1110} and PfCRT^{Dd2}) have reduced capacities for peptide transport, both in terms of maximum velocity of transport and the range of peptides translocated. Given that a reduction in the efflux of peptides via PfCRT^{Dd2} could explain the accumulation of host-derived peptides within the C4^{Dd2} parasite line, we sought to determine if there was a correlation between the host-derived peptides that had been identified as substrates of PfCRT^{3D7} in the *Xenopus* oocyte system and those found to accumulate within the C4^{Dd2} line. It was important to use the PfCRT^{3D7} *trans*-stimulation dataset, rather than that generated for PfCRT^{Dd2}, because the peptides most likely to accumulate in the C4^{Dd2} parasite lines will include those that are very poor substrates of (or no longer transported by) PfCRT^{Dd2}. These data-points, which represent a significant portion of the *trans*-stimulation dataset, will be lost to the analysis unless the PfCRT^{3D7} dataset is correlated with the levels of the respective peptides in the C4^{Dd2} parasite line. This comparison has enabled us to identify a positive relationship between the ability of a given host-derived peptide to serve as a substrate of wild-type PfCRT and an increase in its accumulation within the C4^{Dd2} line (consistent with the reduced capacity of PfCRT^{Dd2} for effluxing said peptides out of the digestive vacuole).

Revisions: None required.

C.4. The *trans*-stimulation results presented in Fig. 2 and described in lines 79 – 92 were performed at much higher mM peptide concentrations than those used for the *cis*-inhibition studies. Some discussion of these much higher concentrations, and of their relatively small effects on [3H]CQ accumulation in the *trans*-stimulation experiments, would be helpful.

Our response: A *cis*-inhibition assay measures the ability of an unlabeled solute to compete with the binding of a radiolabeled substrate to the transporter. Inhibition is detected as a decrease in the rate of radiolabel transport relative to the positive control (which is the transport of the radiolabeled substrate via the carrier in the absence of an unlabeled test solute). The high signal-to-background ratio for [3H]chloroquine uptake in the PfCRT oocyte system, as well as the reproducibility of the assay, enables relatively modest decreases in the transport rate to be quantified. This, together with the observation that the IC₅₀ values for the inhibition of [3H]chloroquine transport by known substrates of PfCRT^{Dd2} (e.g. quinine, methylene blue, quinacrine, verapamil, and amantadine) range from 13 μM to 1.3 mM (Bellanca et al. *J Biol Chem* 2014; Richards et al. *PLoS Pathog* 2016; van Schalkwyk et al. *J Infect Dis* 2016), supported testing the unlabeled solutes for *cis*-inhibition at 0.5 – 2 mM.

By contrast, *trans*-stimulation measures an increase in the transport rate above that measured in the 'zero-trans' positive control. The extent to which a carrier can be *trans*-stimulated above its 'zero-trans' rate of transport will depend on factors that are inherent to its structure-function, such as the activation energy (E_a) for substrate translocation (Stein 1986). For some transporters, *trans*-stimulation increases the transport rate by several fold, whereas other carriers exhibit more modest increases – for example, the

trans-stimulated rate may only be 20-50% above the 'zero-*trans*' rate. Hence, *trans*-stimulation can be more difficult to quantify than *cis*-inhibition, especially if the unlabelled test solute is a poor substrate and/or the re-orientation of the empty carrier is not a substantially rate-limiting step in the translocation cycle. We therefore used high concentrations of the test solutes in the *trans*-stimulation assays (5 mM and 35/45 mM) to optimise the detection of increases in the rate of [³H]chloroquine transport.

Revisions: These are very technical details that are unlikely to be of interest to most readers and are also unnecessary to understand the key findings of the manuscript. We have, nonetheless, expanded Supplementary Notes 1 and 2 to include brief summaries of the above information. The new text is as follows:

Supplementary Note 1:

"The *cis*-inhibition assay measures the ability of an unlabeled solute to compete with the binding of [³H]CQ to PfCRT^{Ecu1110} and PfCRT^{Dd2}. Inhibition is detected as a decrease in the rate of [³H]CQ transport relative to the positive control (which is [³H]CQ transport via PfCRT^{Ecu1110} or PfCRT^{Dd2} in the absence of an unlabeled test solute). The high signal-to-background ratio for CQ uptake in the PfCRT oocyte system, as well as the sensitivity of the assay, enables relatively modest decreases in the [³H]CQ transport rate to be quantified. This, together with the observation that the IC₅₀ values for the inhibition of [³H]CQ transport by known substrates of PfCRT^{Dd2} (e.g. quinine, methylene blue, quinacrine, verapamil, and amantadine) range from 13 μM to 1.3 mM⁶⁻⁸, supported testing the unlabeled solutes for *cis*-inhibition at 0.5 – 2 mM."

Supplementary Note 2:

"*Trans*-stimulation has occurred if there is an increase in the transport rate above that measured in the relevant positive control. The extent to which a carrier can be *trans*-stimulated above its 'zero-*trans*' rate of transport will depend on factors that are inherent to its structure-function, such as the activation energy (E_a) for substrate translocation¹⁶. For some transporters, *trans*-stimulation increases the transport rate by several fold, whereas other carriers exhibit more modest increases – for example, the *trans*-stimulated rate may only be 20-50% above the 'zero-*trans*' rate. Hence, *trans*-stimulation can be more difficult to quantify than *cis*-inhibition, especially if the unlabelled test solute is a poor substrate and/or the re-orientation of the empty carrier is not a substantially rate-limiting step in the translocation cycle. We therefore used high concentrations of the test solutes in the *trans*-stimulation assays (5 mM and 35/45 mM) to optimise the detection of increases in the rate of [³H]CQ transport."

C.5. Regarding lines 68 – 69: "... detect unlabelled solutes that inhibit [3H]CQ uptake, consistent with them interacting with the mutant transporter's substrate-binding cavity (Fig. 1a)".

In a previous study Bellanca et al. (submission ref #36) found that PfCRT contains multiple substrate-binding sites, and that compounds can compete for transport via PfCRT by mixed-type inhibition. Could mixed-type inhibition be involved in the outcomes shown in Fig. 1 of the present submission? This possibility needs to be addressed.

Our response: The reviewer has conflated the aim of this manuscript (to identify the native substrates of PfCRT) with the goal of our Bellanca et al. J Biol Chem 2014 study, which was to understand the specific nature of the *cis*-inhibition of chloroquine transport via PfCRT^{Dd2} by (1) quinine and (2) verapamil.

Both quinine and verapamil are substrates of PfCRT^{Dd2} (but not of PfCRT^{3D7}) and both *cis*-inhibit [³H]chloroquine transport via PfCRT^{Dd2} (Bellanca et al. J Biol Chem 2014; Hrycyna et al. ACS Chem Biol 2014). In the Bellanca et al. study, we used detailed inhibition kinetics to identify the particular type of *cis*-inhibition – e.g. pure competitive, mixed-type, or partial mixed-type – exerted by these two drugs. However, such analyses are not relevant here. Instead, in this manuscript we used the *cis*-inhibition assay – which screened solutes for the ability to *cis*-inhibit [³H]chloroquine transport via PfCRT^{Dd2} and PfCRT^{Ecu1110} – as the first step in our strategy to identify the natural substrates of PfCRT. Whether a solute caused *cis*-

inhibition via pure competitive or mixed-type inhibition is of no consequence to the conclusions. Moreover, by including all of the test solutes in the second step of the strategy (the *trans*-stimulation assay) we allowed for the unlikely scenario that a test solute causes little or no *cis*-inhibition of PfCRT-mediated chloroquine transport, but is nonetheless a substrate of PfCRT. That is, if a solute does not use the same points of attachment as chloroquine inside the binding cavity, and is also translocated at an appreciable rate, it could be a relatively poor inhibitor of [³H]chloroquine transport. However, such a solute would still *trans*-stimulate [³H]chloroquine transport via PfCRT; hence the inclusion of all test solutes in both the *cis*-inhibition and *trans*-stimulation assays.

Revisions: None required.

C.6. Line 213: "Together, these datasets provide a robust demonstration of peptide transport via PfCRT *in situ*."

For reasons detailed above, this reviewer does not agree with this statement that the datasets provide a robust demonstration of peptide transport via PfCRT *in situ* – not at all.

Our response: We believe that many of the comments provided by Reviewer 2 have arisen from misunderstandings of the manuscript, the methods, and/or the datasets as well as an incomplete understanding of several of our previous papers. We hope that the changes we have made to the text and figures, including the addition of extensive explanations in the Supplementary Notes, have clarified these points. We also undertook new experiments to address some of Reviewer 2's queries, and the resulting datasets have been included in the revised manuscript.

Finally, we note that Reviewers 1 & 3 are very much of the view that our manuscript has provided robust demonstrations, both *in vitro* and *in situ*, of peptide transport via PfCRT. For example:

Reviewer 1: "*These data provide compelling evidence that PfCRT is able to transport a range of peptides, preferably 4-11 amino acids in length and derived mostly from degraded host haemoglobin. The authors focus on two peptides (termed VF-6 and HM-5) to show that they can serve as transported substrates and also inhibit the transport of tritiated CQ. Observations in Xenopus oocytes are complemented by studies with transgenic parasite lines that confirm peptide accumulation in the resistant lines and that show this to be H⁺-dependent, as measured using concanamycin A.*"

Reviewer 1: "*this present manuscript presents detailed and extensive evidence showing a wide range of peptides transported by PfCRT and extends this to show differences between the WT and variant isoforms and how they impact drug transport. The work is comprehensive and a notable contribution to our understanding of PfCRT function.*"

Reviewer 3: "*The data represent a heroic amount of work which provides important insights into the role of PfCRT as a peptide transporter, a very important protein in the malaria field.*"

Reviewer #3

Reviewer #3 (Remarks to the Author):

Shafit et al. "The natural function of the malaria parasite's chloroquine resistance 1 transporter"

I would like to congratulate the authors for the very well written manuscript. The data represent a heroic amount of work which provides important insights into the role of PfCRT as a peptide transporter, a very important protein in the malaria field.

Technically, the experiments were well thought of. The data appear to be internally consistent throughout. While I expect that the experiments would be fairly sophisticated for general readers who are not familiar with membrane transport, I think that the authors have done a good job explaining the 'gist' of each experiment.

This manuscript is one of the very few that I do not have things to criticize. I fully recommend the journal to publish this work.

Our response: We thank the Reviewer for assessing our manuscript and for their appreciation of the extent and quality of the datasets, and the significance of our findings.

Revisions: None required.

References

- Martin, R. E., et al. Chloroquine transport via the malaria parasite's chloroquine resistance transporter. *Science*. **325**, 1680-1682 (2009).
- Summers, R. L., et al. Diverse mutational pathways converge on saturable chloroquine transport via the malaria parasite's chloroquine resistance transporter. *Proc Natl Acad Sci U S A*. **111**, E1759-1767 (2014).
- Lehane, A. M., Hayward, R., Saliba, K. J., & Kirk, K. A verapamil-sensitive chloroquine-associated H⁺ leak from the digestive vacuole in chloroquine-resistant malaria parasites. *J Cell Sci*. **121**, 1624-1632 (2008).
- Lehane, A. M., & Kirk, K. Chloroquine resistance-conferring mutations in *pfcr*t give rise to a chloroquine-associated H⁺ leak from the malaria parasite's digestive vacuole. *Antimicrob Agents Chemother*. **52**, 4374-4380 (2008).
- Lehane, A. M., & Kirk, K. Efflux of a range of antimalarial drugs and 'chloroquine resistance reversers' from the digestive vacuole in malaria parasites with mutant PfCRT. *Mol Microbiol*. **77**, 1039-1051 (2010).
- Bellanca, S., et al. Multiple drugs compete for transport via the *Plasmodium falciparum* chloroquine resistance transporter at distinct but interdependent sites. *J Biol Chem*. **289**, 36336-36351 (2014).
- Richards, S. N., et al. Molecular mechanisms for drug hypersensitivity induced by the malaria parasite's chloroquine resistance transporter. *PLoS Pathog*. **12**, e1005725 (2016).
- van Schalkwyk, D. A., et al. Verapamil-sensitive transport of quinacrine and methylene blue via the *Plasmodium falciparum* chloroquine resistance transporter reduces the parasite's susceptibility to these tricyclic drugs. *J Infect Dis*. **213**, 800-810 (2016).
- Hrycyna, C. A., et al. Quinine dimers are potent inhibitors of the *Plasmodium falciparum* chloroquine resistance transporter and are active against quinoline-resistant *P. falciparum*. *ACS Chem Biol*. **9**, 722-730 (2014).
- Kumar, S., & Bandyopadhyay, U. Free heme toxicity and its detoxification systems in human. *Toxicol Lett*. **157**, 175-188 (2005).
- Saliba, K. J., Horner, H. A., & Kirk, K. Transport and metabolism of the essential vitamin pantothenic acid in human erythrocytes infected with the malaria parasite *Plasmodium falciparum*. *J Biol Chem*. **273**, 10190-10195 (1998).
- Pulcini, S., et al. Mutations in the *Plasmodium falciparum* chloroquine resistance transporter, PfCRT, enlarge the parasite's food vacuole and alter drug sensitivities. *Sci Rep*. **5**, 14552-14567 (2015).
- Kim, J., et al. Structure and drug resistance of the *Plasmodium falciparum* transporter PfCRT. *Nature*. **576**, 315-320 (2019).
- Wang, Z., et al. Genome-wide association analysis identifies genetic loci associated with resistance to multiple antimalarials in *Plasmodium falciparum* from China-Myanmar border. *Sci. Rep*. **6**, 33891 (2016).
- Cowell, A. N., et al. Mapping the malaria parasite druggable genome by using *in vitro* evolution and chemogenomics. *Science*. **359**, 191-199 (2018).

- Kinga Modrzynska, K., et al. Quantitative genome re-sequencing defines multiple mutations conferring chloroquine resistance in rodent malaria. *BMC Genomics*. **13**, 106 (2012).
- Summers, R. L., Nash, M. N., & Martin, R. E. Know your enemy: understanding the role of PfCRT in drug resistance could lead to new antimalarial tactics. *Cell Mol Life Sci*. **69**, 1967-1995 (2012).
- Martin, R. E., & Kirk, K. Transport of the essential nutrient isoleucine in human erythrocytes infected with the malaria parasite *Plasmodium falciparum*. *Blood*. **109**, 2217-2224 (2007).
- Marchetti, R. V., et al. A lactate and formate transporter in the intraerythrocytic malaria parasite, *Plasmodium falciparum*. *Nat Commun*. **6**, 6721 (2015).
- Hapuarachchi, S. V., et al. The malaria parasite's lactate transporter PfFNT is the target of antiplasmodial compounds identified in whole cell phenotypic screens. *PLoS pathogens*. **13**, e1006180-e1006180 (2017).
- Elliott, J. L., Saliba, K. J., & Kirk, K. Transport of lactate and pyruvate in the intraerythrocytic malaria parasite, *Plasmodium falciparum*. *Biochem J*. **355**, 733-739 (2001).
- Yanagida, O., et al. Human L-type amino acid transporter 1 (LAT1): characterization of function and expression in tumor cell lines. *Biochim Biophys Acta*. **1514**, 291-302 (2001).
- Luteijn, R. D., et al. SLC19A1 transports immunoreactive cyclic dinucleotides. *Nature*. **573**, 434-438 (2019).
- Enomoto, A., et al. Molecular identification of a renal urate anion exchanger that regulates blood urate levels. *Nature*. **417**, 447-452 (2002).
- Reig, N., et al. The light subunit of system b(o,+) is fully functional in the absence of the heavy subunit. *Embo j*. **21**, 4906-4914 (2002).
- Geier, E. G., et al. Structure-based ligand discovery for the Large-neutral Amino Acid Transporter 1, LAT-1. *Proc Natl Acad Sci U S A*. **110**, 5480-5485 (2013).
- Stein, W. 1986. *Transport and diffusion across cell membranes*, New York: Academic Press.

Reviewers' Comments:

Reviewer #1:

Remarks to the Author:

I commend the senior author and her team for an outstanding revision and rebuttal. As reviewer 1, I had many questions and the rebuttal was very instructive in clarifying many of these queries. The response included the additional of multiple Supplementary Notes to help readers like myself who want to understand the science and underlying transport principles, without being experts in this area of transporter physiology. The response to reviewer 2 was in my opinion also grounded in well-constructed arguments. Reviewer 3 was very positive from the outset and had not requested any changes.

Of the 21 pages of rebuttal, I could find no area where additional clarification or revised interpretation was required. As evidence of its outstanding quality, I found one solitary typo ("stragey" on page 19).

Overall, I find the science exciting and exceptionally well presented. The data are an important advance in understanding the native function of PfCRT, the major determinant of Plasmodium falciparum resistance to heme-binding antimalarials of major clinical importance including chloroquine and more recently piperaquine.

Reviewer #2:

Remarks to the Author:

NCOMMS-19-33349A-Z

Evaluation of responses to reviewer comments (Reviewer #2)

(The Comment numbers listed here are taken from the review of NCOMMS-19-33349-T)

Comment B.1. The third point the reply, response B.1.3, answers the criticism with a new hypothesis that is entered into Supplementary Information as a statement of fact, namely: "peptide transport via PfCRT is dependent both on protons and a second solute that remains to be identified, but which is naturally present within the oocyte. All three co-substrates must be present on the same side of the membrane for the transport of host-derived peptides to occur."

This second solute hypothesis is used to explain why it has not been possible to demonstrate the transport of peptides other than the endorphin peptide YPWF-NH₂ from the outside of PfCRT-transformed oocytes, where the second solute may not be available. If the hypothesized second solute can be discovered, will transport of a full set of hemoglobin-derived peptides from outside the PfCRT-transformed oocyte then become demonstrable? Such demonstration might then provide direct support for the claim in the manuscript that "datasets we have obtained using a set of complementary in vitro and in situ assays reveal that PfCRT exports host-derived peptides containing 4-11 residues out of the parasite's DV".

In the meantime, transport from the outside of the PfCRT-transformed oocyte of endorphin peptide YPWF-NH₂, peptide mimic saquinavir, chloroquine and other quinoline drugs is explained by no requirement for (decoupling of) the unknown second solute. In questionable support of this decoupling hypothesis, reference is made to a 2001 Biochim Biophys Acta paper on the Human L-type Amino Acid Transporter 1 in Tumor Cell Lines (Yanagida et al.).

If the hypothesis of a second solute is correct, and is not a deus ex machina for the authors' model in face of no directly demonstrated transport of hemoglobin-derived peptides from outside the PfCRT-transformed oocytes, what might the second solute be? This is an important question to address, one worth bringing with the hypothesis into the maintext and perhaps the Abstract. In response to

Reviewer #1, the hypothesized second solute is suggested to be a negatively-charged ion that may help to account for some membrane potential findings, but this evidence is not strong. Responses B.1.1. and B.1.2 provide tangential answers and don't really meet the criticism of comment B.1. Regarding B.1.1: the 'H⁺-assays' are in live parasites, which is obviously a switch from PfCRT-transformed oocytes and brings a whole different system of cellular pathways and experimental measures into the picture. Regarding B.1.2: saquinavir, the peptide mimic, is of course not a hemoglobin peptide and is hypothesized (as with the endorphin peptide) to be decoupled from the unknown second solute.

Comment B.2 quoted from the submission and asked that inconsistencies with previously published peptide cis-inhibitory activities and previous claims about shared elements of the 'CQR reverser pharmacophore' in inhibitory peptides (Martin RE, et al. Science. 2009. PMID: 19779197) be addressed. The response explains that a buffer improvement was incorporated into the present study and that detection of cis-inhibitory activity are now sensitive enough to detect activities of peptides that were previous thought to be without effect. The authors also appear to be abandoning some previous ideas about the 'CQR reverser pharmacophore'. These are acceptable shifts of findings and hypothesis; but the reply to B.2 "Revisions: none required" is not acceptable. The inconsistencies with previously published peptide cis-inhibitory activities and previous claims about shared elements of the 'CQR reverser pharmacophore' in inhibitory peptides need to be addressed and explained in the manuscript. (If this has been done, I apologize for missing it; if not done, a few short sentences will do).

Comment B.3 Publications presenting evidence for PfCRT activity as a chloride channel were primarily behind this comment. The authors have now performed experiments that tested for an effect of chloride on [3H]-VF-6 efflux or [3H]-chloroquine influx by PfCRT after replacement of chloride with gluconate. Removal of chloride had no effect. The addition of these findings to the manuscript is appreciated.

Comment B.4 "Verapamil and Q2C inhibited the peptide-induced H⁺ leaks in C2GC03, C67G8, and C4Dd2 parasites (Fig. 5b), providing further evidence that these solutes are effluxed from the DV via PfCRT." The reply states that "the solutes" in the sentence was not meant to refer to Verapamil and Q2C, which was an implication in obvious discrepancy with previous publication of Hrycyna et al. "The solutes" was meant instead to refer to peptide mimics and modified peptides. In the revision, the corrected sentence now reads "Verapamil and Q2C inhibited the peptide-induced H⁺ leaks in C2GC03, C67G8, and C4Dd2 parasites (Fig. 5b), providing further evidence that the peptide mimics and modified peptides are effluxed from the DV via PfCRT." This revision takes care of the matter.

Comment C.1 The new data on an additional four peptides that are 16 – 25 residues in length is appreciated, as are the findings that these peptides cis-inhibited [3H]-chloroquine transport via PfCRTEcu1110 and PfCRTDd2 but did not trans-stimulate [3H]-chloroquine transport via PfCRT.

Comment C.2 This reply to this comment relates to reply B.2 above. Different results resulting from methodological changes are of course acceptable, as are new hypotheses. But the reply "Revisions: none required" is not acceptable. The differences attributed to pH change in the authors' response need to be addressed and explained in the manuscript. (If this has been done, I apologize for missing it; if not done, a few short sentences will do).

Comment C.3 This reply provides a complex answer, especially in view of the new hypothesis that peptide transport via PfCRT is dependent on a second solute, the identity of which remains to be discovered. A system of a multitude of different peptide specificities and transport velocities from the cis side (requiring the undiscovered second solute), with many substrate specificities potentially

affected or fully compromised by mutations of PfCRT, is indeed a knotty one to study and clarify by trans-stimulation experiments. My recommendation here is to include sentences/text from the authors' response here into a paragraph of supplementary information.

Comment C.4 The additional information, new text and supplementary notes are fine.

Comment C.5 The kinetics in question might be relevant also to the authors' new hypothesis that peptide transport via PfCRT is coupled to a second, undiscovered solute, and that "three co-substrates must be present on the same side of the membrane for peptide transport to occur" (Response B.1.3).

Comment C.6 The manuscript is substantially improved with the revisions in response to the comments above. However, "robust demonstration of peptide transport via PfCRT" remains to be provided from the transformed oocyte system. Will this be possible if evidence for the "second, undiscovered solute: three co-substrates required on the same side of the membrane for peptide transport to occur" hypothesis can be developed and make a breakthrough? As mentioned above in discussion of response B.1.3, the manuscript would do well to bring this hypothesis forward more prominently and perhaps feature it in the Abstract.

reviewed by T.E. Wellems

Reviewer #4:

Remarks to the Author:

PfCRT is known as key modulator of multidrug resistance in *P. falciparum*, especially in respect to chloroquine resistance, while the natural substrates of the transporter remain unidentified. Sarah Shafik and colleagues focused on these transporter substrates and systematically tested a library of solute compounds for its cis-inhibitory activity on [³H]CQ transport. In their manuscript NCOMMS-19-33349A-Z entitled „The natural function of the malaria parasite's chloroquine resistance transporter" the authors distinguished between inhibitors and substrates and pointed out that peptides with 4-11 aa are substrates of PfCRT, as they trans-stimulated the transporter. These data were strengthened by numerous functional and electro-physiological assays with e.g. haemoglobin-derived VF-6 and HM-5 (concentration dependent trans-stimulation), peptide mimetics as trans-stimulator and direct transport of labelled VF-6. The characteristics of the mutant pFCRT isoforms clearly differed from the wildtype form (good overview in Fig. 2f and 2g), and the data explained previous observations in respect to CQ resistance and *P. falciparum* fitness in vivo. The oocyte experiments are here confirmed by analysis in transgenic *P. falciparum*. I am impressed by the multitude of assays and methods performed in the present study. The manuscript is well structured and written, the figures are coherent and well explained in its legends. The work convinces and, for the first time, gives clear answers on the natural function of PfCRT in Plasmodium. I recommend publication of the present study in Nature Communications after editing of the following point.

The dependence of the transport on the proton-gradient via the vacuole membrane is clearly demonstrated, however, I am not convinced about the role of the co-substrate. Why should it be necessary for peptide transport but not for saquinavir and CQ? The extreme variance in length and aa sequence for the accepted substrates speaks for a high flexibility in the 3D structure of the PfCRT pore. Does the transporter use various mechanisms for inward-directed and outward-directed transport? As PfCRT belongs to the carrier-type transporter, it can transport in both directions and I doubt that it uses various modifications of its pore. Please comment in more detail on these findings.

Reviewer #1

Reviewer #1 (Remarks to the Author):

I commend the senior author and her team for an outstanding revision and rebuttal. As reviewer 1, I had many questions and the rebuttal was very instructive in clarifying many of these queries. The response included the additional of multiple Supplementary Notes to help readers like myself who want to understand the science and underlying transport principles, without being experts in this area of transporter physiology. The response to reviewer 2 was in my opinion also grounded in well-constructed arguments. Reviewer 3 was very positive from the outset and had not requested any changes.

Of the 21 pages of rebuttal, I could find no area where additional clarification or revised interpretation was required. As evidence of its outstanding quality, I found one solitary typo ("stragey" on page 19).

Overall, I find the science exciting and exceptionally well presented. The data are an important advance in understanding the native function of PfCRT, the major determinant of Plasmodium falciparum resistance to heme-binding antimalarials of major clinical importance including chloroquine and more recently piperaquine.

Our response: We thank the Reviewer again for his/her time and effort in assessing our work and for the positive appraisal of the revised manuscript. We agree that the changes made to address his/her comments have strengthened the manuscript as well as greatly improving its accessibility to non-specialist readers.

Revisions: None required.

Reviewer #2

Reviewer #2 (Remarks to the Author):

Evaluation of responses to reviewer comments
(The Comment numbers listed here are taken from the review of NCOMMS-19-33349-T)

Our response: We thank Reviewer 2 for considering our revised submission and the Response to the Reviewers document. Whilst Reviewer 2 is satisfied with most of our responses to his comments, he has raised a few additional points and has suggested further text be added to the manuscript to address these concerns. We have expanded the text to accommodate these requests where relevant, and have also replaced the reference we had used as an example of a carrier that requires a co-substrate for the translocation of some substrates and yet transports others without this co-substrate. The original paper we cited referred to the L-type amino acid transporters, which do not display this unusual property. We have corrected this oversight by citing papers on the intended example – the y^L amino acid transporters. These carriers mediate the Na⁺-dependent transport of neutral amino acids and the Na⁺-independent transport of cationic amino acids. We have also added the Nramp transporter as another example; Nramp translocates some substrates with H⁺ and others without this ion.

It should be noted that several of Reviewer 2's new comments have arisen from his assertion that we have inserted a "new hypothesis" into the revised manuscript and that this hypothesis was mentioned only in the Supplementary Information. This claim is incorrect. Our finding that the transport of peptides via PfCRT is dependent on both protons and a second (unidentified) co-substrate, as well as the datasets that gave rise to this finding, were already described in the main text of the original manuscript. We have, nonetheless, expanded the text to further emphasize this aspect of the study. Our detailed responses and

the corresponding revisions are set out below.

Comment B.1. The third point the reply, response B.1.3, answers the criticism with a new hypothesis that is entered into Supplementary Information as a statement of fact, namely: “peptide transport via PfCRT is dependent both on protons and a second solute that remains to be identified, but which is naturally present within the oocyte. All three co-substrates must be present on the same side of the membrane for the transport of host-derived peptides to occur.” This second solute hypothesis is used to explain why it has not been possible to demonstrate the transport of peptides other than the endorphin peptide YPWF-NH₂ from the outside of PfCRT-transformed oocytes, where the second solute may not be available. If the hypothesized second solute can be discovered, will transport of a full set of hemoglobin-derived peptides from outside the PfCRT-transformed oocyte then become demonstrable? Such demonstration might then provide direct support for the claim in the manuscript that “datasets we have obtained using a set of complementary *in vitro* and *in situ* assays reveal that PfCRT exports host-derived peptides containing 4-11 residues out of the parasite’s DV”.

In the meantime, transport from the outside of the PfCRT-transformed oocyte of endorphin peptide YPWF-NH₂, peptide mimic saquinavir, chloroquine and other quinoline drugs is explained by no requirement for (decoupling of) the unknown second solute. In questionable support of this decoupling hypothesis, reference is made to a 2001 *Biochim Biophys Acta* paper on the Human L-type Amino Acid Transporter 1 in Tumor Cell Lines (Yanagida et al.). If the hypothesis of a second solute is correct, and is not a *deus ex machina* for the authors’ model in face of no directly demonstrated transport of hemoglobin-derived peptides from outside the PfCRT-transformed oocytes, what might the second solute be? This is an important question to address, one worth bringing with the hypothesis into the main text and perhaps the Abstract. In response to Reviewer #1, the hypothesized second solute is suggested to be a negatively-charged ion that may help to account for some membrane potential findings, but this evidence is not strong. Responses B.1.1. and B.1.2 provide tangential answers and don’t really meet the criticism of comment B.1. Regarding B.1.1: the ‘H⁺-assays’ are in live parasites, which is obviously a switch from PfCRT-transformed oocytes and brings a whole different system of cellular pathways and experimental measures into the picture. Regarding B.1.2: saquinavir, the peptide mimic, is of course not a hemoglobin peptide and is hypothesized (as with the endorphin peptide) to be decoupled from the unknown second solute.

Our response: Reviewer 2 has incorrectly claimed that we introduced “a new hypothesis” into the revised manuscript and that this hypothesis appears only in the Supplementary Information. Our finding that the transport of peptides via PfCRT is dependent on both protons and a second (yet-to-be identified) co-substrate was already described in the main text of the original manuscript. Reviewer 2 also appears to have overlooked the series of experiments and resulting datasets that led us to ascertain that a second co-substrate was required, as well as the experiments we conducted in an attempt to identify this solute. The relevant paragraphs of the original manuscript were as follows:

“... Moreover, saquinavir *trans*-stimulated [³H]VF-6 transport via all three PfCRT isoforms. Given this result, and the fact that [³H]CQ is readily taken up into the oocyte from the extracellular solution via PfCRT^{Dd2} and PfCRT^{Ecu1110}, we explored whether high extracellular concentrations of CQ would *trans*-stimulate, rather than *trans*-inhibit, the efflux of [³H]VF-6. We observed modest *trans*-stimulation of [³H]VF-6 transport via all three PfCRT isoforms when [CQ] ≥ 500 μM.

These findings led us to test whether a subset of the host-peptides that *trans*-stimulated CQ transport via PfCRT were likewise able to *trans*-stimulate [³H]VF-6 transport. However, all of the test peptides (including unlabelled VF-6) *trans*-inhibited [³H]VF-6 transport via PfCRT, whereas the compounds included as negative controls (LH and free amino acids) were without effect (Fig. 3j). This result, together with the inability of PfCRT to take up [³H]VF-6 from the external solution, indicated that a co-substrate is required for peptide transport via PfCRT (but not for the transport of saquinavir or CQ). This solute is evidently present in the oocyte cytosol, but absent from the extracellular buffer. We screened many solutes, but were unable to identify the co-substrate (Supplementary Table 1). Moreover, re-screening of a range of radiolabelled

metabolites as potential substrates via injection into the oocyte failed to detect transport via PfCRT (Fig. 3k), nor were any of these radiolabelled solutes taken up via PfCRT when added to the external solution (Supplementary Fig. 2).”

In any case, as demonstrated by the large and multifaceted datasets we have presented, discovering the identity of the second co-substrate was not required to measure peptide transport via PfCRT *in vitro* or *in situ*.

We have already addressed Reviewer 2’s assertion that we have not adequately demonstrated peptide transport via PfCRT in the previous Response to Reviewers (e.g. our responses to comments B.1 and C.6). We also note again that Reviewers 1, 3, & 4 are very much of the view that our manuscript has provided robust and compelling demonstrations of peptide transport via PfCRT, including the export of modified peptides and peptide mimics out of the parasite’s digestive vacuole, and view the array of complementary *in vitro* and *in situ* assays we employed as a strength of the study.

Reviewer 2 queried the example we provided of a carrier that translocates some of its substrates with a co-substrate and others without this co-substrate. The carrier we intended to refer to – System γ^+L – mediates the Na⁺-dependent transport of neutral amino acids and the Na⁺-independent transport of cationic amino acids. We believe we have contributed to Reviewer 2’s confusion by mistakenly citing a paper on the related System L amino acid carrier (which does not display this unusual transport property) instead of a paper on System γ^+L . We apologise for this oversight and the confusion it caused.

Regarding the proposal that the second co-substrate could be negatively charged, we believe that it is already clear from the current wording of the text that we are putting forward one possible scenario that is consistent with all of our datasets. The relevant sentences of Supplementary Note 4 are as follows:

“The resulting dataset (Fig. 3d) indicates that the efflux of [³H]VF-6 and its co-substrates involves the movement of a net negative-charge out of the oocyte. Given that the peptide is negatively-charged and is translocated in symport with a proton, one possible scenario would be that the second co-transported ion is also negatively-charged (as this would produce a net negative charge).”

Revisions: As noted above, the finding that the transport of peptides via PfCRT is dependent on both protons and a second (yet-to-be identified) co-substrate, as well as the datasets that gave rise to this discovery and our attempts to identify the co-substrate, were already described in the main text of the original manuscript.

To further emphasize this aspect of the study, we have undertaken the following revisions:

1. The relevant text in the Results section of the main next has been expanded and now reads:

“... Moreover, saquinavir *trans*-stimulated [³H]VF-6 transport via all three PfCRT isoforms. Given this result, and the fact that [³H]CQ is readily taken up into the oocyte from the extracellular solution via PfCRT^{Dd2} and PfCRT^{Ecu1110}, we explored whether high extracellular concentrations of CQ would *trans*-stimulate, rather than *trans*-inhibit, the efflux of [³H]VF-6. We observed modest *trans*-stimulation of [³H]VF-6 transport via all three PfCRT isoforms when [CQ] ≥ 500 μM.

These findings led us to test whether a subset of the host-peptides that *trans*-stimulated CQ transport via PfCRT were likewise able to *trans*-stimulate [³H]VF-6 transport. However, all of the test peptides (including unlabelled VF-6) *trans*-inhibited [³H]VF-6 transport via PfCRT, whereas the compounds included as negative controls (LH and free amino acids) were without effect (Fig. 3j). This result, together with the inability of PfCRT to take up [³H]VF-6 from the external solution, indicated that a co-substrate is required for peptide transport via PfCRT. All three solutes – peptide, proton, and the unidentified co-substrate – must be present on the same side of the membrane for the PfCRT-mediated transport of host-derived peptides to occur. The requirement for a co-substrate was unexpected given that the drugs transported by PfCRT – e.g.

CQ, quinine, quinidine, quinacrine, methylene blue, and saquinavir – only require protons to undergo translocation^{15, 21, 39, 41} (see also Supplementary Note 4). However, the decoupling of some substrates, but not others, from the translocation of a co-substrate is a known phenomenon in carrier-type transporters⁴²⁻⁴⁴. The co-substrate is evidently present in the oocyte cytosol, but absent from the extracellular buffer. We screened many solutes, including Fe²⁺, Fe³⁺, and glutathione, but were unable to identify the co-substrate (Supplementary Data 1). Moreover, re-screening of a range of radiolabelled metabolites as potential substrates via injection into the oocyte failed to detect transport via PfCRT (Fig. 3k), nor were any of these radiolabelled solutes taken up via PfCRT when added to the external solution (Supplementary Fig. 3). Yet within the same assays, [³H]CQ and [³H]hypoxanthine were transported (both out of and into the oocyte) via mutant PfCRT and PfNT1, respectively (Fig. 3k, Supplementary Fig. 3i).”

2. The following summary paragraph has been added to the end of the Introduction (with the text relevant to the co-substrate shown here in bold):

“Here, we use the *Xenopus* oocyte expression system in conjunction with measurements of solute transport, drug activity, and metabolite levels in transgenic parasite lines to provide a detailed elucidation of the native substrate-specificity and physiological role of PfCRT. We show that PfCRT functions to export host-derived peptides 4-11 residues in length out of the parasite’s DV, and that the protein does not behave as a non-specific transporter of other metabolites and/or ions. **The transport of peptides and peptide mimics via PfCRT is saturable, blocked by known PfCRT inhibitors, and is dependent on protons as well as on a co-substrate that remains to be identified.** Relative to the wild-type protein, the drug-resistance-conferring isoforms of PfCRT recognise far fewer peptides and peptide mimics, and have markedly lower maximum rates of peptide transport. The reduced capacities of mutant PfCRT isoforms for transporting peptides out of the DV results in the accumulation of the substrate peptides in drug-resistant parasite lines. Together, our findings provide a molecular basis for why PfCRT is essential for parasite survival and why drug-resistance-conferring mutations in the transporter impart a fitness cost. We present a mechanistic model in which the PfCRT-mediated transport of peptides from the DV to the cytosol serves to (1) provide a source of amino acids to support the parasite’s high growth rate and (2) reduce peptide levels within the DV and thereby prevent the osmotic stress, swelling, and dysfunction of this organelle.”

3. We have added text to the relevant paragraph of the Discussion to highlight our finding regarding the involvement of a co-substrate. The revised text now reads (with the text relevant to the co-substrate shown here in bold):

“Peptide transport via PfCRT shows a slight bias towards negatively-charged peptides, is saturable at micromolar concentrations, and is blocked by known PfCRT inhibitors, including verapamil, chlorpheniramine, and Q₂C. **Moreover, the transport of peptides is dependent on protons as well as on a second solute that remains to be identified, but which is naturally present in the *Xenopus* oocyte.**”

4. We have removed the incorrect reference and replaced it with papers that describe the transport properties of the y⁺L System and the Nrap transporter. The revised text now reads:

“The decoupling of some substrates, but not others, from the translocation of a co-substrate is a known phenomenon in carrier-type transporters. For example, the mammalian System y⁺L mediates the Na⁺-dependent transport of neutral amino acids and the Na⁺-independent transport of cationic amino acids^{20, 21}, and a bacterial Nrap transporter mediates the H⁺-dependent transport of Mn²⁺ as well as the H⁺-independent transport of Cd²⁺²².”

Comment B.2 quoted from the submission and asked that inconsistencies with previously published peptide cis-inhibitory activities and previous claims about shared elements of the ‘CQR reverser pharmacophore’ in inhibitory peptides (Martin RE, et al. Science. 2009. PMID: 19779197) be addressed. The response explains that a buffer improvement was incorporated into the present study and that detection of cis-inhibitory activity are now sensitive enough to detect activities of peptides

that were previously thought to be without effect. The authors also appear to be abandoning some previous ideas about the 'CQR reverser pharmacophore'. These are acceptable shifts of findings and hypothesis; but the reply to B.2 "Revisions: none required" is not acceptable. The inconsistencies with previously published peptide cis-inhibitory activities and previous claims about shared elements of the 'CQR reverser pharmacophore' in inhibitory peptides need to be addressed and explained in the manuscript. (If this has been done, I apologize for missing it; if not done, a few short sentences will do).

Our response: In making these claims about the 'CQ-resistance-reverser' pharmacophore, Reviewer 2 has misconstrued several of our previous papers. In particular, the Reviewer has taken findings and hypotheses that were specific to a mutant isoform of PfCRT – PfCRT^{Dd2} – and has extrapolated them here to apply to PfCRT in general, which is not appropriate or supportable.

For example, in our 2010 *Virulence* paper¹ we wrote the following about the 10 peptides that had inhibited [³H]chloroquine transport via PfCRT^{Dd2} (aka **PfCRT^{CQR}**; ref. [2]) (bold emphasis added):

"Most of the peptides that are active against PfCRT^{CQR} possess the key elements of the 'CQ-resistance reverser' pharmacophore shared by verapamil and the quinoline drugs (a hydrogen bond acceptor and two hydrophobic aromatic rings). Our work suggests that this pharmacophore reflects the core components recognized by PfCRT^{CQR} in its substrates and inhibitors."

We expanded upon this line of thought in our *Cell Mol Life Sci* 2012 paper³ and noted that the observed structural overlap between the CQ-resistance reverser pharmacophore and the known substrates & inhibitors of PfCRT^{Dd2} (which includes the 10 peptides we identified in ref. [2]) may not in fact be a consequence of similarities between these compounds and the native substrate(s) of PfCRT (bold emphasis added).

"... at this stage it is unclear whether the interaction of peptides with PfCRT^{CQR} arises from their resemblance to the endogenous substrate or whether it might instead be due to their structural similarity to verapamil and the quinoline drugs."

Hence, we do not agree that we previously held the CQ-resistance-reverser pharmacophore as being a predictor of the physiochemical properties of the natural substrates of PfCRT. Nor do we believe the current manuscript 'abandons', or is 'inconsistent' with, the view that many of the drugs & other compounds known to be substrates/inhibitors of PfCRT^{Dd2} share elements of the CQ-resistance-reverser pharmacophore. The transporter has a large substrate-binding cavity and the two activities being contrasted here – i.e. 1) the tendency of PfCRT^{Dd2} to bind/translocate drugs & compounds that possess aspects of the 'CQ-resistance-reverser pharmacophore' and 2) its ability to transport host-derived peptides – are not necessarily mutually exclusive. Indeed, our work shows that PfCRT^{Dd2} displays both of these transport functions. Moreover, we found that in gaining the ability to transport these 'CQ-resistance-reverser pharmacophore' drugs & compounds, PfCRT^{Dd2} has undergone a reduction in its capacity to recognise and translocate its native peptide substrates.

The details of the oocyte transport assays used in this study, including the pH of the reaction buffers, are already described in the Methods section. Furthermore, we have already published this version of the method elsewhere⁴ (see Supplementary Figure S3 of ref. [4] for the high signal-to-background ratio obtained, which enabled small differences in chloroquine transport activity to be measured between treatments). Likewise, we have already explained why three of the peptides studied in ref. [2] were not included in this manuscript (the rationale employed to select peptides was described in the Methods section of the original manuscript). Moreover, we do not believe that a description of the steps we have implemented over the last decade to improve the sensitivity of the PfCRT transport assay, and of how these changes have expanded the type, magnitude, and range of transport activities we can detect in the oocyte system, belongs in this manuscript. Such information would be more suited to a specialist review.

Revisions: None required.

Comment B.3 Publications presenting evidence for PfCRT activity as a chloride channel were primarily behind this comment. The authors have now performed experiments that tested for an effect of chloride on [3H]-VF-6 efflux or [3H]-chloroquine influx by PfCRT after replacement of chloride with gluconate. Removal of chloride had no effect. The addition of these findings to the manuscript is appreciated.

Our response: Thank you.

Comment B.4 “Verapamil and Q2C inhibited the peptide-induced H⁺ leaks in C2GC03, C67G8, and C4Dd2 parasites (Fig. 5b), providing further evidence that these solutes are effluxed from the DV via PfCRT.” The reply states that “the solutes” in the sentence was not meant to refer to Verapamil and Q2C, which was an implication in obvious discrepancy with previous publication of Hrycyna et al. “The solutes” was meant instead to refer to peptide mimics and modified peptides. In the revision, the corrected sentence now reads “Verapamil and Q2C inhibited the peptide-induced H⁺ leaks in C2GC03, C67G8, and C4Dd2 parasites (Fig. 5b), providing further evidence that the peptide mimics and modified peptides are effluxed from the DV via PfCRT.” This revision takes care of the matter.

Our response: Thank you.

Comment C.1 The new data on an additional four peptides that are 16 – 25 residues in length is appreciated, as are the findings that these peptides cis-inhibited [3H]-chloroquine transport via PfCRTEcu1110 and PfCRTDd2 but did not trans-stimulate [3H]-chloroquine transport via PfCRT.

Our response: Thank you.

Comment C.2 This reply to this comment relates to reply B.2 above. Different results resulting from methodological changes are of course acceptable, as are new hypotheses. But the reply “Revisions: none required” is not acceptable. The differences attributed to pH change in the authors’ response need to be addressed and explained in the manuscript. (If this has been done, I apologize for missing it; if not done, a few short sentences will do).

Our response: We believe this comment was addressed satisfactorily in the previous Response to the Reviewers. To reiterate two key points, the YPWF-NH₂ peptide is not present in human haemoglobin (and was therefore not included in this study; our approach for peptide selection is already detailed in the Methods section) and the YPWF-NH₂ dataset presented in ref. [2] does not contradict the datasets presented in this manuscript. Moreover, as noted above, we do not agree that this manuscript needs to include a section on the steps we have implemented over the last decade to improve the sensitivity of the PfCRT transport assay, and of how these changes have expanded the type, magnitude, and range of transport activities we can detect in the oocyte system. Such information would be more suited to a specialist review.

Revisions: None required.

Comment C.3 This reply provides a complex answer, especially in view of the new hypothesis that peptide transport via PfCRT is dependent on a second solute, the identity of which remains to be discovered. A system of a multitude of different peptide specificities and transport velocities from the cis side (requiring the undiscovered second solute), with many substrate specificities potentially affected or fully compromised by mutations of PfCRT, is indeed a knotty one to study and clarify by

trans-stimulation experiments. My recommendation here is to include sentences/text from the authors' response here into a paragraph of supplementary information.

Our response: We again note that the requirement for a second co-substrate is not a new hypothesis. This finding, and the experiments that gave rise to it, were presented in the original manuscript. Moreover, we do not agree that the substrate specificities and capacities of the wild-type and mutant isoforms of PfCRT cannot be inferred by *trans*-stimulation transport assays (we have achieved this very task with the datasets presented in this manuscript).

Given that the explanation we provided in the previous Response to the Reviewers helped to clarify the objective of the analysis presented in Fig. 6a (Fig. 7 of the current manuscript), we have included a key sentence from that response in the Fig. 7 legend and have also expanded Supplementary Note 6 to incorporate a more detailed explanation.

Revisions: The relevant sections of revised text now read:

“Fig. 7. The peptide substrates of PfCRT accumulate in parasites expressing mutant isoforms of the transporter. Host-derived peptides in erythrocytes infected with C2^{GC03} or C4^{Dd2} parasites were quantified using tandem liquid-chromatography mass-spectrometry and the peptide levels within the C4^{Dd2} line were expressed relative to those measured in the C2^{GC03} parasites (Supplementary Data 3,5). For peptides containing 4-11 residues, a positive relationship exists between the ability to *trans*-stimulate CQ transport via PfCRT^{3D7} and accumulation within the CQ-resistant C4^{Dd2} parasites. The analysis used the PfCRT^{3D7} *trans*-stimulation dataset, rather than that generated for PfCRT^{Dd2}, because the peptides most likely to accumulate in the C4^{Dd2} line will include those that are very poor substrates of (or no longer transported by) PfCRT^{Dd2}. An analysis of the data with a Bayesian Information Criteria model identified two distinct populations. The *trans*-stimulation data are the mean of 4 independent experiments (each yielding similar results) and the peptide accumulation data are the mean of 2-6 independent experiments (each yielding similar results). Error bars are shown for data points that are $n \geq 3$; the Y error is the SD and the X error is the SEM. The source datasets are provided as a Source Data file.”

“Supplementary Note 6

The correlation between the accumulation within CQ-resistant parasites of peptides 4-11 residues in length and their capacity to *trans*-stimulate PfCRT. Our datasets reveal that wild-type PfCRT transports a broad range of host-derived peptides from the parasite's DV, and that PfCRT^{Ecu1110} and PfCRT^{Dd2} have reduced capacities for peptide transport – both in terms of maximum velocity of transport and the range of peptides translocated. Given that the accumulation of host-derived peptides within the CQ-resistant lines could be explained by the reduced capacities of the mutant PfCRT isoforms for peptide efflux, we investigated the relationship between the peptides found to accumulate within the C4^{Dd2} line and those that had been identified as substrates of PfCRT^{3D7} in the *Xenopus* oocyte system (Fig. 7). It was important to use the PfCRT^{3D7} *trans*-stimulation dataset, rather than that generated for PfCRT^{Dd2}, because the peptides most likely to accumulate in the C4^{Dd2} parasite lines will include those that are very poor substrates of (or no longer transported by) PfCRT^{Dd2}. This comparison identified a positive correlation between the ability of a given host-derived peptide to serve as a substrate of wild-type PfCRT and an increase in its accumulation within the C4^{Dd2} line (consistent with the reduced capacity of PfCRT^{Dd2} for exporting said peptides out of the DV) (Fig. 7).”

Comment C.4 The additional information, new text and supplementary notes are fine.

Our response: Thank you.

Comment C.5 The kinetics in question might be relevant also to the authors' new hypothesis that peptide transport via PfCRT is coupled to a second, undiscovered solute, and that “three co-substrates

must be present on the same side of the membrane for peptide transport to occur” (Response B.1.3).

Our response: We again note that the requirement for a second co-substrate is not a new hypothesis; Reviewer 2 overlooked this finding in the original manuscript. The “kinetics” referred to by Reviewer 2 were the kinetics of inhibition presented in Bellanca et al. *J Biol Chem* 2014⁵. In the Bellanca et al. study, we used detailed inhibition kinetics to identify the particular type of *cis*-inhibition – e.g. pure competitive, mixed-type, or partial mixed-type – exerted by 1) quinine or 2) verapamil on [³H]chloroquine transport via PfCRT^{Dd2}. Since none of these drugs require the second co-substrate for translocation via PfCRT, there is little value in considering the kinetics of their *cis*-inhibition of chloroquine transport via PfCRT^{Dd2} in the context of the second co-substrate.

Revisions: None required.

Comment C.6 The manuscript is substantially improved with the revisions in response to the comments above. However, “robust demonstration of peptide transport via PfCRT” remains to be provided from the transformed oocyte system. Will this be possible if evidence for the “second, undiscovered solute: three co-substrates required on the same side of the membrane for peptide transport to occur” hypothesis can be developed and make a breakthrough? As mentioned above in discussion of response B.1.3, the manuscript would do well to bring this hypothesis forward more prominently and perhaps feature it in the Abstract.

Our response: We again note that Reviewers 1, 3, & 4 are very much of the view that our manuscript has provided robust demonstrations, both *in vitro* and *in situ*, of peptide transport via PfCRT. We also reiterate that the experiments and datasets that led us to ascertain that a second co-substrate was required for peptide transport via PfCRT, as well as our attempts to identify this solute, were presented in the main text of the original manuscript. In any case, as was noted above, knowledge of the identity of the second co-substrate was not required to measure peptide transport via PfCRT *in vitro* or *in situ*.

Revisions: The revisions we have undertaken to the main text to increase the visibility of our findings on the second co-substrate are described in our response to B.1.

Reviewer #3

No new comments or suggestions

Our response: None required.

Reviewer #4

Reviewer #4 (Remarks to the Author):

PfCRT is known as key modulator of multidrug resistance in *P. falciparum*, especially in respect to chloroquine resistance, while the natural substrates of the transporter remain unidentified. Sarah Shafik and colleagues focused on these transporter substrates and systematically tested a library of solute compounds for its *cis*-inhibitory activity on [3H]CQ transport. In their manuscript NCOMMS-19-33349A-Z entitled „The natural function of the malaria parasite’s chloroquine resistance transporter“ the authors distinguished between inhibitors and substrates and pointed out that peptides with 4-11 aa are substrates of PfCRT, as they trans-stimulated the transporter. These data were strengthened by numerous functional and electro-physiological assays with e.g. haemoglobin-derived VF-6 and HM-5

(concentration dependent trans-stimulation), peptide mimetics as trans-stimulator and direct transport of labelled VF-6. The characteristics of the mutant pFCRT isoforms clearly differed from the wildtype form (good overview in Fig. 2f and 2g), and the data explained previous observations in respect to CQ resistance and *P. falciparum* fitness in vivo. The oocyte experiments are here confirmed by analysis in transgenic *P. falciparum*.

I am impressed by the multitude of assays and methods performed in the present study. The manuscript is well structured and written, the figures are coherent and well explained in its legends. The work convinces and, for the first time, gives clear answers on the natural function of PfCRT in Plasmodium. I recommend publication of the present study in Nature Communications after editing of the following point.

The dependence of the transport on the proton-gradient via the vacuole membrane is clearly demonstrated, however, I am not convinced about the role of the co-substrate. Why should it be necessary for peptide transport but not for saquinavir and CQ? The extreme variance in length and aa sequence for the accepted substrates speaks for a high flexibility in the 3D structure of the PfCRT pore. Does the transporter use various mechanisms for inward-directed and outward-directed transport? As PfCRT belongs to the carrier-type transporter, it can transport in both directions and I doubt that it uses various modifications of its pore. Please comment in more detail on these findings.

Our response: We thank the Reviewer for his/her detailed appraisal of our manuscript and for their appreciation of our work – from the breadth and relevance of the assays we employed to the quality and importance of the datasets we generated.

We agree that the requirement for a second co-substrate for peptide transport via PfCRT is unusual given that a number of drugs, including chloroquine and saquinavir, do not need this co-substrate for translocation via PfCRT. However, the decoupling of some substrates, but not others, from the translocation of a co-substrate is not a new phenomenon for carrier-type transporters.

One example we refer to in the manuscript – the mammalian System γ^L – mediates the Na⁺-dependent transport of neutral amino acids and the Na⁺-independent transport of cationic amino acids^{6, 7}. The mechanics of how this is achieved are not known and a structure is not currently available for this carrier. It does appear, however, that the γ^L carrier requires a net positive charge in its binding site before it can undergo translocation and this has resulted in it needing Na⁺ to be co-transported with the neutral amino acids, whereas the cationic amino acids are translocated without Na⁺. Another example is the Nramp transporter, which mediates the H⁺-dependent transport of Mn²⁺ as well as the H⁺-independent transport of Cd²⁺⁸.

In the table shown below, we identify three features that differ (for the most part) between the PfCRT substrates that require the second co-substrate (host-derived peptides) and those that do not (drugs such as chloroquine and saquinavir). However, given that saquinavir was the only peptide mimic or modified peptide to be included in the assay that detected decoupling from the second co-substrate, there is a limit to what can be extrapolated about this phenomenon from our current knowledge.

Furthermore, although a structure has recently been published for PfCRT, much remains to be understood about the molecular mechanics of how it mediates transport. PfCRT belongs to a family of 10 transmembrane-domain Drug/Metabolite Transporters (10-TMD DMTs). The 10-TMD DMTs for which a structure has been determined – including PfCRT – share a unique structure that differs substantially from all other known transporter structures. Moreover, the “alternating-access” transport mechanism that has been proposed for the 10-TMD DMTs on the basis of this unique structure is likewise novel and completely different from all other known transport mechanisms.

Given these restrictions and knowledge limits, it is difficult to arrive at a clear mechanistic model for what determines the independence/dependence of a PfCRT substrate upon the second co-substrate. There are, however, features that are generally present in one group of substrates and not the other.

Feature	Solutes transported with 2 nd co-substrate (host-derived peptides)	Solutes transported without 2 nd co-substrate (chloroquine, quinine, quinidine, quinacrine, methylene blue, saquinavir, amantadine, verapamil, & YPWF-NH ₂)
Zwitterion	YES	NO except for YPWF-NH ₂
Complete peptide backbone	YES	NO except for YPWF-NH ₂
Quinoline moiety	NO	YES except for YPWF-NH ₂ , amantadine, and verapamil

It is worth noting that the three exceptions listed in the far-right column – the peptide YPWF-NH₂ and the drugs amantadine and verapamil – are poor substrates of PfCRT^{Dd2} in the oocyte system. Indeed, of all the substrates characterised in this system to date, YPWF-NH₂, amantadine, and verapamil produced by far the lowest signal-to-background ratios for transport via PfCRT^{Dd2}. Hence, it may well be that the translocation rates of these three solutes would increase in the presence of the second co-substrate. In which case, their transport may be considered to be mostly dependent on the second co-substrate, with a degree of ‘slippage’ occurring. Slippage is defined as the translocation of a substrate without one or more of its usual co-substrates. This property has been observed in a number of carriers⁹⁻¹¹.

These caveats aside, it is possible that one or more of the three features we have identified – zwitterion, complete peptide backbone, and quinoline moiety – determine whether or not the transport of a solute via PfCRT requires the second co-substrate. Further characterisation of this phenomenon in the oocyte system, such as the testing of a broader range of drugs, peptide mimics, and modified peptides for the ability to be transported via PfCRT without the second co-substrate, will be necessary to distinguish between these possibilities.

Revisions: The revisions undertaken to expand the description of our findings on the second co-substrate are outlined in our response to B.1. In addition, we have significantly extended our discussion of this phenomenon in Supplementary Note 4. The relevant text now reads:

“In this regard, we note that several substrates of PfCRT – e.g. CQ, quinine, quinidine, quinacrine, methylene blue, saquinavir, amantadine, verapamil, and the endomorphin peptide YPWF-NH₂ – only require protons to undergo translocation^{1, 6-8}. That is, their transport is decoupled from the second ion, thus enabling measurements of their uptake from the extracellular solution into the oocyte. The decoupling of some substrates, but not others, from the translocation of a co-substrate is a known phenomenon in carrier-type transporters. For example, the mammalian System y^L mediates the Na⁺-dependent transport of neutral amino acids and the Na⁺-independent transport of cationic amino acids^{20, 21}, and a bacterial Nramp transporter mediates the H⁺-dependent transport of Mn²⁺ as well as the H⁺-independent transport of Cd²⁺²².

In the table shown below, we identify three features that differ (for the most part) between the PfCRT substrates that require the second co-substrate and those that do not. However, given that saquinavir was the only peptide mimic or modified peptide to be included in the assay that detected decoupling from the second co-substrate, there is a limit to what can be extrapolated about this phenomenon from our current knowledge.

Feature	Solutes transported with 2 nd co-substrate (host-derived peptides)	Solutes transported without 2 nd co-substrate (chloroquine, quinine, quinidine, quinacrine, methylene blue, saquinavir, amantadine, verapamil, & YPWF-NH ₂)
Zwitterion	YES	NO except for YPWF-NH ₂
Complete peptide backbone	YES	NO except for YPWF-NH ₂
Quinoline moiety	NO	YES except for YPWF-NH ₂ , amantadine, and verapamil

It is worth noting that the three exceptions listed in the far-right column – YPWF-NH₂, amantadine, and verapamil – are poor substrates of PfCRT^{Dd2} in the oocyte system^{1, 6, 7}. Indeed, of all the substrates characterised in this system to date, YPWF-NH₂, amantadine, and verapamil produced by far the lowest signal-to-background ratios for transport via PfCRT^{Dd2}. Hence, it may well be that the translocation rates of these three solutes would increase in the presence of the second co-substrate. In which case, their transport may be considered to be mostly dependent on the second co-substrate, with a degree of ‘slippage’ occurring. Slippage is defined as the translocation of a substrate without one or more of its usual co-substrates. This property has been observed in a number of carriers²³⁻²⁵.

These caveats aside, it is possible that one or more of the three features we have identified – zwitterion, complete peptide backbone, and quinoline moiety – determine whether or not the transport of a solute via PfCRT requires the second co-substrate. Further characterisation of this phenomenon in the oocyte system, such as the testing of a broader range of drugs, peptide mimics, and modified peptides for the ability to be transported via PfCRT without the second co-substrate, will be necessary to distinguish between these possibilities.”

References

1. Summers, R. L., & Martin, R. E. Functional characteristics of the malaria parasite's "chloroquine resistance transporter": implications for chemotherapy. *Virulence*. 10.4161/viru.1.4.12012 **1**, 304-308 (2010).
2. Martin, R. E., Marchetti, R. V., Cowan, A. I., Howitt, S. M., Broer, S., & Kirk, K. Chloroquine transport via the malaria parasite's chloroquine resistance transporter. *Science*. <https://doi.org/10.1126/science.1175667> **325**, 1680-1682 (2009).
3. Summers, R. L., Nash, M. N., & Martin, R. E. Know your enemy: understanding the role of PfCRT in drug resistance could lead to new antimalarial tactics. *Cell Mol Life Sci*. <https://doi.org/10.1007/s00018-011-0906-0> **69**, 1967-1995 (2012).
4. Pulcini, S., Staines, H. M., Lee, A. H., Shafik, S. H., Bouyer, G., Moore, C. M., Daley, D. A., Hoke, M. J., Altenhofen, L. M., Painter, H. J., Mu, J., Ferguson, D. J., Llinas, M., Martin, R. E., Fidock, D. A., Cooper, R. A., & Krishna, S. Mutations in the *Plasmodium falciparum* chloroquine resistance transporter, PfCRT, enlarge the parasite's food vacuole and alter drug sensitivities. *Sci Rep*. <https://doi.org/10.1038/srep14552> **5**, 14552-14567 (2015).
5. Bellanca, S., Summers, R. L., Meyrath, M., Dave, A., Nash, M. N., Dittmer, M., Sanchez, C. P., Stein, W. D., Martin, R. E., & Lanzer, M. Multiple drugs compete for transport via the *Plasmodium falciparum* chloroquine resistance transporter at distinct but interdependent sites. *J Biol Chem*. <https://doi.org/10.1074/jbc.M114.614206> **289**, 36336-36351 (2014).
6. Deves, R., Chavez, P., & Boyd, C. A. Identification of a new transport system (y+L) in human erythrocytes that recognizes lysine and leucine with high affinity. *J Physiol*. <https://doi.org/10.1113/jphysiol.1992.sp019275> **454**, 491-501 (1992).

7. Torrents, D., Estévez, R., Pineda, M., Fernández, E., Lloberas, J., Shi, Y. B., Zorzano, A., & Palacín, M. Identification and characterization of a membrane protein (y⁺L amino acid transporter-1) that associates with 4F2hc to encode the amino acid transport activity y⁺L. A candidate gene for lysinuric protein intolerance. *J Biol Chem*. <https://doi.org/10.1074/jbc.273.49.32437> **273**, 32437-32445 (1998).
8. Bozzi, A. T., Bane, L. B., Zimanyi, C. M., & Gaudet, R. Unique structural features in an Nramp metal transporter impart substrate-specific proton cotransport and a kinetic bias to favor import. *J Gen Physiol*. <https://doi.org/10.1085/jgp.201912428> **151**, 1413-1429 (2019).
9. Bazzone, A., Zabadne, A. J., Salisowski, A., Madej, M. G., & Fendler, K. A Loose Relationship: Incomplete H⁽⁺⁾/Sugar Coupling in the MFS Sugar Transporter GlcP. *Biophys J*. <https://doi.org/10.1016/j.bpj.2017.09.038> **113**, 2736-2749 (2017).
10. Dohán, O., Portulano, C., Basquin, C., Reyna-Neyra, A., Amzel, L. M., & Carrasco, N. The Na⁺/I symporter (NIS) mediates electroneutral active transport of the environmental pollutant perchlorate. *Proc Natl Acad Sci U S A*. <https://doi.org/10.1073/pnas.0707207104> **104**, 20250-20255 (2007).
11. Hussey, G. A., Thomas, N. E., & Henzler-Wildman, K. A. Highly coupled transport can be achieved in free-exchange transport models. *J Gen Physiol*. <https://doi.org/10.1085/jgp.201912437> **152** (2020).